# On the Role of Label Noise in the Feature Learning Process

**Andi Han** [* 1 2]  **Wei Huang** [* 1]  **Zhanpeng Zhou** [* 3]
**Gang Niu** [1]  **Wuyang Chen** [4]  **Junchi Yan** [3]  **Akiko Takeda** [1 5]  **Taiji Suzuki** [1 5]

## Abstract

Deep learning with noisy labels presents significant challenges. In this work, we theoretically characterize the role of label noise from a feature learning perspective. Specifically, we consider a *signal-noise* data distribution, where each sample comprises a label-dependent signal and label-independent noise, and rigorously analyze the training dynamics of a two-layer convolutional neural network under this data setup, along with the presence of label noise. Our analysis identifies two key stages. In *Stage I*, the model perfectly fits all the clean samples (i.e., samples without label noise) while ignoring the noisy ones (i.e., samples with noisy labels). During this stage, the model learns the signal from the clean samples, which generalizes well on unseen data. In *Stage II*, as the training loss converges, the gradient in the direction of noise surpasses that of the signal, leading to overfitting on noisy samples. Eventually, the model memorizes the noise present in the noisy samples and degrades its generalization ability. Furthermore, our analysis provides a theoretical basis for two widely used techniques for tackling label noise: early stopping and sample selection. Experiments on both synthetic and real-world setups validate our theory.

## 1. Introduction

One of the key challenges in deep learning lies in its susceptibility to label noise (Angluin & Laird, 1988). The success of deep learning stems from its exceptional ability to approximate arbitrary functions (Hornik et al., 1989; Funahashi, 1989), yet this ability becomes problematic in the presence of noisy labels. Over-parameterized neural networks, which have sufficient capacity to memorize training data, tend to overfit noisy labels, leading to poor generalization on unseen data (Zhang et al., 2021). Although many studies (Patrini et al., 2017; Ma et al., 2018; Yu et al., 2019; Tanaka et al., 2018; Han et al., 2018; Liu et al., 2023b; Chen et al., 2023a; Xia et al., 2024) have developed methods to mitigate the effects of label noise in practice, our theoretical understanding of how label noise affects the learning process remains incomplete. A crucial step toward bridging this gap is to develop a comprehensive theory describing neural network training dynamics in the presence of label noise.

Existing works have attempted to analyze the effects of label noise theoretically, but many such studies focus on the *lazy training* regime (Jacot et al., 2018; Chizat et al., 2019a). For instance, Li et al. (2020) constrained the distance between model weights and their initialization, while Liu et al. (2023a) adopted an infinitely-wide neural network, where the dynamics can be described by a static kernel function. However, the lazy regime, which typically occurs in wide neural networks with relatively large initialization, is considered undesirable in practice (Chizat et al., 2019a; Ghorbani et al., 2020). Understanding how label noise influences training dynamics *beyond* the lazy regime remains an active research frontier.

Recently, a new line of research (Allen-Zhu & Li, 2020; Frei et al., 2022; Cao et al., 2022; Kou et al., 2023; Xu et al., 2024) has developed the feature learning theory to better capture the complex, nonlinear training behaviors of neural networks. The core idea of the feature learning theory, which dates back to Rumelhart et al. (1986), is to assume a simplified data distribution and analyze how neural networks learn useful *features* from it—an approach that stands in contrast to the linear dynamics observed in lazy training. However, few works have examined how label noise shapes these nonlinear feature-learning dynamics. To address this, we pose the following question:

*How does label noise affect the feature learning process of neural networks?*

**Our main contributions.** In this work, we analyze the

---

[*]Equal contribution [1]RIKEN AIP [2]University of Sydney [3]School of Computer Science & AI, Shanghai Jiao Tong University [4]Simon Fraser University [5]University of Tokyo. Correspondence to: Andi Han <andi.han@sydney.edu.au>, Wei Huang <wei.huang.vr@riken.jp>, Zhanpeng Zhou <zzp1012@sjtu.edu.cn>.

*Proceedings of the $42^{nd}$ International Conference on Machine Learning*, Vancouver, Canada. PMLR 267, 2025. Copyright 2025 by the author(s).

| | | |
|---|---|---|
| **w/ label noise** | *Stage I.* fit all clean samples and ignore noisy ones; learn signal. | Theorem 4.1 |
| | *Stage II.* over-fit noisy samples; learn noise; degrade generalization. | Theorem 4.2 |
| **w/o label noise** | the loss converges; fit all clean samples; generalize well. | Theorem 4.4 |

Table 1: Overview of the two-phase picture and corresponding theoretical results.

feature learning process of neural networks in the presence of label noise, unveiling a two-stage behavior. Our analysis is based on a *signal-noise* data distribution. Specifically, we consider the classification task where each data point consists of a label-dependent *signal* and a label-independent *noise* patch. This simplified data setup has been widely adopted in recent theoretical advances on feature learning (Allen-Zhu & Li, 2020; Frei et al., 2022; Cao et al., 2022; Kou et al., 2023). We introduce label noise by flipping the label into other classes with a certain probability. We then rigorously analyze the training dynamics of a two-layer convolutional neural network under the above setup and characterize the following *two-stage* picture:

- *Stage I.* The model perfectly fits all the clean samples while ignoring the noisy ones[1]. The model mainly learns the signal from clean samples, which generalizes well on unseen data.
- *Stage II.* The training loss converges and the model over-fits to the noisy samples. The model memorizes the noise features from noisy samples, degrading its generalization.

For comparison, we also show that when training without label noise, the model tends to fit all training samples throughout the training, and the test error remains well-bounded. See Table 1 for a summary. Our two-stage picture successfully explains existing empirical phenomena on label noise: neural networks tend to first learn simple patterns from clean samples and then proceed to memorize the noisy ones (Arpit et al., 2017; Han et al., 2018).

Our theoretical analysis also provides valuable insights into the effectiveness of two common techniques used to address the label noise problem, i.e., *early stopping* (Liu et al., 2020; Bai et al., 2021) and *sample selection* (Han et al., 2018; 2020). i) For early stopping, we show that stopping the training process at the end of the first stage ensures a low test error, even in the presence of label noise. ii) For sample selection, we verify that the small-loss criterion [2] (Han et al., 2018) can provably identify the clean samples from noisy ones. These insights ground the path towards more effective techniques tackling the label noise problem.

In summary, our theoretical analysis provides a rich and trackable view of training dynamics under label noise while providing practically relevant insights, a contribution that addresses a notable gap in deep learning theory.

## 2. Related Work

**Feature Learning Theory.** Feature learning (Geiger et al., 2020; Woodworth et al., 2020), or the rich regime, is often used to broadly describe learning behaviors that deviate from the lazy regime. Despite significant interest in exploring the mechanism behind feature learning, our understanding remains limited (Tu et al., 2024). Recent works (Allen-Zhu & Li, 2020; Frei et al., 2022; Allen-Zhu & Li, 2022; Cao et al., 2022; Kou et al., 2023; Zou et al., 2023; Chen et al., 2023b; Huang et al., 2023b; Xu et al., 2024; Bu et al., 2024) have developed a systematic theoretical framework to study feature learning, namely *feature learning theory*. By simplifying the data setup, feature learning theory provides a tractable view of the complex non-linear dynamics of neural networks and has offered significant insights into the success of deep learning. For instance, feature learning theory explains the implicit bias in various architectures, such as convolutional neural networks (Cao et al., 2022; Kou et al., 2023), vision transformers (Jelassi et al., 2022; Li et al., 2023; Jiang et al., 2024), graph neural networks (Huang et al., 2023a), and diffusion models (Han et al., 2024; 2025). Many attempts have also been made to understand different training schemes, including gradient descent with momentum (Jelassi & Li, 2022), Adam (Zou et al., 2021), sharpness-aware minimization (Chen et al., 2024), mixup (Zou et al., 2023; Chidambaram et al., 2023) and contrastive learning (Huang et al., 2024).

**Label Noise Theories.** Many existing works have tried to theoretically analyze the training of neural networks under label noise. Li et al. (2020); Liu et al. (2023a) studied label noise in the lazy-training regime (Jacot et al., 2018; Chizat et al., 2019b), constraining the distance between model weights and their initialization, and thus cannot capture the process of learning features from data (Rumelhart et al., 1986; Damian et al., 2022). On the other hand, Frei et al. (2021) focused on the early dynamics of neural networks trained with SGD, which can be largely explained by the behavior of a linear classifier (Kalimeris et al., 2019; Hu et al., 2020), thereby overlooking the adverse effects of label noise on generalization in later training stages.

---

[1]To clarify, "clean samples" refers to samples without noisy labels, while "noisy samples" refers to those with noisy labels. The same terminology applies for the rest of this paper.

[2]The samples with small loss are more likely to be the ones which are correctly labeled, i.e., the clean samples.

*In comparison*, our work characterizes the whole training process of neural networks under label noise from a feature-learning perspective, unveiling a novel two-stage behavior in the training dynamics. Some recent works (Kou et al., 2023; Meng et al., 2023; Xu et al., 2024) in feature learning theory also considered label noise in their analysis. However, they fail to characterize the two-stage distinct behaviors due to differences in their setups.

## 3. Problem Setup

**Basic Notation.** We use bold letters for vectors and matrices, and scalars otherwise. The Euclidean norm of a vector and spectral norm of a matrix are denoted by $\|\cdot\|_2$, and the Frobenius norm of a matrix by $\|\cdot\|_F$. We use $y$ to represent the true label and $\tilde{y}$ to represent the observed label. For a logical variable $a$, let $\mathbb{1}(a) = 1$ if $a$ is true, otherwise $\mathbb{1}(a) = 0$. Denote $\mathbf{I}_d$ as the $d \times d$ identity matrix and $[n] = \{1, 2, \ldots, n\}$.

**Binary Classification.** We consider a binary-classification data distribution $\mathcal{D}_{\text{tr}}$[3], following prior works (Cao et al., 2022; Kou et al., 2023), where each data point consists of a label-dependent *signal* and a label-independent *noise*. More precisely, let $\boldsymbol{\mu} \in \mathbb{R}^d$ be a fixed vector representing the signal and let $\boldsymbol{\xi} \in \mathbb{R}^d$ be a random vector sampled from $\mathcal{N}(\mathbf{0}, \sigma_\xi^2 \mathbf{I}_d)$ representing the noise. Then each data point $\mathbf{x} \in \mathbb{R}^{2d}$ is defined as $\mathbf{x} = [\mathbf{x}^{(1)}, \mathbf{x}^{(2)}]$, where one of $\mathbf{x}^{(1)}, \mathbf{x}^{(2)}$ is chosen to be $y\boldsymbol{\mu}$ and the other as $\boldsymbol{\xi}$, Here $y \in \{-1, 1\}$ is the corresponding label, generated from the Rademacher distribution, i.e., $\mathbb{P}(y = 1) = \mathbb{P}(y = -1) = \frac{1}{2}$. We sample the training set $\{\mathbf{x}_i, y_i\}_{i=1}^n$ from $\mathcal{D}_{\text{tr}}$ and assume the training set is balanced without loss of generality.

**Label Flipping.** We introduce label noise for each training sample, where the observed label $\tilde{y}$ may differ from the ground-truth label $y$ [4]. Specifically, we flip the observed label into the incorrect class with a certain probability, i.e., $\tau_+ = \mathbb{P}(\tilde{y} = -1 | y = 1), \tau_- = \mathbb{P}(\tilde{y} = 1 | y = -1)$. We require $\tau_+, \tau_- \in (0, 1/2)$ such that neural networks can still learn the signal from data under label noise. In addition, we denote the clean sample set as $\mathcal{S}_t := \{i \in [n] : \tilde{y}_i = y_i\}$ and the noisy sample set as $\mathcal{S}_f := \{i \in [n] : \tilde{y}_i \neq y_i\}$.

**Two-Layer ReLU CNN.** We consider a two-layer convolutional neural network (CNN) with ReLU activation, as in (Cao et al., 2022; Kou et al., 2023). Formally, given the input data $\mathbf{x}$, the output of the neural network is defined as $f(\mathbf{W}, \mathbf{x}) = F_{+1}(\mathbf{W}_{+1}, \mathbf{x}) - F_{-1}(\mathbf{W}_{-1}, \mathbf{x})$, where

---

[3]The analysis can be extended to multi-class classification, where each class is represented by distinct, orthogonal signal vector, i.e., $\boldsymbol{\mu}_1, \boldsymbol{\mu}_2, \ldots, \boldsymbol{\mu}_k$, along with one-hot labels.

[4]In experiments, the ground-truth labels are inaccessible, and we can only evaluate models based on the observed labels.

$F_{+1}(\mathbf{W}_{+1}, \mathbf{x})$ and $F_{-1}(\mathbf{W}_{-1}, \mathbf{x})$ are given by

$$F_j(\mathbf{W}_j, \mathbf{x}) = \frac{1}{m} \sum_{r=1}^m \Big( \sigma\big(\langle \mathbf{w}_{j,r}, y\boldsymbol{\mu} \rangle\big) + \sigma\big(\langle \mathbf{w}_{j,r}, \boldsymbol{\xi} \rangle\big) \Big),$$

for $j = \pm 1$. Here, $\sigma(\cdot)$ is the ReLU activation function, defined as $\sigma(x) = \max\{0, x\}$. This corresponds to a two-layer CNN with second layer weights fixed to be $\pm 1/m$.

We initialize the entries of $\mathbf{W}$ independently from a zero-mean Gaussian distribution with variance $\sigma_0^2$, i.e., $\mathbf{w}_{j,r}^{(0)} \sim \mathcal{N}(0, \sigma_0^2 \mathbf{I}_d)$ for all $j = \pm 1, r \in [m]$. We also fix the second layer weights uniformly to $\pm 1$, which is a common setting used in previous studies (Arora et al., 2019; Chatterji et al., 2021).

**Gradient Descent with Logistic Loss.** We consider the logistic loss $\ell(f, \tilde{y}) = \log\big(1 + \exp(-f \cdot \tilde{y})\big)$. Then the training loss, or the empirical risk, can be written as:

$$L_S(\mathbf{W}) = \frac{1}{n} \sum_{i=1}^n \ell\left(f(\mathbf{W}, \mathbf{x}_i), \tilde{y}_i\right),$$

To minimize this empirical risk, we use gradient descent (GD) with a constant learning rate $\eta > 0$,

$$\begin{aligned}
\mathbf{w}_{j,r}^{(t+1)} &= \mathbf{w}_{j,r}^{(t)} - \eta \cdot \nabla_{\mathbf{w}_{j,r}} L_S(\mathbf{W}^{(t)}) \\
&= \mathbf{w}_{j,r}^{(t)} - \frac{\eta}{nm} \sum_{i=1}^n \ell_i'^{(t)} \sigma'(\langle \mathbf{w}_{j,r}^{(t)}, \boldsymbol{\xi}_i \rangle) j \tilde{y}_i \boldsymbol{\xi}_i \\
&\quad - \frac{\eta}{nm} \sum_{i=1}^n \ell_i'^{(t)} \sigma'(\langle \mathbf{w}_{j,r}^{(t)}, y_i \boldsymbol{\mu} \rangle) j y_i \tilde{y}_i \boldsymbol{\mu}, \quad (1)
\end{aligned}$$

where the loss derivative $\ell_i'^{(t)} := \ell'(f(\mathbf{W}^{(t)}, \mathbf{x}_i), \tilde{y}_i) = -\frac{1}{1 + \exp(\tilde{y}_i f(\mathbf{W}^{(t)}, \mathbf{x}_i))}$.

**Evaluation.** We characterize the generalization by evaluating the 0-1 error on the test distribution $\mathcal{D}_{\text{test}}$:

$$L_D^{0-1}(\mathbf{W}) = \mathbb{P}_{(\mathbf{x}, y) \sim \mathcal{D}_{\text{test}}}(y \cdot f(\mathbf{W}, \mathbf{x}) < 0)$$

$\mathcal{D}_{\text{test}}$ mainly follows the settings of $\mathcal{D}_{\text{tr}}$; however, to simulate spurious features in real-world scenarios, for any $(\mathbf{x}, y) \in \mathcal{D}_{\text{test}}$, we define $\mathbf{x} = [y\boldsymbol{\mu}, \boldsymbol{\xi} + \boldsymbol{\zeta}]$, where $\boldsymbol{\xi} \sim \text{Unif}(\{\boldsymbol{\xi}_i\}_{i=1}^n)$ and $\boldsymbol{\zeta} \sim \mathcal{N}(0, \sigma_\xi^2 \mathbf{I})$. In practice, spurious features exist, which occur in both the training and test set but lack causal relationships with the ground-truth label $y$ (Sagawa et al., 2020; Zhou et al., 2021; Singla & Feizi, 2021; Izmailov et al., 2022). We consider the label-independent noise $\boldsymbol{\xi}$ in the training set as spurious features and randomly incorporate them into the test distribution.

**Signal-Noise Decomposition.** In our analysis, we utilize a proof technique termed *signal-noise decomposition*, which has been widely adopted by (Li et al., 2019; Allen-Zhu &

Li, 2020; 2022; Cao et al., 2022). The signal-noise decomposition breaks down the weight $\mathbf{w}_{j,r}^{(t)}$ into signal and noise components. Formally, we express:

$$\mathbf{w}_{j,r}^{(t)} = \mathbf{w}_{j,r}^{(0)} + j\gamma_{j,r}^{(t)}\|\boldsymbol{\mu}\|_2^{-2}\boldsymbol{\mu} + \sum_{i=1}^{n}\rho_{j,r,i}^{(t)}\|\boldsymbol{\xi}_i\|_2^{-2}\boldsymbol{\xi}_i, \quad (2)$$

where $\gamma_{j,r}^{(t)}$ and $\rho_{j,r,i}^{(t)}$ represent the signal and noise coefficients, respectively. The normalization factors $\|\boldsymbol{\mu}\|_2^{-2}$ and $\|\boldsymbol{\xi}_i\|_2^{-2}$ ensure that $\gamma_{j,r}^{(t)} \approx \langle\mathbf{w}_{j,r}^{(t)}, \boldsymbol{\mu}\rangle$, and $\rho_{j,r,i}^{(t)} \approx \langle\mathbf{w}_{j,r}^{(t)}, \boldsymbol{\xi}_i\rangle$. Naturally, $\gamma_{j,r}^{(t)}$ characterizes the process of signal learning, while $\rho_{j,r,i}^{(t)}$ captures the memorization of noise.

To facilitate a finer-grained analysis of the evolution of the noise coefficients, we introduce the notations $\overline{\rho}_{j,r,i}^{(t)} := \rho_{j,r,i}^{(t)}\mathbb{1}(\rho_{j,r,i}^{(t)} \geq 0)$, $\underline{\rho}_{j,r,i}^{(t)} := \rho_{j,r,i}^{(t)}\mathbb{1}(\rho_{j,r,i}^{(t)} \leq 0)$, following (Cao et al., 2022). Consequently, the weight decomposition can be further expressed as:

$$\mathbf{w}_{j,r}^{(t)} = \mathbf{w}_{j,r}^{(0)} + j\gamma_{j,r}^{(t)}\|\boldsymbol{\mu}\|_2^{-2}\boldsymbol{\mu}$$
$$+ \sum_{i=1}^{n}\overline{\rho}_{j,r,i}^{(t)}\|\boldsymbol{\xi}_i\|_2^{-2}\boldsymbol{\xi}_i + \sum_{i=1}^{n}\underline{\rho}_{j,r,i}^{(t)}\|\boldsymbol{\xi}_i\|_2^{-2}\boldsymbol{\xi}_i. \quad (3)$$

This decomposition translates the dynamics of weights into dynamics of signal and noise coefficients (See Lemma C.1).

# 4. Main Results

Before presenting our main results, we first state our main condition.

**Notations.** We denote $\text{SNR} := \|\boldsymbol{\mu}\|_2/(\sigma_\xi\sqrt{d})$ to be the signal-to-noise ratio and $T^* = \widetilde{\Theta}(\eta^{-1}\epsilon^{-1}nm\sigma_\xi^{-1}d^{-1})$ to be the maximum iterations for any given $\epsilon > 0$.

**Condition 4.1.** Suppose there exists a sufficiently large constant $C$ such that the following holds.

1. The signal-to-noise ratio and label flipping probability satisfy $n \cdot \text{SNR}^2 = \Theta(1)$, $\tau_+, \tau_- = \Theta(1)$.

2. The dimension of a single data patch $d$ satisfies $d \geq C\max\left\{n^2\log(nm/\delta)\log(T^*)^2, n\|\boldsymbol{\mu}\|_2\sigma_\xi^{-1}\sqrt{\log(n/\delta)}\right\}$.

3. The size of training sample $n$ and model width $m$ satisfy $m \geq C\log(n/\delta), n \geq C\log(m/\delta)$.

4. The magnitude of signal patch $\|\boldsymbol{\mu}\|_2$ satisfies $\|\boldsymbol{\mu}\|_2^2 \geq C\sigma_\xi^2\log(n/\delta)$.

5. The standard deviation $\sigma_0$ of the Gaussian distribution for weights initialization satisfies $\sigma_0 \leq C^{-1}\min\left\{\sqrt{n}\sigma_\xi^{-1}d^{-1}, \|\boldsymbol{\mu}\|_2^{-1}\log(m/\delta)^{-1/2}\right\}$.

6. The positive constant learning rate $\eta$ satisfies $\eta \leq C^{-1}\min\left\{\sigma_\xi^{-2}d^{-3/2}n^2m\sqrt{\log(n/\delta)}, \sigma_\xi^{-2}d^{-1}n\right\}$.

**Remarks on Condition 4.1.** We first require the signal-to-noise ratio $n \cdot \text{SNR}^2$ and label flipping probability $\tau_+, \tau_-$ to be of constant order. Such a condition is critical for the subsequent characterization of the two-stage behaviors. Other conditions, including sufficiently large $d, m, n, \|\boldsymbol{\mu}\|_2$ and sufficiently small $\sigma_0, \eta$, are commonly adopted in existing analysis (Cao et al., 2022; Kou et al., 2023; Meng et al., 2023), to ensure sufficient over-parameterization and that the training loss converges under gradient descent.

## 4.1. Feature Learning Process with Label Noise

In this subsection, we analyze the feature learning process of neural networks under label noise. We identify two stages where the learning dynamics exhibit distinct behaviors.

**Stage I. Model Fits Clean Data.** Theorem 4.1 characterizes the learning outcome at the end of Stage I.

**Theorem 4.1.** *Under Condition 4.1, there exists $T_1 = \Theta\left(\eta^{-1}nm\sigma_\xi^{-2}d^{-1}\right)$ such that $\overline{\rho}_{\tilde{y}_i,r,i}^{(T_1)} = \Theta(1)$ for all $i \in [n]$, $r \in [m]$ with $\langle\mathbf{w}_{\tilde{y}_i,r}^{(0)}, \boldsymbol{\xi}_i\rangle \geq 0$ and $\gamma_{j,r}^{(T_1)} = \Theta(1)$ for all $j = \pm 1, r \in [m]$, and*

1. *$\gamma_{j,r}^{(T_1)} > \overline{\rho}_{\tilde{y}_i,r,i}^{(T_1)}$ for all $j = \pm 1, r \in [m], i \in [n]$.*

2. *All clean samples $i \in \mathcal{S}_t$ satisfy $\tilde{y}_i f(\mathbf{W}^{(T_1)}, \mathbf{x}_i) \geq 0$.*

3. *All noisy samples $i \in \mathcal{S}_f$ satisfy $\tilde{y}_i f(\mathbf{W}^{(T_1)}, \mathbf{x}_i) \leq 0$.*

Theorem 4.1 states that at the end of Stage I, the signal coefficients $\gamma_{j,r}^{(T_1)}$ are larger than the noise coefficients $\overline{\rho}_{\tilde{y}_i,r,i}^{(T_1)}$, suggesting signal learning dominates the feature learning process. During this stage, the model correctly classifies all clean samples, i.e., $\forall i \in \mathcal{S}_t, \tilde{y}_i f(\mathbf{W}^{(T_1)}, \mathbf{x}_i) \geq 0$, and classifies all noisy samples to the ground-truth classes, i.e., $\forall i \in \mathcal{S}_f, y_i f(\mathbf{W}^{(T_1)}, \mathbf{x}_i) = -\tilde{y}_i f(\mathbf{W}^{(T_1)}, \mathbf{x}_i) \geq 0$.

**Insights from Theorem 4.1.** In Stage I, the model perfectly fits all the clean samples while disregarding the noisy ones. The model initially learns the signal from the data, leading to an optimal generalization ability.

**Stage II. Loss Converges and Model Fits Noisy Data.** Theorem 4.2 formalizes the learning behavior in Stage II as the training loss converges.

**Theorem 4.2.** *Under Condition 4.1, for arbitrary $\epsilon > 0$, there exists $t^* \in [T_1, T^*]$, such that training loss converges, i.e., $L_S(\mathbf{W}^{(t^*)}) \leq \epsilon$ and*

1. *All clean samples, i.e., $i \in \mathcal{S}_t$, it holds that $\tilde{y}_i f(\mathbf{W}^{(t^*)}, \mathbf{x}_i) \geq 0$.*

2. *There exists a constant $0 < \tau' \leq \frac{\tau_+ + \tau_-}{2}$ such that there are $\tau'n$ noisy samples, i.e., $i \in \mathcal{S}_f$ that satisfy $\frac{1}{m}\sum_{r=1}^{m}\overline{\rho}_{\tilde{y}_i,r,i}^{(t^*)} > \frac{1}{m}\sum_{r=1}^{m}\gamma_{-\tilde{y}_i,r}^{(t^*)}$ and $\tilde{y}_i f(\mathbf{W}^{(t^*)}, \mathbf{x}_i) \geq 0$.*

*3. Test error $L_D^{0-1}(\mathbf{W}^{(t^*)}) \geq 0.5 \min\{\tau_+, \tau_-\}$.*

Theorem 4.2 states that in Stage II, as the training loss converges, the model continues to make correct predictions on clean samples, consistent with Stage I; however, for certain noisy samples, the averaged noise coefficient across all neurons surpasses the averaged signal coefficient, i.e., $\frac{1}{m}\sum_{r=1}^{m} \overline{\rho}_{\tilde{y}_i,r,i}^{(t^*)} > \frac{1}{m}\sum_{r=1}^{m} \gamma_{-\tilde{y}_i,r}^{(t^*)}$. Consequently, for these noisy samples, the model's predictions align with the noisy observed labels $\tilde{y}$. In addition, if evaluating the model on the test distribution introduced in Section 3, it results in a constant, non-vanishing test error, which is lower-bounded by the label flipping probability in the training set.

**Insights from Theorem 4.2.** With the presence of label noise and sufficient training iterations, the model inevitably learns the noise features from the data, which leads to degraded generalization.

### 4.2. Theoretical Supports for Techniques: Early Stopping and Sample Selection.

Our two-stage picture provides theoretical support for two widely used techniques to address label noise: *early stopping* (Liu et al., 2020; Bai et al., 2021) and *sample selection* (Han et al., 2018; 2020). Intuitively, early stopping aims to stop training before the loss converges, preventing overfitting to noisy samples. On the other hand, sample selection leverages the small-loss criterion (Han et al., 2018) to distinguish clean samples from noisy ones, assuming that samples with smaller losses are more likely to be clean. Proposition 4.3 formally supports the effectiveness of these two strategies.

**Proposition 4.3** (Early stopping and sample selection). *Under the same conditions as in Theorem 4.1, we have that*

- *Early stopping: suppose the training is terminated early at $T_1$, then the test loss is bounded by $L_D^{0-1}(\mathbf{W}^{(T_1)}) \leq \exp(-dn^{-1}/C')$ for some constant $C' > 0$.*

- *Sample selection: At iteration $T_1$, clean and noisy samples can be well-separated based on the training loss, i.e., for all $i \in \mathcal{S}_t$, $\ell_i^{(T_1)} \leq \log(2)$ and for all $i \in \mathcal{S}_f$, $\ell_i^{(T_1)} \geq \log(2)$.*

Proposition 4.3 implies that if training is stopped during Stage I $t \leq T_1$, before the loss converges, the test error can be upper bounded arbitrarily small under the condition that $d = \tilde{\Omega}(n^2)$. Proposition 4.3 also states that noisy samples tend to have higher loss values compared to clean samples, and there exists a hard threshold $\log(2)$, which allows for a perfect separation of clean samples from noisy ones.

**Remarks on Proposition 4.3.** We note that directly computing $T_1$ in real-world scenarios is not feasible due to the

complexity of the training dynamics and unknown data distribution. However, our theory suggests the existence of a point $T_1$, beyond which further training may degrade generalization performance. This implies that validation accuracy can serve as a practical surrogate for identifying , which aligns with the common practice of early stopping at the point of maximum validation accuracy. Indeed, our theory explains the effectiveness of the common practice.

### 4.3. Feature learning Process without Label Noise

For comparison, we also analyze the feature learning process of neural networks without label noise. The analysis follows a similar two-stage framework as the analysis conducted with label noise.

**Model Fits All Data and Generalizes Well.** Theorem 4.4 characterizes the learning outcome without label noise.

**Theorem 4.4.** *Under Condition 4.1 with $\tau_+, \tau_- = 0$, there exists $T_1 = \Theta(\eta^{-1}\epsilon^{-1}nm\sigma_\xi^{-2}d^{-1})$ such that for all $T_1 \leq t \leq T^*$, $y_i f(\mathbf{W}^{(t)}, \mathbf{x}_i) \geq 0$ for all $i \in [n]$. In addition, there exists a time $t^* \in [T_1, T^*]$ such that training loss converges, i.e., $L_S(\mathbf{W}^{(t^*)}) \leq \epsilon$ and*

1. *$y_i f(\mathbf{W}^{(t^*)}, \mathbf{x}_i) \geq 0$ for all $i \in [n]$,*

2. *Test error is bounded as $L_D^{0-1}(\mathbf{W}^{(t^*)}) \leq \exp\left(\frac{d}{n} - \frac{n\|\boldsymbol{\mu}\|_2^4}{C_D\sigma_\xi^4 d}\right)$ for some constant $C_D > 0$.*

Theorem 4.4 states that without label noise, all samples can be classified correctly at the end of training, suggesting that the model fits all the samples throughout the training process. Theorem 4.4 also shows that when evaluating, the test error can be upper bounded by $\exp(\frac{d}{n} - \frac{n\|\boldsymbol{\mu}\|_2^4}{C_D\sigma_\xi^4 d})$. Thus, if $n \cdot \text{SNR}^2 \geq 2C_D = \Theta(1)$, it follows that $L_D^{0-1}(\mathbf{W}^{(t^*)}) \leq \exp(-\frac{n\|\boldsymbol{\mu}\|^4}{2C_D\sigma_\xi^4 d})$, which is small given the requirement on $d$ (see Condition 4.1).

## 5. Proof Sketch

In this section, we provide an overview of our proof techniques for our theoretical results in Section 4.

**Technical Novelty.** Compared with existing works in feature learning theory, we mainly introduce two novel conditions: i) $n \cdot \text{SNR}^2$ and ii) $\tau_+, \tau_-$ are of the constant order. These two conditions establish a significantly different training regime and are crucial for identifying the two-stage picture in learning dynamics. We delve into the technical details of these distinctions in the subsections below and in Appendix A.

## 5.1. Feature Learning Process with Label Noise

The analysis of feature learning with label noise critically relies on Lemma 5.1, which shows the difference in terms of model predictions between clean and noisy samples.

**Lemma 5.1.** *Under Condition 4.1, there exists a sufficiently large constant $C_1$ such that for all $t \in [0, T^*]$, the following are satisfied:*

- $\frac{1}{m} \sum_{r=1}^{m} \left( \gamma_{\tilde{y}_i,r}^{(t)} + \overline{\rho}_{\tilde{y}_i,r,i}^{(t)} \right) - 1/C_1 \leq \tilde{y}_i f(\mathbf{W}^{(t)}, \mathbf{x}_i) \leq \frac{1}{m} \sum_{r=1}^{m} \left( \gamma_{\tilde{y}_i,r}^{(t)} + \overline{\rho}_{\tilde{y}_i,r,i}^{(t)} \right) + 1/C_1$ *for all clean samples, i.e., $i \in \mathcal{S}_t$.*

- $\frac{1}{m} \sum_{r=1}^{m} \left( \overline{\rho}_{\tilde{y}_i,r,i}^{(t)} - \gamma_{-\tilde{y}_i,r}^{(t)} \right) - 1/C_1 \leq \tilde{y}_i f(\mathbf{W}^{(t)}, \mathbf{x}_i) \leq \frac{1}{m} \sum_{r=1}^{m} \left( \overline{\rho}_{\tilde{y}_i,r,i}^{(t)} - \gamma_{-\tilde{y}_i,r}^{(t)} \right) + 1/C_1$ *for all noisy samples, i.e., $i \in \mathcal{S}_f$.*

Lemma 5.1 suggests for clean samples $i \in \mathcal{S}_t$, the model prediction $\tilde{y}_i f(\mathbf{W}^{(t)}, \mathbf{x}_i)$ is determined by $\frac{1}{m} \sum_{r=1}^{m} \left( \gamma_{\tilde{y}_i,r}^{(t)} + \overline{\rho}_{\tilde{y}_i,r,i}^{(t)} \right)$, while for noisy samples $i \in \mathcal{S}_f$, $\tilde{y}_i f(\mathbf{W}^{(t)}, \mathbf{x}_i)$ is determined by $\frac{1}{m} \sum_{r=1}^{m} \left( \overline{\rho}_{\tilde{y}_i,r,i}^{(t)} - \gamma_{-\tilde{y}_i,r}^{(t)} \right)$.

Besides Lemma 5.1, we also need to bound the scale of coefficients throughout the training process.

**Proposition 5.2.** *Under Condition 4.1, for any $0 \leq t \leq T^*$, we can bound*

$$0 \leq \overline{\rho}_{j,r,i}^{(t)}, \gamma_{j,r}^{(t)} \leq \Theta(\log(T^*)),$$
$$0 \geq \underline{\rho}_{j,r,i}^{(t)} \geq -\widetilde{O}(\max\{\sigma_0 \|\boldsymbol{\mu}\|_2, \sigma_0 \sigma_\xi \sqrt{d}, n d^{-1/2}\}).$$

Proposition 5.2 states that $|\underline{\rho}_{j,r,i}^{(t)}|$ is lower bounded by a small term based on Condition 4.1. In addition, both $\overline{\rho}_{j,r,i}^{(t)}, \gamma_{j,r}^{(t)}$ are positive and cannot grow faster than a logarithmic order of $T^*$.

**Comparison with Previous Studies.** Some previous works also need to bound the scale of coefficients during training; however, their analysis is not applicable to our case due to our condition that $n \cdot \mathrm{SNR}^2 = \Theta(1)$. As a representative example, the analysis in Kou et al. (2023) requires the automatic balance of updates, i.e., the loss derivatives $\ell_i'^{(t)}$ are balanced across all samples. However, in our case, due to the SNR condition, signal coefficients are on the same scale as noise coefficients. Consequently, the loss derivatives $\ell_i'^{(t)}$ are no longer solely determined by $\frac{1}{m} \sum_{r=1}^{m} \overline{\rho}_{\tilde{y}_i,r,i}^{(t)}$. Thus, the automatic balance of updates cannot be derived.

**Two-Stage Analysis.** To prove Proposition 5.2 as well as the main theorems in Table 1 under our Condition 4.1, we separately consider two stages.

In *Stage I*, before the maximum of the coefficients reaches a constant order, all loss derivatives can be lower bounded by

a constant, i.e., $|\ell_i'^{(t)}| \geq C_\ell$ for all $i \in [n]$. This ensures the balance of loss derivatives across all samples as $|\ell_i'^{(t)}| \leq 1$. Such a condition allows both $\overline{\rho}_{j,r,i}^{(t)}, \gamma_{j,r}^{(t)}$ to increase to a constant order, enabling the establishment of the bound in Proposition 5.2. Furthermore, by applying Lemma 5.1, we can assert that $\tilde{y}_i f(\mathbf{W}^{(t)}, \mathbf{x}_i) \geq 0$ for all $i \in \mathcal{S}_t$. On the other hand, as long as $n \cdot \mathrm{SNR}^2 \geq c'$, for some constant $c' > 0$, we can demonstrate that signal learning slightly surpasses noise memorization, concluding that $\tilde{y}_i f(\mathbf{W}^{(t)}, \mathbf{x}_i) \leq 0$ for noisy samples $i \in \mathcal{S}_f$.

In *Stage II*, after the coefficients reach a constant order, the loss derivatives can no longer be lower-bounded by a constant. To establish that Proposition 5.2 still holds in this stage, we first rewrite the signal learning dynamics in Lemma C.1 as follows:

$$\gamma_{j,r}^{(t+1)} = \gamma_{j,r}^{(t)} + \frac{\eta}{nm} \Big( \sum_{i \in \mathcal{S}_t} |\ell_i'^{(t)}| \mathbb{1}(\langle \mathbf{w}_{j,r}^{(t)}, y_i \boldsymbol{\mu} \rangle \geq 0)$$
$$- \sum_{i \in \mathcal{S}_f} |\ell_i'^{(t)}| \mathbb{1}(\langle \mathbf{w}_{j,r}^{(t)}, y_i \boldsymbol{\mu} \rangle \geq 0) \Big) \|\boldsymbol{\mu}\|_2^2.$$

Recall $|\ell_i'^{(t)}| = \frac{1}{1 + \exp(\tilde{y}_i f(\mathbf{W}^{(t)}, \mathbf{x}_i))}$. Based on Lemma 5.1, $|\ell_i'^{(t)}| \mathbb{1}(i \in \mathcal{S}_f)$ can be larger than $|\ell_i'^{(t)}| \mathbb{1}(i \in \mathcal{S}_t)$, which causes $\gamma_{j,r}^{(t)}$ to decrease. However, we show by contradiction that $\gamma_{j,r}^{(t)} \leq 0$ cannot occur. When $\gamma_{j,r}^{(t)}$ decreases, the gap between $|\ell_i'^{(t)}| \mathbb{1}(i \in \mathcal{S}_t)$ and $|\ell_i'^{(t)}| \mathbb{1}(i \in \mathcal{S}_f)$ diminishes, allowing $\gamma_{j,r}^{(t)}$ to eventually increase in subsequent iterations. Therefore, the upper bound for both $\gamma_{j,r}^{(t)}, \overline{\rho}_{j,r,i}^{(t)}$ can be derived by showing that $|\ell_i'^{(t)}|$ converges at a rate of $O(1/t)$ when either $\gamma_{j,r}^{(t)}$ or $\overline{\rho}_{j,r,i}^{(t)}$ grows to a logarithmic order.

Additionally, we demonstrate that training loss converges at some iteration $t^*$. Upon convergence, because of the monotonicity of $\overline{\rho}_{\tilde{y}_i,r,i}^{(t)}$ and positivity of $\gamma_{j,r}^{(t)}$, we can show $\tilde{y}_i f(\mathbf{W}^{t^*}, \mathbf{x}_i) \geq 0$ for all $i \in \mathcal{S}_t$. On the other hand, we establish by contradiction that there must exist a constant fraction of noisy samples satisfying $\tilde{y}_i f(\mathbf{W}^{t^*}, \mathbf{x}_i) \geq 0$. Because otherwise, we would have $\frac{1}{m} \sum_{r=1}^{m} (\overline{\rho}_{\tilde{y}_i,r,i}^{(t^*)} - \gamma_{-\tilde{y}_i,r}^{(t^*)}) \leq C_\epsilon$ for some constant $C_\epsilon > 0$ over a constant fraction of samples. This suggests the training loss $L_S(\mathbf{W}^{(t^*)})$ can be lower bounded by a strictly positive constant $c_l > 0$, which leads to a contradiction.

Finally, to establish the test error, we show that, based on the test distribution $\mathcal{D}_{\text{test}}$ introduced in Section 3, there exists some sufficiently large constant $C'$ that

$$\mathbb{P}(y f(\mathbf{W}^{(t^*)}, \mathbf{x}) < 0)$$
$$\geq \frac{1}{n} \sum_{i=1}^{n} \mathbb{P} \Big( \frac{1}{m} \sum_{r=1}^{m} \sigma(\langle \mathbf{w}_{-y,r}^{(t^*)}, \boldsymbol{\xi}_i + \boldsymbol{\zeta} \rangle)$$

$$-\frac{1}{m}\sum_{r=1}^{m}\sigma(\langle \mathbf{w}_{y,r}^{(t^*)},\boldsymbol{\xi}_i+\boldsymbol{\zeta}\rangle)\geq \frac{1}{m}\sum_{r=1}^{m}\gamma_{y,r}^{(t^*)}+1/C')$$

Next, we show that for any $y=\pm 1$, there exists some sufficiently large constant $C_2$ that for any $i\in\mathcal{S}_f\cap\mathcal{S}_y$, $\langle \mathbf{w}_{-y,r}^{(t^*)},\boldsymbol{\xi}_i+\boldsymbol{\zeta}\rangle\geq \overline{\rho}_{-y,r,i}^{(t^*)}-1/C_2, \langle \mathbf{w}_{y,r}^{(t^*)},\boldsymbol{\xi}_i+\boldsymbol{\zeta}\rangle\leq 1/C_2$. Then, based on the scale that $\frac{1}{m}\sum_{r=1}^{m}(\overline{\rho}_{\tilde{y}_i,r,i}^{(t^*)}-\gamma_{-\tilde{y}_i,r}^{(t^*)})\geq C_\epsilon$, we can show $\mathbb{P}(yf(\mathbf{W}^{(t^*)},\mathbf{x})<0)\geq 0.5\tau_+$ if $y=1$ and $\mathbb{P}(yf(\mathbf{W}^{(t^*)},\mathbf{x})<0)\geq 0.5\tau_-$ if $y=-1$.

### 5.2. Feature Learning Process without Label Noise

The analysis of feature learning without label noise follows a similar two-stage framework as the analysis with label noise. Our analysis without label noise relies on Lemma 5.3, where we show that under Condition 4.1, the loss derivatives are balanced across all samples, and both the noise coefficients and signal coefficients are of the same order.

**Lemma 5.3.** *Under Condition 4.1, for any $0\leq t\leq T^*$, there exists constants $\kappa,\tilde{C}_\ell>0$ such that*

1. *$\frac{1}{m}\sum_{r=1}^{m}(\overline{\rho}_{y_i,r,i}^{(s)}+\gamma_{y_i,r}^{(s)}-\overline{\rho}_{y_k,r,k}^{(s)}-\gamma_{y_k,r}^{(s)})\leq \kappa$ for all $i,k\in[n]$.*

2. *$\ell_i'^{(s)}/\ell_k'^{(s)}\leq \tilde{C}_\ell$ for all $i,k\in[n]$.*

Lemma 5.3 allows us to establish Proposition 5.2 for all training iterations. Initially, we can demonstrate that both signal and noise coefficients reach a constant order during Stage I, as in Section 5.1. In Stage II, we can show that the scales of the signal and noise coefficients remain on the same order, i.e., $\gamma_{j,r}^{(t)}/\sum_{i=1}^{n}\overline{\rho}_{j,r,i}^{(t)}=\Theta(\mathrm{SNR}^2)$. Finally, by carefully analyzing the test error, we can upper bound the test error in terms of $n\cdot\mathrm{SNR}^2$ (see details in Appendix D.2).

## 6. Experiments

In this section, we provide empirical evidence under both synthetic and real-world setups to support our theory.

### 6.1. Synthetic Experiments

First, we conduct experiments under the synthetic setup.

**Synthetic Setup with Label Noise.** We follow precisely the problem setup introduced in Section 3:

- **SNR.** The fixed signal vector is set to be $\boldsymbol{\mu}=[y\mu,0,\cdots,0]\in\mathbb{R}^d$, where $\mu=20$ and $d=2000$, and the random noise vector $\boldsymbol{\xi}$ is sampled from $\mathcal{N}(0,\mathbf{I}_d)$. This setting corresponds to $n\cdot\mathrm{SNR}^2=20$.

- **Label noise.** The $n=100$ training samples is generated with balanced class labels and each sample's label is flipped with a probability of 0.1.

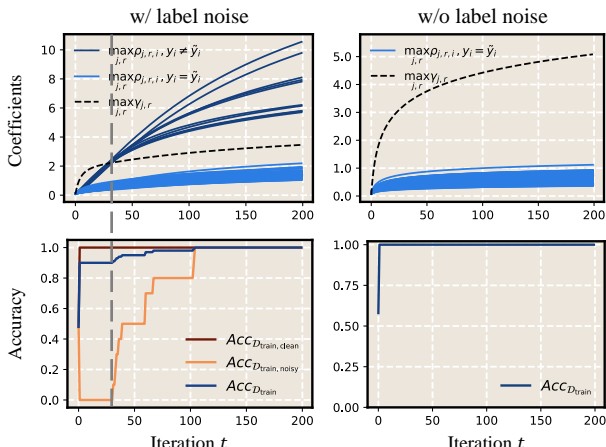

Figure 1: **Experimental validation under the synthetic setup, with label noise (left) and without label noise (right). (Top)** The change in $\max_{j,r}\gamma_{j,r}$ (signal learning) and $\max_{j,r}\rho_{j,r,i}$ (noise memorization) on noisy (i.e., when $y_i\neq\tilde{y}_i$) and clean samples (i.e., when $y_i=\tilde{y}_i$) w.r.t the training iteration $t$. **(Bottom)** The change in overall training accuracy $Acc_{\mathcal{D}_{\mathrm{train}}}$, as well as the accuracy on clean $Acc_{\mathcal{D}_{\mathrm{train,\,clean}}}$ and noisy samples $Acc_{\mathcal{D}_{\mathrm{train,\,noisy}}}$, w.r.t the training iteration $t$ for models under different settings. Note that there are no noisy samples when training without label noise; thus we only plot noise memorization on clean samples and the overall training accuracy. The gray dashed line separates the two stages for training with label noise. More experimental results are in Appendix F.

- **Model and training settings.** A two-layer CNN (as defined in Section 3) is trained using gradient descent, with a total of $T=200$ iterations and a learning rate of $\eta=0.1$.

**The Two-Stage Picture Emerges in the Feature Learning Process with Label Noise.** Specifically, we demonstrate the signal learning process in the two-layer CNN by showing how $\max_{j,r}\gamma_{j,r}$ changes during training. We also present the noise memorization process by illustrating the evolution of $\max_{j,r}\rho_{j,r,i}$. In Figure 1 (left), a clear two-stage pattern is observed in the learning process:

- **Stage I.** The values of $\max_{j,r}\gamma_{j,r}$ are significantly larger than those of $\max_{j,r}\rho_{j,r,i}$, indicating that the signal learning initially dominates;

- **Stage II.** The values of $\max_{j,r}\rho_{j,r,i}$ on noisy samples (i.e., when $y_i\neq\tilde{y}_i$) increasingly surpass those of $\max_{j,r}\gamma_{j,r}$, implying that the noise memorization process, particularly for noisy samples, gradually takes over.

Additionally, we provide the training accuracy curves. In Figure 1 (bottom, left), the accuracy on noisy samples ini-

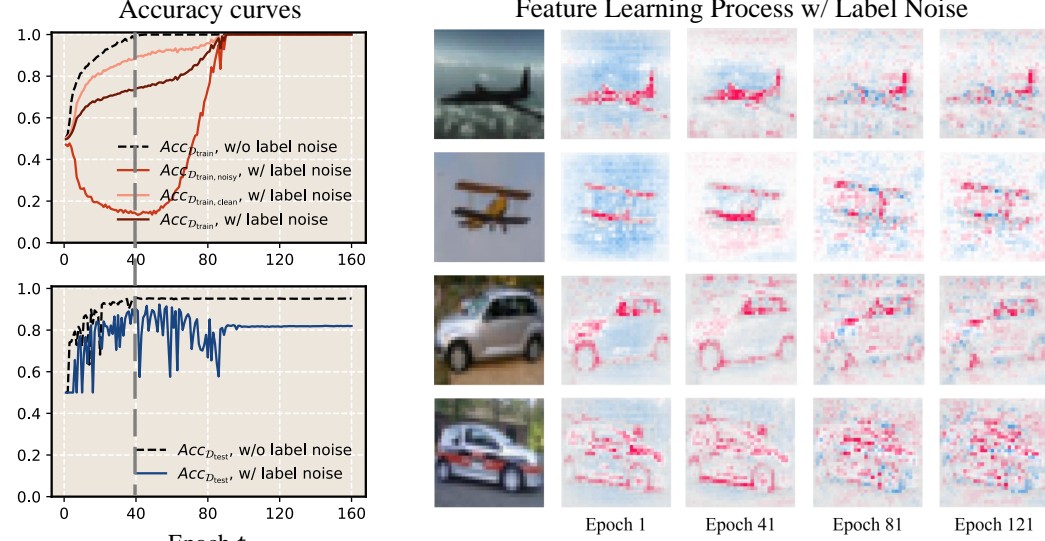

Figure 2: **Experimental validation in real-world scenarios.** Two VGG nets are trained on the first two categories of CIFAR-10 under nearly identical settings. One is trained with label noise and the other without. **(Left)** The accuracy curves for the two models. Here, $Acc_{\mathcal{D}_{\text{train}}}$ and $Acc_{\mathcal{D}_{\text{test}}}$ represent the accuracy on the entire training and test sets, respectively, while $Acc_{\mathcal{D}_{\text{train, clean}}}$ and $Acc_{\mathcal{D}_{\text{train, noisy}}}$ specifically denote the accuracy on clean and noisy samples from the training set. **(Right)** Visualization of model predictions (via SHAP (Lundberg & Lee, 2017)) for noisy samples across multiple epochs. Red regions indicate positive contributions to model predictions, while blue regions denote negative contributions, with darker regions signifying greater contributions. More experimental results are in Appendix F.

tially drops to 0 during the early stage and then gradually increases, as predicted by our theory in Section 4.1

**Synthetic Setup without Label Noise.** For comparison, we also train a baseline model under nearly identical settings but without label flipping.

**Signal Learning Dominates in the Feature Learning Process without Label Noise.** Similarly, we focus on the evolution of signal and noise coefficients. In Figure 1 (top, right), the values of $\max_{j,r} \gamma_{j,r}$ are larger than those of $\max_{j,r} \rho_{j,r,i}$ throughout the training, suggesting that the signal learning dominates the feature learning process. Furthermore, in Figure 1 (bottom, right), the training accuracy remains consistently high along the training. These results closely align with our theory in Section 4.3.

### 6.2. Real-World Experiments

Taking a step further, we also validate our theoretical analysis in the real-world scenario. The code for replicating the results is available on `https://github.com/zzp1012/label-noise-theory`.

**Real-World Setup with and without Label Noise.** We perform experiments on the commonly used image classification dataset CIFAR-10 (Krizhevsky et al., 2009), using the standard network architecture VGG net (Simonyan & Zisserman, 2015). Specifically, we train the VGG net with

stochastic gradient descent (SGD) on samples from the first two categories of CIFAR-10, where each sample's label is flipped with a probability of 0.2. Similar to the synthetic experiment, for comparison, we also train another VGG net under the same settings but without label flipping.

**The Two Stage Picture: Accuracy.** Accuracy curves demonstrate the two-stage picture with label noise in real-world scenarios. In Figure 2 (left), when training with label noise, the accuracy on noisy samples follows a similar two-stage pattern to the synthetic experiments — an initial drop followed by a gradual increase to 1 — while the test set accuracy remains consistently lower than when training without label noise. In comparison, when training without label noise, the accuracies on both training and test sets consistently increase during the training.

**The Two Stage Picture: Visualization of the Feature Learning Process.** As deep models are black-boxes, we visualize their feature learning process using post-hoc interpretability methods. Specifically, we choose SHAP (Lundberg & Lee, 2017), which interprets model predictions by attributing the contribution of each input variable (e.g., pixels for image inputs). In Figure 2 (right), it is evident that a two-stage behavior emerges. In the first stage (reflected by Epoch 1 and 41), clear patterns are observed in the interpretations, such as the wings of "airplane" class and contours of "automobile" class, implying the model relies on the general-

izable features for predictions. However, in the second stage (reflected by Epoch 81 and 121), the interpretations appear messy, and the model overfits to the spurious features, such as the noise in backgrounds.

## 7. Conclusion and Limitations

In conclusion, our work offers an exact learning dynamics analysis of training neural networks with label noise, We identify two distinct stages in the feature learning process, offering a solid explanation for the effectiveness of techniques such as early stopping and sample selection. Our theoretical results, along with sufficient practical insights, are significant contributions that have been largely absent from the deep learning theory literature. In addition, experiments under both synthetic and real-world setups back up our theory.

**Limitations.** Our current analysis is limited to random label noise and does not account for data-dependent label noise, where mislabeling probability varies based on sample characteristics. Extending our framework to structured or adversarial noise remains an important direction for future research. Additionally, our theoretical results are derived for a two-layer CNN, which, while analytically tractable, may not fully capture the complexities of deeper architectures. Investigating whether our two-stage learning dynamics persist in deeper networks with advanced components (e.g., residual connections, normalization) is crucial for improving robustness.

## Acknowledgments

WH was supported by JSPS KAKENHI Grant Number 24K20848. TS was partially supported by JSPS KAKENHI (24K02905) and JST CREST (JPMJCR2115). This research is supported by the National Research Foundation, Singapore and the Ministry of Digital Development and Information under the AI Visiting Professorship Programme (award number AIVP-2024-004). Any opinions, findings and conclusions or recommendations expressed in this material are those of the author(s) and do not reflect the views of National Research Foundation, Singapore and the Ministry of Digital Development and Information.

## Impact Statement

This paper presents work whose goal is to advance the field of Machine Learning. There are many potential societal consequences of our work, none which we feel must be specifically highlighted here.

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

## A. Comparison of Technical Quantities to (Kou et al., 2023)

Among the various differences in conditions compared to (Kou et al., 2023), the most critical distinction lies in the scale of the SNR. Because we aim to characterize the two-stage behaviors induced by label noise, we require the SNR to satisfy $n \cdot \mathrm{SNR}^2 = \Theta(1)$. This enables the signal learning to dominate the noise learning in the first stage while noise learning dominates signal learning in the second stage. Such a distinct two-stage dynamics cannot be captured by (Kou et al., 2023) due to $n \cdot \mathrm{SNR}^2 = o(1)$.

More specifically, in the following, we explicitly compares the key differences in the analysis techniques compared to (Kou et al., 2023):

- **Non-Time-invariant coefficients**: One of the key techniques (Key Technique 1 in (Kou et al., 2023)) is the derivation of time-invariant order of the coefficient ratio: $\gamma_{j,r}^{(t)} / \sum_{i=1}^{n} \overline{\rho}_{j,r,i}^{(t)} = \Theta(\mathrm{SNR}^2)$, which is critical for their generalization analysis. However, in our case, due to the setting of constant order $n \cdot \mathrm{SNR}^2$, the noisy samples exhibit different behaviors as the clean samples (which is the main goal we wish to show), such time-invariance may not hold for all iterations.

- **Non-balancing of the updates**: Another key technique employed in (Kou et al., 2023) is the automatic balancing of coefficient updates, which requires to show $\ell_i'^{(t)} / \ell_k'^{(t)} \leq C$ for all $i, k \in [n]$. That is, the loss derivatives across all samples are approximately balanced, which is critical for their convergence analysis. Because in our case $n \cdot \mathrm{SNR}^2$, the loss derivatives of noisy samples may be significantly larger than that of clean samples, we cannot guarantee the balance of updates across all samples.

Without the above two results in our case, the convergence and generalization analysis becomes challenging. To address the challenges, we require developing novel techniques via refined analysis on clean and noisy samples, which cannot be addressed in the prior works.

To better comprehend the differences to the analysis of (Kou et al., 2023), we present the following tables that compare the different quantities at each training stage. These differences require non-trivial analysis.

| | | First Stage | Second Stage |
|---|---|---|---|
| Monotonicity of signal | (Kou et al., 2023) | Monotonic increase | |
| | Our work | Monotonic increase | No monotonicity |
| Signal-noise magnitude | (Kou et al., 2023) | Noise dominates | |
| | Our work | Signal dominates | Noise dominates |
| Determining factors of $\tilde{y}_i f(\mathbf{W}^{(t)}, \mathbf{x}_i)$ | (Kou et al., 2023) | $\frac{1}{m} \sum_{r=1}^{m} \overline{\rho}_{\tilde{y}_i, r, i}^{(t)} \pm o(1)$ | |
| | Our work | $\begin{cases} \frac{1}{m} \sum_{r=1}^{m} (\gamma_{\tilde{y}_i, r}^{(t)} + \overline{\rho}_{\tilde{y}_i, r, i}^{(t)}) \pm o(1), & \text{for } i \in \mathcal{S}_t \\ \frac{1}{m} \sum_{r=1}^{m} (\overline{\rho}_{\tilde{y}_i, r, i}^{(t)} - \gamma_{-\tilde{y}_i, r}^{(t)}) \pm o(1), & \text{for } i \in \mathcal{S}_f \end{cases}$ | |
| Prediction | (Kou et al., 2023) | $\tilde{y}_i f(\mathbf{W}^{(t)}, \mathbf{x}_i) \geq 0, \forall i \in [n]$ | |
| | Our work | $\begin{cases} \tilde{y}_i f(\mathbf{W}^{(t)}, \mathbf{x}_i) \geq 0, & i \in \mathcal{S}_t \\ \tilde{y}_i f(\mathbf{W}^{(t)}, \mathbf{x}_i) \leq 0, & i \in \mathcal{S}_f \end{cases}$ | $\tilde{y}_i f(\mathbf{W}^{(t)}, \mathbf{x}_i) \geq 0, \forall i \in [n]$ |
| Test error $L_D^{0-1}(\mathbf{W}^{(T_1)})$ | (Kou et al., 2023) | $\begin{cases} o(1), & \text{if } n\|\boldsymbol{\mu}\|_2^4 > C - 1\sigma_\xi^4 d \\ \Omega(1), & \text{if } n\|\boldsymbol{\mu}\|_2^4 \leq C_3\sigma_\xi^4 d \end{cases}$ | |
| | Our work | $o(1)$ | $\Omega(1)$ |

Table 2: Comparisons of key quantities in the analysis at each stage.

## B. Preliminary Lemmas

This section introduces a few lemmas that are critical to bound the parameters at initialization.

**Lemma B.1** ((Cao et al., 2022; Kou et al., 2023)). *Suppose $d = \Omega(\log(6n/\delta))$. Then with probability at least $1 - \delta$,*

$$\sigma_\xi^2 d/2 \leq \|\boldsymbol{\xi}_i\|_2^2 \leq 3\sigma_\xi^2 d/2,$$

$$|\langle \boldsymbol{\xi}_i, \boldsymbol{\xi}_{i'} \rangle| \le 2\sigma_\xi^2 \sqrt{d \log(6n^2/\delta)},$$
$$|\langle \boldsymbol{\xi}_i, \boldsymbol{\mu} \rangle| \le \|\boldsymbol{\mu}\|_2 \sigma_\xi \sqrt{2 \log(6n/\delta)}.$$

**Lemma B.2** ((Cao et al., 2022; Kou et al., 2023))**.** *Suppose that $d = \Omega(\log(nm/\delta))$, $m = \Omega(\log(1/\delta))$. Then with probability at least $1 - \delta$,*

$$\sigma_0^2 d/2 \le \|\mathbf{w}_{j,r}^0\|_2^2 \le 3\sigma_0^2 d/2$$
$$|\langle \mathbf{w}_{j,r}^{(0)}, \boldsymbol{\mu} \rangle| \le \sqrt{2 \log(12m/\delta)} \cdot \sigma_0 \|\boldsymbol{\mu}\|_2,$$
$$|\langle \mathbf{w}_{j,r}^{(0)}, \boldsymbol{\xi}_i \rangle| \le 2\sqrt{\log(12mn/\delta)} \cdot \sigma_0 \sigma_\xi \sqrt{d}.$$

**Lemma B.3** ((Kou et al., 2023))**.** *Let $\mathcal{S}_i^{(t)} := \{r \in [m] : \langle \mathbf{w}_{\tilde{y}_i,r}^{(t)}, \boldsymbol{\xi}_i \rangle > 0\}$ and $\mathcal{S}_{j,r}^{(t)} := \{i \in [n] : j = \tilde{y}_i, \langle \mathbf{w}_{j,r}^{(t)}, \boldsymbol{\xi}_i \rangle > 0\}$. Then for any $\delta > 0$, and $m \ge 50 \log(4n/\delta)$, $n \ge 32 \log(8m/\delta)$, we have with probability at least $1 - \delta$,*

$$|\mathcal{S}_i^{(0)}| \ge 0.4m, \quad \forall i \in [n]$$
$$|\mathcal{S}_{j,r}^{(0)}| \ge n/8, \quad \forall j = \pm 1, r \in [m].$$

## C. Analysis with label noise

Without loss of generality, for the subsequent analysis, we assume $|\mathcal{S}_1 \cap \mathcal{S}_t| = \frac{(1-\tau_+)n}{2}, |\mathcal{S}_1 \cap \mathcal{S}_f| = \frac{\tau_+ n}{2}, |\mathcal{S}_{-1} \cap \mathcal{S}_t| = \frac{(1-\tau_-)n}{2}, |\mathcal{S}_{-1} \cap \mathcal{S}_f| = \frac{\tau_- n}{2}$.

### C.1. Coefficients Decomposition Iteration

**Lemma C.1.** *The coefficients $\gamma_{j,r}^{(t)}, \overline{\rho}_{j,r,i}^{(t)}, \underline{\rho}_{j,r,i}^{(t)}$ in decomposition (3) satisfy $\gamma_{j,r}^{(0)}, \overline{\rho}_{j,r,i}^{(0)}, \underline{\rho}_{j,r,i}^{(0)} = 0$ and admit the following iterative update rule:*

$$\gamma_{j,r}^{(t+1)} = \gamma_{j,r}^{(t)} - \frac{\eta}{nm} \sum_{i=1}^n \ell_i'^{(t)} \sigma'(\langle \mathbf{w}_{j,r}^{(t)}, y_i \boldsymbol{\mu} \rangle) y_i \tilde{y}_i \|\boldsymbol{\mu}\|_2^2,$$
$$\overline{\rho}_{j,r,i}^{(t+1)} = \overline{\rho}_{j,r,i}^{(t)} - \frac{\eta}{nm} \ell_i'^{(t)} \sigma'(\langle \mathbf{w}_{j,r}^{(t)}, \boldsymbol{\xi}_i \rangle) \|\boldsymbol{\xi}_i\|_2^2 \mathbb{1}(\tilde{y}_i = j),$$
$$\underline{\rho}_{j,r,i}^{(t+1)} = \underline{\rho}_{j,r,i}^{(t)} + \frac{\eta}{nm} \ell_i'^{(t)} \sigma'(\langle \mathbf{w}_{j,r}^{(t)}, \boldsymbol{\xi}_i \rangle) \|\boldsymbol{\xi}_i\|_2^2 \mathbb{1}(\tilde{y}_i = -j).$$

*Proof of Lemma C.1.* By iterating the gradient descent update, we can show

$$\mathbf{w}_{j,r}^{(t)} = \mathbf{w}_{j,r}^{(0)} - \frac{\eta}{nm} \sum_{s=0}^{t-1} \sum_{i=1}^n \ell_i'^{(s)} \sigma'(\langle \mathbf{w}_{j,r}^{(s)}, \boldsymbol{\xi}_i \rangle) j \tilde{y}_i \boldsymbol{\xi}_i - \frac{\eta}{nm} \sum_{s=0}^{t-1} \sum_{i=1}^n \ell_i'^{(s)} \sigma'(\langle \mathbf{w}_{j,r}^{(s)}, y_i \boldsymbol{\mu} \rangle) j y_i \tilde{y}_i \boldsymbol{\mu}$$

Because $\boldsymbol{\xi}_i, \boldsymbol{\mu}$ are linearly independent almost surely for all $i \in [n]$. Then from the definition:

$$\mathbf{w}_{j,r}^{(t)} = \mathbf{w}_{j,r}^{(0)} + j\gamma_{j,r}^{(t)} \|\boldsymbol{\mu}\|_2^{-2} \boldsymbol{\mu} + \sum_{i=1}^n \rho_{j,r,i}^{(t)} \|\boldsymbol{\xi}_i\|_2^{-2} \boldsymbol{\xi}_i$$

there exists a unique decomposition as

$$\gamma_{j,r}^{(t)} = -\frac{\eta}{nm} \sum_{s=0}^{t-1} \sum_{i=1}^n \ell_i'^{(s)} \sigma'(\langle \mathbf{w}_{j,r}^{(s)}, y_i \boldsymbol{\mu} \rangle) y_i \tilde{y}_i \|\boldsymbol{\mu}\|_2^2$$
$$\rho_{j,r,i}^{(t)} = -\frac{\eta}{mn} \sum_{s=0}^{t-1} \ell_i'^{(s)} \sigma'(\langle \mathbf{w}_{j,r}^{(s)}, \boldsymbol{\xi}_i \rangle) j \tilde{y}_i \|\boldsymbol{\xi}_i\|_2^2.$$

By definition of $\overline{\rho}_{j,r,i}^{(t)}, \underline{\rho}_{j,r,i}^{(t)}$, and the fact that $\ell_i' \leq 0$,

$$\overline{\rho}_{j,r,i}^{(t)} = -\frac{\eta}{nm} \sum_{s=0}^{t-1} \ell_i'^{(s)} \sigma'(\langle \mathbf{w}_{j,r}^{(s)}, \boldsymbol{\xi}_i \rangle) \|\boldsymbol{\xi}_i\|_2^2 \mathbb{1}(\tilde{y}_i = j)$$

$$\underline{\rho}_{j,r,i}^{(t)} = \frac{\eta}{nm} \sum_{s=0}^{t-1} \ell_i'^{(s)} \sigma'(\langle \mathbf{w}_{j,r}^{(s)}, \boldsymbol{\xi}_i \rangle) \|\boldsymbol{\xi}_i\|_2^2 \mathbb{1}(\tilde{y}_i = -j)$$

Then the iterative updates of the coefficients follow directly. □

## C.2. Scale of Coefficients

Here we start to provide a global bound for the decomposition coefficients. We show for a sufficiently large number of iterations $T^* = \Theta(\eta^{-1}\epsilon^{-1}nmd^{-1}\sigma_\xi^{-2})$, the scale of the coefficients can be upper bounded up to some logarithmic factors.

We consider the following definition:

$$\beta = 2\max_{i,j,r}\{|\langle \mathbf{w}_{j,r}^{(0)}, \boldsymbol{\mu} \rangle|, |\langle \mathbf{w}_{j,r}^{(0)}, \boldsymbol{\xi}_i \rangle|\}, \qquad \text{SNR} = \frac{\|\boldsymbol{\mu}\|}{\sigma_\xi\sqrt{d}}, \qquad \alpha = C_t\log(T^*)$$

for some constant $C_t > 0$ to be determined later. Then by Lemma B.2, we can bound as $\beta \leq \sigma_0 \max\left\{\sqrt{2\log(12m/\delta)}\|\boldsymbol{\mu}\|_2, 2\sqrt{\log(12mn/\delta)}\sigma_\xi\sqrt{d}\right\}$.

We next provide the main proposition that bounds the scale of coefficients.

**Proposition C.2** (Restatement of Proposition 5.2). *Under Condition 4.1, for any $0 \leq t \leq T^*$*

$$0 \leq \overline{\rho}_{j,r,i}^{(t)} \leq \alpha, \tag{4}$$

$$0 \geq \underline{\rho}_{j,r,i}^{(t)} \geq -\beta - 10\sqrt{\frac{\log(6n^2/\delta)}{d}}n\alpha \geq -\alpha, \tag{5}$$

$$0 \leq \gamma_{j,r}^{(t)} \leq C_\gamma\alpha \tag{6}$$

*for some constant $C_\gamma > 0$.*

We aim to prove Proposition C.2 using induction. This requires several intermediate lemmas through the induction process.

**Lemma C.3.** *Under Condition 4.1, suppose* (4), (5), (6) *hold at iteration t. Then for all $r \in [m]$, $j \in \{\pm 1\}, i \in [n]$,*

$$|\langle \mathbf{w}_{j,r}^{(t)} - \mathbf{w}_{j,r}^{(0)}, \boldsymbol{\mu} \rangle - j \cdot \gamma_{j,r}^{(t)}| \leq \text{SNR}\sqrt{\frac{8\log(6n/\delta)}{d}}n\alpha,$$

$$|\langle \mathbf{w}_{j,r}^{(t)} - \mathbf{w}_{j,r}^{(0)}, \boldsymbol{\xi}_i \rangle - \overline{\rho}_{j,r,i}^{(t)}| \leq 5\sqrt{\frac{\log(6n^2/\delta)}{d}}n\alpha, \quad \tilde{y}_i = j$$

$$|\langle \mathbf{w}_{j,r}^{(t)} - \mathbf{w}_{j,r}^{(0)}, \boldsymbol{\xi}_i \rangle - \underline{\rho}_{j,r,i}^{(t)}| \leq 5\sqrt{\frac{\log(6n^2/\delta)}{d}}n\alpha, \quad \tilde{y}_i = -j$$

*Proof of Lemma C.3.* From signal-noise decomposition (3),

$$|\langle \mathbf{w}_{j,r}^{(t)} - \mathbf{w}_{j,r}^{(0)}, \boldsymbol{\mu} \rangle - j \cdot \gamma_{j,r}^{(t)}| = \left| \sum_{i=1}^{n} \overline{\rho}_{j,r,i}^{(t)} \cdot \|\boldsymbol{\xi}_i\|_2^{-2} \cdot \langle \boldsymbol{\xi}_i, \boldsymbol{\mu} \rangle + \sum_{i=1}^{n} \underline{\rho}_{j,r,i}^{(t)} \cdot \|\boldsymbol{\xi}_i\|_2^{-2} \cdot \langle \boldsymbol{\xi}_i, \boldsymbol{\mu} \rangle \right|$$

$$\leq \sum_{i=1}^{n} (|\overline{\rho}_{j,r,i}^{(t)}| + |\underline{\rho}_{j,r,i}^{(t)}|) \|\boldsymbol{\xi}_i\|_2^{-2} \cdot |\langle \boldsymbol{\xi}_i, \boldsymbol{\mu} \rangle|$$

$$\leq \text{SNR}\sqrt{\frac{8\log(6n/\delta)}{d}} \sum_{i=1}^{n} (|\overline{\rho}_{j,r,i}^{(t)}| + |\underline{\rho}_{j,r,i}^{(t)}|)$$

$$\leq \text{SNR}\sqrt{\frac{8\log(6n/\delta)}{d}}n\alpha$$

where the second inequality is due to Lemma B.1 and the last inequality is by (4), (5). The second inequality follows similarly.

Then, for $\tilde{y}_i = j$, we have $\underline{\rho}_{j,r,i}^{(t)} = 0, \forall t \geq 0$ and hence

$$
\begin{aligned}
&|\langle \mathbf{w}_{j,r}^{(t)} - \mathbf{w}_{j,r}^{(0)}, \boldsymbol{\xi}_i \rangle - \overline{\rho}_{j,r,i}^{(t)}| \\
&= \left| j \cdot \gamma_{j,r}^{(t)} \cdot \|\boldsymbol{\mu}\|_2^{-2} \langle \boldsymbol{\mu}, \boldsymbol{\xi}_i \rangle + \sum_{i' \neq i} \overline{\rho}_{j,r,i'}^{(t)} \cdot \|\boldsymbol{\xi}_{i'}\|_2^{-2} \cdot \langle \boldsymbol{\xi}_i, \boldsymbol{\xi}_{i'} \rangle + \sum_{i' \neq i} \underline{\rho}_{j,r,i'}^{(t)} \cdot \|\boldsymbol{\xi}_{i'}\|_2^{-2} \cdot \langle \boldsymbol{\xi}_i, \boldsymbol{\xi}_{i'} \rangle \right| \\
&\leq \|\boldsymbol{\mu}\|_2^{-2} \cdot |\langle \boldsymbol{\mu}, \boldsymbol{\xi}_i \rangle| \cdot |\gamma_{j,r}^{(t)}| + \sum_{i' \neq i}^{n} \left( |\overline{\rho}_{j,r,i'}^{(t)}| + |\underline{\rho}_{j,r,i'}^{(t)}| \right) \|\boldsymbol{\xi}_{i'}\|_2^{-2} \cdot |\langle \boldsymbol{\xi}_{i'}, \boldsymbol{\xi}_i \rangle| \\
&\leq \mathrm{SNR}\sqrt{\frac{2\log(6n/\delta)}{d}} C_\gamma n\alpha + 4\sqrt{\frac{\log(6n^2/\delta)}{d}} n\alpha \\
&\leq (2C_\gamma \mathrm{SNR} + 4)\sqrt{\frac{\log(6n^2/\delta)}{d}} n\alpha \\
&\leq 5\sqrt{\frac{\log(6n^2/\delta)}{d}} n\alpha
\end{aligned}
$$

where we use Lemma B.1 and (6) in the second inequality. In the third inequality, we use $2\log(6n/\delta) \leq 4\log(6n^2/\delta)$. In the fourth inequality, we note that the condition on SNR ensures that $\mathrm{SNR} = \Theta(1/\sqrt{n})$.

For $\tilde{y}_i \neq j$, the proof follow exactly the same strategy as for $\tilde{y}_i = j$ and hence is omitted. $\qquad\square$

**Lemma C.4.** *Under Condition 4.1 and suppose* (4), (5), (6) *hold at time $t$, then there exists a sufficiently large constant $C_1 > 0$ such that*

$$
\frac{1}{m}\sum_{r=1}^{m} \left( \gamma_{\tilde{y}_i,r}^{(t)} + \overline{\rho}_{\tilde{y}_i,r,i}^{(t)} \right) - 1/C_1 \leq \tilde{y}_i f(\mathbf{W}^{(t)}, \mathbf{x}_i) \leq \frac{1}{m}\sum_{r=1}^{m} \left( \gamma_{\tilde{y}_i,r}^{(t)} + \overline{\rho}_{\tilde{y}_i,r,i}^{(t)} \right) + 1/C_1 \quad \text{when } i \in \mathcal{S}_t
$$

$$
\frac{1}{m}\sum_{r=1}^{m} \left( \overline{\rho}_{\tilde{y}_i,r,i}^{(t)} - \gamma_{-\tilde{y}_i,r}^{(t)} \right) - 1/C_1 \leq \tilde{y}_i f(\mathbf{W}^{(t)}, \mathbf{x}_i) \leq \frac{1}{m}\sum_{r=1}^{m} \left( \overline{\rho}_{\tilde{y}_i,r,i}^{(t)} - \gamma_{-\tilde{y}_i,r}^{(t)} \right) + 1/C_1 \quad \text{when } i \in \mathcal{S}_f
$$

*Proof of Lemma C.4.* We first see

$$
\begin{aligned}
\tilde{y}_i f(\mathbf{W}^{(t)}, \mathbf{x}_i) &= \frac{1}{m}\sum_{j,r} \tilde{y}_i \cdot j \cdot \left( \sigma(\langle \mathbf{w}_{j,r}^{(t)}, y_i\boldsymbol{\mu} \rangle) + \sigma(\langle \mathbf{w}_{j,r}^{(t)}, \boldsymbol{\xi}_i \rangle) \right) \\
&= \frac{1}{m}\sum_{r=1}^{m} \left( \sigma(\langle \mathbf{w}_{\tilde{y}_i,r}^{(t)}, y_i\boldsymbol{\mu} \rangle) + \sigma(\langle \mathbf{w}_{\tilde{y}_i,r}^{(t)}, \boldsymbol{\xi}_i \rangle) \right) - \frac{1}{m}\sum_{r=1}^{m} \left( \sigma(\langle \mathbf{w}_{-\tilde{y}_i,r}^{(t)}, y_i\boldsymbol{\mu} \rangle) + \sigma(\langle \mathbf{w}_{-\tilde{y}_i,r}^{(t)}, \boldsymbol{\xi}_i \rangle) \right).
\end{aligned}
$$

Recall from the gradient descent update and Lemma C.3,

$$
|\langle \mathbf{w}_{j,r}^{(t)}, \boldsymbol{\mu} \rangle - \langle \mathbf{w}_{j,r}^{(0)}, \boldsymbol{\mu} \rangle - j \cdot \gamma_{j,r}^{(t)}| = \mathrm{SNR}\sqrt{\frac{8\log(6n/\delta)}{d}} n\alpha
$$

Then it can be verified that when $\tilde{y}_i = y_i$,

$$
\langle \mathbf{w}_{\tilde{y}_i,r}^{(t)}, y_i\boldsymbol{\mu} \rangle \leq |\langle \mathbf{w}_{\tilde{y}_i,r}^{(0)}, \boldsymbol{\mu} \rangle| + \gamma_{\tilde{y}_i,r}^{(t)} + \mathrm{SNR}\sqrt{\frac{8\log(6n/\delta)}{d}} n\alpha
$$

$$
\langle \mathbf{w}_{\tilde{y}_i,r}^{(t)}, -y_i\boldsymbol{\mu} \rangle \leq |\langle \mathbf{w}_{\tilde{y}_i,r}^{(0)}, \boldsymbol{\mu} \rangle| - \gamma_{\tilde{y}_i,r}^{(t)} + \mathrm{SNR}\sqrt{\frac{8\log(6n/\delta)}{d}} n\alpha
$$

$$
\leq |\langle \mathbf{w}_{\tilde{y}_i,r}^{(0)}, \boldsymbol{\mu} \rangle| + \mathrm{SNR}\sqrt{\frac{8\log(6n/\delta)}{d}} n\alpha
$$

$$\langle \mathbf{w}_{\tilde{y}_i,r}^{(t)}, \boldsymbol{\xi}_i \rangle \leq |\langle \mathbf{w}_{\tilde{y}_i,r}^{(0)}, \boldsymbol{\xi}_i \rangle| + \overline{\rho}_{\tilde{y}_i,r,i}^{(t)} + 5\sqrt{\frac{\log(6n^2/\delta)}{d}} n\alpha$$

$$\langle \mathbf{w}_{-\tilde{y}_i,r}^{(t)}, -y_i\boldsymbol{\mu} \rangle \geq \gamma_{-\tilde{y}_i,r}^{(t)} - |\mathbf{w}_{-\tilde{y}_i,r}^{(0)}, \boldsymbol{\mu}| - \mathrm{SNR}\sqrt{\frac{8\log(6n/\delta)}{d}} n\alpha$$

Using these inequalities, we can upper bound when $\tilde{y}_i = y_i$, i.e., $i \in \mathcal{S}_t$,

$$\tilde{y}_i f(\mathbf{W}^{(t)}, \mathbf{x}_i) \leq \frac{1}{m} \sum_{r=1}^{m} \left( \sigma(\langle \mathbf{w}_{\tilde{y}_i,r}^{(t)}, y_i\boldsymbol{\mu} \rangle) + \sigma(\langle \mathbf{w}_{\tilde{y}_i,r}^{(t)}, \boldsymbol{\xi}_i \rangle) \right)$$

$$\leq \frac{1}{m} \sum_{r=1}^{m} \left( \gamma_{\tilde{y}_i,r}^{(t)} + \overline{\rho}_{\tilde{y}_i,r,i}^{(t)} \right) + 2\beta + \widetilde{O}(n\alpha/\sqrt{d})$$

$$\leq \frac{1}{m} \sum_{r=1}^{m} \left( \gamma_{\tilde{y}_i,r}^{(t)} + \overline{\rho}_{\tilde{y}_i,r,i}^{(t)} \right) + 1/C_1$$

where we use Lemma C.3 and the Condition 4.1 where we choose a sufficiently large $C_1$.

Similarly, we can lower bound

$$\tilde{y}_i f(\mathbf{W}^{(t)}, \mathbf{x}_i) \geq \frac{1}{m} \sum_{r=1}^{m} \left( \gamma_{\tilde{y}_i,r}^{(t)} + \overline{\rho}_{\tilde{y}_i,r,i}^{(t)} \right) - 1/C_1$$

On the other hand, when $\tilde{y}_i \neq y_i$, it can be shown that

$$\langle \mathbf{w}_{\tilde{y}_i,r}^{(t)}, y_i\boldsymbol{\mu} \rangle \leq |\langle \mathbf{w}_{\tilde{y}_i,r}^{(0)}, \boldsymbol{\mu} \rangle| - \gamma_{\tilde{y}_i,r}^{(t)} + \mathrm{SNR}\sqrt{\frac{8\log(6n/\delta)}{d}} n\alpha \leq |\langle \mathbf{w}_{\tilde{y}_i,r}^{(0)}, \boldsymbol{\mu} \rangle| + \mathrm{SNR}\sqrt{\frac{8\log(6n/\delta)}{d}} n\alpha$$

$$\langle \mathbf{w}_{-\tilde{y}_i,r}^{(t)}, y_i\boldsymbol{\mu} \rangle \leq |\langle \mathbf{w}_{-\tilde{y}_i,r}^{(0)}, \boldsymbol{\mu} \rangle| + \gamma_{-\tilde{y}_i,r}^{(t)} + \mathrm{SNR}\sqrt{\frac{8\log(6n/\delta)}{d}} n\alpha$$

$$\langle \mathbf{w}_{-\tilde{y}_i,r}^{(t)}, y_i\boldsymbol{\mu} \rangle \geq \gamma_{-\tilde{y}_i,r}^{(t)} - |\langle \mathbf{w}_{-\tilde{y}_i,r}^{(0)}, \boldsymbol{\mu} \rangle| - \mathrm{SNR}\sqrt{\frac{8\log(6n/\delta)}{d}} n\alpha$$

$$\langle \mathbf{w}_{\tilde{y}_i,r}^{(t)}, \boldsymbol{\xi}_i \rangle \leq \overline{\rho}_{\tilde{y}_i,r,i}^{(t)} + |\langle \mathbf{w}_{\tilde{y}_i,r}^{(0)}, \boldsymbol{\xi}_i \rangle| + 5\sqrt{\frac{\log(6n^2/\delta)}{d}} n\alpha$$

where we notice $\gamma_{j,r}^{(t)} \geq 0$.

Then we can upper bound when $\tilde{y}_i \neq y_i$ as

$$\tilde{y}_i f(\mathbf{W}^{(t)}, \mathbf{x}_i) \leq \frac{1}{m} \sum_{r=1}^{m} \left( \sigma(\langle \mathbf{w}_{\tilde{y}_i,r}^{(t)}, y_i\boldsymbol{\mu} \rangle) + \sigma(\langle \mathbf{w}_{\tilde{y}_i,r}^{(t)}, \boldsymbol{\xi}_i \rangle) \right) - \frac{1}{m} \sum_{r=1}^{m} \sigma(\langle \mathbf{w}_{-\tilde{y}_i,r}^{(t)}, y_i\boldsymbol{\mu} \rangle)$$

$$\leq \beta + \mathrm{SNR}\sqrt{\frac{8\log(6n/\delta)}{d}} C_\rho n + \frac{1}{m} \sum_{r=1}^{m} \overline{\rho}_{\tilde{y}_i,r,i}^{(t)} + 5\sqrt{\frac{\log(6n^2/\delta)}{d}} C_\rho n$$

$$- \frac{1}{m} \sum_{r=1}^{m} \gamma_{-\tilde{y}_i,r}^{(t)} + \frac{1}{m} \sum_{r=1}^{m} |\langle \mathbf{w}_{-\tilde{y}_i,r}^{(0)}, y_i\boldsymbol{\mu} \rangle| + \mathrm{SNR}\sqrt{\frac{8\log(6n/\delta)}{d}} C_\rho n$$

$$\leq \frac{1}{m} \sum_{r=1}^{m} \left( \overline{\rho}_{\tilde{y}_i,r,i}^{(t)} - \gamma_{-\tilde{y}_i,r}^{(t)} \right) + 1/C_1$$

where the second inequality uses Lemma C.3 and last inequality is by the Condition 4.1.

Similarly, we can lower bound $\tilde{y}_i f(\mathbf{W}^{(t)}, \mathbf{x}_i)$ as

$$\tilde{y}_i f(\mathbf{W}^{(t)}, \mathbf{x}_i) \geq \frac{1}{m} \sum_{r=1}^{m} \sigma(\langle \mathbf{w}_{\tilde{y}_i,r}^{(t)}, \boldsymbol{\xi}_i \rangle) - \frac{1}{m} \sum_{r=1}^{m} \left( \sigma(\langle \mathbf{w}_{-\tilde{y}_i,r}^{(t)}, y_i\boldsymbol{\mu} \rangle) + \sigma(\langle \mathbf{w}_{-\tilde{y}_i,r}^{(t)}, \boldsymbol{\xi}_i \rangle) \right)$$

$$\geq \frac{1}{m} \sum_{r=1}^{m} \left( \overline{\rho}_{\tilde{y}_i,r,i}^{(t)} - \gamma_{\tilde{y}_i,r}^{(t)} \right) - 1/C_1$$

where we use Lemma C.3. $\qquad\qquad\qquad\qquad\qquad\qquad\qquad\qquad\qquad\qquad\qquad\qquad\qquad\qquad\square$

**Lemma C.5.** *Under Condition 4.1 and suppose* (4), (5), (6) *hold at time t. If* $\max_{j,r,i}\{\gamma_{j,r}^{(t)}, \overline{\rho}_{j,r,i}^{(t)}\} = O(1)$*, we have* $\tilde{y}_i f(\mathbf{W}^{(t)}, \mathbf{x}_i) = O(1)$ *and* $\ell_i'^{(t)} = \Omega(1)$ *for all* $i \in [n]$.

*Proof of Lemma C.5.* The proof trivially from Lemma C.4 and the definition of loss. Specifically, we denote the upper bound as $C''$. For $i \in \mathcal{S}_t$, by Lemma C.4,

$$|\ell_i'^{(t)}| = \frac{1}{1 + \exp(\tilde{y}_i f(\mathbf{W}^{(t)}, \mathbf{x}_i))} \geq \frac{1}{1 + \exp\left(\frac{1}{m}\sum_{r=1}^{m}\left(\gamma_{\tilde{y}_i,r}^{(t)} + \overline{\rho}_{\tilde{y}_i,r,i}^{(t)}\right) + 1/C_1\right)}$$

$$\geq \frac{1}{1 + \exp(2C'' + 1/C_1)}$$

For $i \in \mathcal{S}_f$, by Lemma C.4,

$$|\ell_i'^{(t)}| = \frac{1}{1 + \exp(\tilde{y}_i f(\mathbf{W}^{(t)}, \mathbf{x}_i))} \geq \frac{1}{1 + \exp\left(\frac{1}{m}\sum_{r=1}^{m}\left(-\gamma_{-\tilde{y}_i,r}^{(t)} + \overline{\rho}_{\tilde{y}_i,r,i}^{(t)}\right) + 1/C_1\right)}$$

$$\geq \frac{1}{1 + \exp(C'' + 1/C_1)} > \frac{1}{1 + \exp(2C'' + 1/C_1)}$$

where the second inequality is by $\gamma_{-\tilde{y}_i,r}^{(t)} \geq 0$. Thus for all $i \in [n]$, we can show that $|\ell_i'^{(t)}| \geq (1 + \exp(2C'' + C_1^{-1}))^{-1}$. $\quad\square$

Recall $\mathcal{S}_i^{(s)} := \{r \in [m] : \langle \mathbf{w}_{\tilde{y}_i,r}^{(s)}, \boldsymbol{\xi}_i \rangle > 0\}$ and $\mathcal{S}_{j,r}^{(s)} := \{i \in [n] : y_i = j, \langle \mathbf{w}_{j,r}^{(s)}, \boldsymbol{\xi}_i \rangle > 0\}$.

The next lemma shows that in the first stage where the loss derivatives can be lower bounded, the inner product between weights and noise is increasing.

**Lemma C.6.** *Under Condition 4.1 and suppose for any* $t \leq T^*$, (4), (5), (6) *hold for all* $s \leq t$. *Then we can show*

$$\mathcal{S}_i^{(0)} \subseteq \mathcal{S}_i^{(s)}, \quad \mathcal{S}_{j,r}^{(0)} \subseteq \mathcal{S}_{j,r}^{(s)}.$$

*for any* $s \leq t$.

*Proof of Lemma C.6.* The proof is by induction where we separately consider two stages. First at $t = 0$, it is trivial to verify that both claims hold. In the first stage where $\max_{j,r,i}\{\gamma_{j,r}^{(t)}, \overline{\rho}_{j,r,i}^{(t)}\} = O(1)$, we can lower bound the loss derivatives by a constant according to Lemma C.5, i.e., $|\ell_i'^{(t)}| \geq C_\ell$ for all $i \in [n]$. Let $T_1$ be the termination time of the first stage. Suppose there exists a time $\tilde{t} \leq T_1$ such that the claims hold for all $s \leq \tilde{t} - 1$, we now prove it also holds at $\tilde{t}$.

By the gradient descent update, for any $r \in \mathcal{S}_i^{(0)}$, we have $r \in \mathcal{S}_i^{(\tilde{t}-1)}$ and thus

$$\langle \mathbf{w}_{\tilde{y}_i,r}^{(\tilde{t})}, \boldsymbol{\xi}_i \rangle = \langle \mathbf{w}_{\tilde{y}_i,r}^{(\tilde{t}-1)}, \boldsymbol{\xi}_i \rangle - \frac{\eta}{nm}\sum_{i'=1}^{n} \ell_{i'}'^{(\tilde{t}-1)} \cdot \sigma'(\langle \mathbf{w}_{\tilde{y}_i,r}^{(\tilde{t}-1)}, \boldsymbol{\xi}_{i'} \rangle) \cdot \langle \boldsymbol{\xi}_i, \boldsymbol{\xi}_{i'} \rangle$$

$$- \frac{\eta}{nm}\sum_{i'=1}^{n} \ell_{i'}'^{(\tilde{t}-1)} \cdot \sigma'(\langle \mathbf{w}_{\tilde{y}_i,r}^{(\tilde{t}-1)}, y_{i'}\boldsymbol{\mu} \rangle) \cdot \langle y_{i'}\boldsymbol{\mu}, \boldsymbol{\xi}_i \rangle$$

$$= \langle \mathbf{w}_{\tilde{y}_i,r}^{(\tilde{t}-1)}, \boldsymbol{\xi}_i \rangle \underbrace{- \frac{\eta}{nm}\ell_i'^{(\tilde{t}-1)}\|\boldsymbol{\xi}_i\|_2^2}_{A_1} \underbrace{- \frac{\eta}{nm}\sum_{i'\neq i} \ell_{i'}'^{(\tilde{t}-1)}\sigma'(\langle \mathbf{w}_{\tilde{y}_i,r}^{(\tilde{t}-1)}, \boldsymbol{\xi}_{i'} \rangle) \cdot \langle \boldsymbol{\xi}_i, \boldsymbol{\xi}_{i'} \rangle}_{A_2}$$

$$\underbrace{- \frac{\eta}{nm}\sum_{i'=1}^{n} \ell_{i'}'^{(\tilde{t}-1)} \cdot \sigma'(\langle \mathbf{w}_{\tilde{y}_i,r}^{(\tilde{t}-1)}, y_{i'}\boldsymbol{\mu} \rangle) \cdot \langle y_{i'}\boldsymbol{\mu}, \boldsymbol{\xi}_i \rangle}_{A_3}.$$

We can respectively bound each term as follows.

$$A_1 \geq \frac{\eta \|\boldsymbol{\xi}_i\|_2^2}{nm} \cdot \min_{i \in [n]} |\ell_i'^{(\tilde{t}-1)}| \geq \frac{\eta \sigma_\xi^2 d C_\ell}{2nm}$$

where the last inequality is by Lemma B.1.

For $A_2$, we can upper bound its magnitude as

$$|A_2| \leq \frac{\eta}{m} \cdot |\langle \boldsymbol{\xi}_i, \boldsymbol{\xi}_{i'} \rangle|$$
$$\leq \frac{2\eta}{m} \cdot \sigma_\xi^2 \sqrt{d \log(6n^2/\delta)}$$

where the first inequality is by $|\ell_i'^{(t)}| \leq 1$ for all $t$ and the second inequality is by Lemma B.1.

For $A_3$, similarly, we can bound

$$|A_3| \leq \frac{\eta}{m} \cdot |\langle \boldsymbol{\mu}, \boldsymbol{\xi}_i \rangle| \leq \frac{\eta \|\boldsymbol{\mu}\|_2 \sigma_\xi \sqrt{2\log(6n/\delta)}}{m}$$

where the second inequality is again by Lemma B.1. By requiring $d \geq \max\{32C_\ell^{-2} n^2 \log(6n^2/\delta), 4C_\ell^{-1} n \|\boldsymbol{\mu}\|_2 \sigma_\xi^{-1} \sqrt{2\log(6n/\delta)}\}$, we can show $A_1 \geq \max\{|A_2|/2, |A_3|/2\}$ and thus

$$\langle \mathbf{w}_{\tilde{y}_i,r}^{(\tilde{t})}, \boldsymbol{\xi}_i \rangle = \langle \mathbf{w}_{\tilde{y}_i,r}^{(\tilde{t}-1)}, \boldsymbol{\xi}_i \rangle \geq \langle \mathbf{w}_{\tilde{y}_i,r}^{(\tilde{t}-1)}, \boldsymbol{\xi}_i \rangle + A_1 - |A_2| - |A_3| > \langle \mathbf{w}_{\tilde{y}_i,r}^{(\tilde{t}-1)}, \boldsymbol{\xi}_i \rangle > 0$$

for all $r \in \mathcal{S}_i^{(\tilde{t}-1)}$. Thus, $r \in \mathcal{S}_i^{(\tilde{t})}$ and $\mathcal{S}_i^{(0)} \subseteq \mathcal{S}_i^{(\tilde{t}-1)} \subseteq \mathcal{S}_i^{(\tilde{t})}$.

For the other claim, we follow a similar strategy as above. For $i \in \mathcal{S}_{j,r}^{(0)}$, we have by induction condition that $i \in \mathcal{S}_{j,r}^{(\tilde{t}-1)}$ and thus for $j = \tilde{y}_i$

$$\langle \mathbf{w}_{j,r}^{(\tilde{t})}, \boldsymbol{\xi}_i \rangle = \langle \mathbf{w}_{j,r}^{(\tilde{t}-1)}, \boldsymbol{\xi}_i \rangle - \frac{\eta}{nm} \sum_{i'=1}^n \ell_{i'}'^{(\tilde{t}-1)} \cdot \sigma'(\langle \mathbf{w}_{j,r}^{(\tilde{t}-1)}, \boldsymbol{\xi}_{i'} \rangle) \cdot \langle \boldsymbol{\xi}_i, \boldsymbol{\xi}_{i'} \rangle$$
$$- \frac{\eta}{nm} \sum_{i'=1}^n \ell_{i'}'^{(\tilde{t}-1)} \cdot \sigma'(\langle \mathbf{w}_{j,r}^{(\tilde{t}-1)}, y_{i'} \boldsymbol{\mu} \rangle) \cdot \langle y_{i'} \boldsymbol{\mu}, \boldsymbol{\xi}_i \rangle$$

Following the same analysis, we can show $\langle \mathbf{w}_{j,r}^{(\tilde{t})}, \boldsymbol{\xi}_i \rangle \geq \langle \mathbf{w}_{j,r}^{(\tilde{t}-1)}, \boldsymbol{\xi}_i \rangle > 0$ and thus $i \in \mathcal{S}_{j,r}^{(\tilde{t})}$ and $\mathcal{S}_{j,r}^{(0)} \subseteq \mathcal{S}_{j,r}^{(\tilde{t}-1)} \subseteq \mathcal{S}_{j,r}^{(\tilde{t})}$.

Now at the end of the first stage where $\overline{\rho}_{j,r,i}^{(T_1)} = \Omega(1)$ for all $j = \tilde{y}_i, r \in \mathcal{S}_i^{(0)}$. Then we continue the proof by induction. Suppose there exists a time $\tilde{t} \geq T_1$ such that for all $T_1 \leq s \leq \tilde{t} - 1$, $\overline{\rho}_{j,r,i}^{(s)} \geq C_\rho$ for some constant $C_\rho > 0$. Then by the update of $\overline{\rho}_{j,r,i}^{(t)}$, we can show for $j = \tilde{y}_i, r \in \mathcal{S}_i^{(0)}$,

$$\overline{\rho}_{j,r,i}^{(\tilde{t})} = \overline{\rho}_{j,r,i}^{(\tilde{t}-1)} - \frac{\eta}{nm} \ell_i'^{(t)} \sigma'(\langle \mathbf{w}_{j,r}^{(t)}, \boldsymbol{\xi}_i \rangle) \|\boldsymbol{\xi}_i\|_2^2 \geq \overline{\rho}_{j,r,i}^{(\tilde{t}-1)} \geq C_\rho$$

where we notice that $-\ell_i'^{(t)} \geq 0$. Then we can show from Lemma C.3

$$\langle \mathbf{w}_{j,r}^{(\tilde{t})}, \boldsymbol{\xi}_i \rangle \geq \overline{\rho}_{j,r,i}^{(\tilde{t})} - |\langle \mathbf{w}_{j,r}^{(0)}, \boldsymbol{\xi}_i \rangle| - 5\sqrt{\frac{\log(6n^2/\delta)}{d}} n\alpha \geq C_\rho - 1/C' > 0$$

where we use the condition on $d$ to be sufficiently large and choose $C' > 1/C_\rho$. Thus we have for $r \in \mathcal{S}_i^{(\tilde{t})}$ and thus $\mathcal{S}_i^{(0)} \subseteq \mathcal{S}_i^{(\tilde{t}-1)} \subseteq \mathcal{S}_i^{(\tilde{t})}$. For the other claim, we can use the same argument. $\square$

Next, we proceed to prove Proposition C.2.

*Proof of Proposition C.2.* We prove the claims by induction. It is clear that at $t = 0$, all the claims are satisfied trivially given $\gamma_{j,r}^{(0)}, \overline{\rho}_{j,r,i}^{(0)}, \underline{\rho}_{j,r,i}^{(0)} = 0$ for all $j, r, i$. Suppose there exists $\tilde{T} \le T^*$ such that the results in Proposition C.2 hold for all time $t \le \tilde{T} - 1$. Then we have Lemma C.3, C.4, Lemma C.6 hold for all $t \le \tilde{T} - 1$.

Now we show that the results in Proposition C.2 also hold for $t = \tilde{T}$.

(1) We first show $\underline{\rho}_{j,r,i}^{(t)} \ge -\beta - 10\sqrt{\frac{\log(6n^2/\delta)}{d}}n\alpha$. When $\underline{\rho}_{j,r,i}^{(\tilde{T}-1)} \le -0.5\beta - 5\sqrt{\frac{\log(6n^2/\delta)}{d}}n\alpha$, by Lemma C.3, we have

$$\langle \mathbf{w}_{j,r}^{(\tilde{T}-1)}, \boldsymbol{\xi}_i \rangle \le \underline{\rho}_{j,r,i}^{(\tilde{T}-1)} + |\langle \mathbf{w}_{j,r}^{(0)}, \boldsymbol{\xi}_i \rangle| + 5\sqrt{\frac{\log(6n^2/\delta)}{d}}n\alpha < 0$$

and this suggests

$$
\begin{aligned}
\underline{\rho}_{j,r,i}^{(\tilde{T})} &= \underline{\rho}_{j,r,i}^{(\tilde{T}-1)} + \frac{\eta}{nm}\ell_i'^{(\tilde{T}-1)}\sigma'(\langle \mathbf{w}_{j,r}^{(\tilde{T}-1)}, \boldsymbol{\xi}_i \rangle)\|\boldsymbol{\xi}_i\|_2^2 \\
&= \underline{\rho}_{j,r,i}^{(\tilde{T}-1)} \\
&\ge -\beta - 10\sqrt{\frac{\log(6n^2/\delta)}{d}}n\alpha
\end{aligned}
$$

On the other hand, when $\underline{\rho}_{j,r,i}^{(\tilde{T}-1)} \ge -0.5\beta - 5\sqrt{\frac{\log(6n^2/\delta)}{d}}n\alpha$,

$$
\begin{aligned}
\underline{\rho}_{j,r,i}^{(\tilde{T})} &= \underline{\rho}_{j,r,i}^{(\tilde{T}-1)} + \frac{\eta}{nm}\ell_i'^{(\tilde{T}-1)}\sigma'(\langle \mathbf{w}_{j,r}^{(\tilde{T}-1)}, \boldsymbol{\xi}_i \rangle)\|\boldsymbol{\xi}_i\|_2^2 \\
&\ge -0.5\beta - 5\sqrt{\frac{\log(6n^2/\delta)}{d}}C_\rho n - \frac{3\eta\sigma_\xi^2 d}{2nm} \\
&\ge -0.5\beta - 10\sqrt{\frac{\log(6n^2/\delta)}{d}}C_\rho n \\
&\ge -\beta - 10\sqrt{\frac{\log(6n^2/\delta)}{d}}C_\rho n
\end{aligned}
$$

where we use Lemma B.1 in the first inequality. The second inequality is by the condition on $\eta$ such that $5\sqrt{\frac{\log(6n^2/\delta)}{d}}C_\rho n \ge 3\eta\sigma_\xi^2 d/(2nm)$. This completes the induction for the result on $\underline{\rho}_{j,r,i}^{(t)}$:

(2) We next prove $\gamma_{j,r}^{(\tilde{T})} \ge 0$. Towards this end, we separate the analysis in two stages. In the first stage, the loss derivatives can be lower bounded by a constant, i.e., $|\ell_i'^{(t)}| \ge C_\ell$ for all $i \in [n]$. Recall the update rule for $\gamma_{j,r}^{(t)}$ is

$$\gamma_{j,r}^{(\tilde{T})} = \gamma_{j,r}^{(\tilde{T}-1)} - \frac{\eta}{nm}\sum_{i=1}^n \ell_i'^{(\tilde{T}-1)}\sigma'(\langle \mathbf{w}_{j,r}^{(\tilde{T}-1)}, y_i\boldsymbol{\mu} \rangle)y_i\tilde{y}_i\|\boldsymbol{\mu}\|_2^2.$$

When $\langle \mathbf{w}_{j,r}^{(\tilde{T}-1)}, \boldsymbol{\mu} \rangle \ge 0$, we can show

$$
\begin{aligned}
\gamma_{j,r}^{(\tilde{T})} &= \gamma_{j,r}^{(\tilde{T}-1)} - \frac{\eta}{nm}\Big( \sum_{i \in \mathcal{S}_t \cap \mathcal{S}_1} \ell_i'^{(\tilde{T}-1)} - \sum_{i \in \mathcal{S}_f \cap \mathcal{S}_1} \ell_i'^{(\tilde{T}-1)} \Big)\|\boldsymbol{\mu}\|_2^2 \\
&\ge \gamma_{j,r}^{(\tilde{T}-1)} + \frac{\eta}{nm}\Big( \frac{1-\tau_+}{2}C_\ell - \frac{\tau_+}{2} \Big)\|\boldsymbol{\mu}\|_2^2 \\
&\ge \gamma_{j,r}^{(\tilde{T}-1)} \\
&\ge 0
\end{aligned}
$$

where in the first inequality, we uses $C_\ell \le |\ell_i'^{(t)}| \le 1$. The second inequality is by the choice $\tau_+ \le \frac{C_\ell}{C_\ell+1}$.

Similarly, when $\langle \mathbf{w}_{j,r}^{(\tilde{T}-1)}, \boldsymbol{\mu} \rangle \le 0$, we have

$$\gamma_{j,r}^{(\tilde{T})} = \gamma_{j,r}^{(\tilde{T}-1)} - \frac{\eta}{nm}\Big( \sum_{i \in \mathcal{S}_t \cap \mathcal{S}_{-1}} \ell_i'^{(\tilde{T}-1)} - \sum_{i \in \mathcal{S}_f \cap \mathcal{S}_{-1}} \ell_i'^{(\tilde{T}-1)} \Big)\|\boldsymbol{\mu}\|_2^2$$

$$\geq \gamma_{j,r}^{(\tilde{T}-1)} + \frac{\eta}{nm}\Big(\frac{1-\tau_-}{2}C_\ell - \frac{\tau_-}{2}\Big)\|\boldsymbol{\mu}\|_2^2$$

$$\geq \gamma_{j,r}^{(\tilde{T}-1)}$$

$$\geq 0$$

where we choose $\tau_- \leq \frac{C_\ell}{C_\ell+1}$.

In the second stage, we prove the claim by contradiction. First, without loss of generality that $\langle \mathbf{w}_{j,r}^{(t)}, \boldsymbol{\mu}\rangle \geq 0$, and we write the update as

$$\gamma_{j,r}^{(t+1)} = \gamma_{j,r}^{(t)} - \frac{\eta}{nm}\Big(\sum_{i\in\mathcal{S}_t\cap\mathcal{S}_1} \ell_i'^{(t)} - \sum_{i\in\mathcal{S}_f\cap\mathcal{S}_1} \ell_i'^{(t)}\Big)\|\boldsymbol{\mu}\|_2^2$$

$$= \gamma_{j,r}^{(t)} + \frac{\eta}{nm}\Big(\underbrace{\sum_{i\in\mathcal{S}_t\cap\mathcal{S}_1} \frac{1}{1+\exp\big(\tilde{y}_i f(\mathbf{W}^{(t)}, \mathbf{x}_i)\big)} - \sum_{i\in\mathcal{S}_f\cap\mathcal{S}_1} \frac{1}{1+\exp\big(\tilde{y}_i f(\mathbf{W}^{(t)}, \mathbf{x}_i)\big)}}_{A_4}\Big)\|\boldsymbol{\mu}\|_2^2$$

Suppose at an iteration $t$, $A_4 < 0$, which leads to a decrease in the $\gamma_{j,r}^{(t)}$. Then by Lemma C.4

$$A_4 \geq \sum_{i\in\mathcal{S}_t\cap\mathcal{S}_1} \frac{1}{1+\exp\big(\frac{1}{m}\sum_{r=1}^m(\overline{\rho}_{\tilde{y}_i,r,i}^{(t)} + \gamma_{\tilde{y}_i,r}^{(t)}) + 1/C_1\big)}$$

$$- \sum_{i\in\mathcal{S}_f\cap\mathcal{S}_1} \frac{1}{1+\exp\big(\frac{1}{m}\sum_{r=1}^m(\overline{\rho}_{\tilde{y}_i,r,i}^{(t)} - \gamma_{-\tilde{y}_i,r}^{(t)}) - 1/C_1\big)}$$

Then we can see the gap between loss derivatives of $\mathcal{S}_t$ and $\mathcal{S}_f$ becomes progressively smaller such that for a given $\tau_+$ (or $\tau_-$ when $\langle \mathbf{w}_{j,r}^{(t)}, \boldsymbol{\mu}\rangle \leq 0$) which is sufficiently small, $A_4 > 0$ and $\gamma_{j,r}^{(t)}$ starts to increase.

(3) Next we show upper bound for $\overline{\rho}_{\tilde{y}_i,r,i}^{(t)}$. Recall the update rule for $\overline{\rho}_{j,r,i}^{(t)}$ is

$$\overline{\rho}_{\tilde{y}_i,r,i}^{(t+1)} = \overline{\rho}_{\tilde{y}_i,r,i}^{(t)} - \frac{\eta}{nm}\ell_i'^{(t)}\sigma'(\langle \mathbf{w}_{j,r}^{(t)}, \boldsymbol{\xi}_i\rangle)\|\boldsymbol{\xi}_i\|_2^2$$

Now suppose $t_{r,i}$ be the last time $t < T^*$ such that $\overline{\rho}_{\tilde{y}_i,r,i}^{(t)} \leq 0.5\alpha$. Then

$$\overline{\rho}_{\tilde{y}_i,r,i}^{(\tilde{T})} = \overline{\rho}_{\tilde{y}_i,r,i}^{(t_{r,i})} - \frac{\eta}{nm}\ell_i'^{(t)}\sigma'(\langle \mathbf{w}_{\tilde{y}_i,r}^{(t_{r,i})}, \boldsymbol{\xi}_i\rangle)\|\boldsymbol{\xi}_i\|_2^2 - \frac{\eta}{nm}\sum_{t_{r,i}<t<\tilde{T}} \ell_i'^{(t)}\sigma'(\langle \mathbf{w}_{\tilde{y}_i,r}^{(t)}, \boldsymbol{\xi}_i\rangle)\|\boldsymbol{\xi}_i\|_2^2$$

$$\leq \overline{\rho}_{\tilde{y}_i,r,i}^{(t_{r,i})} + \frac{3\eta\sigma_\xi^2 d}{2nm} - \frac{\eta}{nm}\sum_{t_{r,i}<t<\tilde{T}} \ell_i'^{(t)}\sigma'(\langle \mathbf{w}_{\tilde{y}_i,r}^{(t)}, \boldsymbol{\xi}_i\rangle)\|\boldsymbol{\xi}_i\|_2^2$$

$$\leq 0.5\alpha + 0.25\alpha - \frac{\eta}{nm}\sum_{t_{r,i}<t<\tilde{T}} \ell_i'^{(t)}\sigma'(\langle \mathbf{w}_{\tilde{y}_i,r}^{(t)}, \boldsymbol{\xi}_i\rangle)\|\boldsymbol{\xi}_i\|_2^2 \qquad (7)$$

where we apply Lemma B.1 for the first inequality and choose $\eta \leq C^{-1}n\sigma_\xi^{-2}d^{-1}$ for the last inequality. Then we bound the last term for $t \in (t_{r,i}, \tilde{T})$ as

$$-\ell_i'^{(t)} = \frac{1}{1+\exp(\tilde{y}_i f(\mathbf{W}^{(t)}, \mathbf{x}_i))} \leq \exp(-\tilde{y}_i f(\mathbf{W}^{(t)}, \mathbf{x}_i))$$

Next we consider two cases depending on whether $i \in \mathcal{S}_t$ or $i \in \mathcal{S}_f$.

- When $i \in \mathcal{S}_t$, we can bound by Lemma C.4

$$\tilde{y}_i f(\mathbf{W}^{(t)}, \mathbf{x}_i) \geq \frac{1}{m}\sum_{r=1}^m(\gamma_{\tilde{y}_i,r}^{(t)} + \overline{\rho}_{\tilde{y}_i,r,i}^{(t)}) - 1/C_1 \geq \frac{1}{m}\sum_{r=1}^m \overline{\rho}_{\tilde{y}_i,r,i}^{(t)} - 1/C_1 \geq 0.5\alpha - 0.1.$$

where the second inequality is by $\gamma_{\tilde{y}_i,r}^{(t)} \geq 0$ and the last inequality is by choosing $C_1 \geq 10$. Then this suggests

$$-\ell_i'^{(t)} \leq \exp(-\tilde{y}_i f(\mathbf{W}^{(t)}, \mathbf{x}_i)) \leq 2\exp(-0.5\alpha) \leq 2/T^*$$

where the last inequality is by choosing $C_t \geq 2$.

- When $i \in \mathcal{S}_f$, we can bound by Lemma C.4

$$\tilde{y}_i f(\mathbf{W}^{(t)}, \mathbf{x}_i) \geq \frac{1}{m} \sum_{r=1}^{m} (\overline{\rho}_{\tilde{y}_i,r,i}^{(t)} - \gamma_{-\tilde{y}_i,r}^{(t)}) - 1/C_1 \geq \frac{1}{m} \sum_{r=1}^{m} \overline{\rho}_{\tilde{y}_i,r,i}^{(t)} - 1/C_1 \geq 0.5\alpha - 0.1.$$

Here we only consider the case when $\frac{1}{m}\sum_{r=1}^{m} \overline{\rho}_{\tilde{y}_i,r,i}^{(t)} > \frac{1}{m}\sum_{r=1}^{m} \gamma_{-\tilde{y}_i,r}^{(t)}$ when deriving the upper bound for $\frac{1}{m}\sum_{r=1}^{m} \overline{\rho}_{\tilde{y}_i,r,i}^{(t)}$ because otherwise, the loss cannot converge to arbitrarily small as we show later. To see this, we suppose $\frac{1}{m}\sum_{r=1}^{m} \overline{\rho}_{\tilde{y}_i,r,i}^{(T^*)} \leq \frac{1}{m}\sum_{r=1}^{m} \gamma_{-\tilde{y}_i,r}^{(T^*)}$ at termination time. Then for such sample, $\tilde{y}_i f(\mathbf{W}^{(t)}, \mathbf{x}_i) \leq \frac{1}{m}\sum_{r=1}^{m}(\overline{\rho}_{\tilde{y}_i,r,i}^{(T^*)} - \gamma_{-\tilde{y}_i,r}^{(T^*)}) + 1/C_1 \leq 0.1$ the loss can be lower bounded as $\ell_i^{(T^*)} = \log(1+\exp(-\tilde{y}_i f(\mathbf{W}^{(T^*)}, \mathbf{x}_i))) \geq \log(1+\exp(-0.1)) \geq 0.6$.

Hence we let $c' := (\frac{1}{m}\sum_{r=1}^{m} \overline{\rho}_{\tilde{y}_i,r,i}^{(t)})/(\frac{1}{m}\sum_{r=1}^{m} \gamma_{-\tilde{y}_i,r}^{(t)}) > 1$. Then

$$\tilde{y}_i f(\mathbf{W}^{(t)}, \mathbf{x}_i) \geq \frac{1}{m} \sum_{r=1}^{m} (1 - 1/c') \overline{\rho}_{\tilde{y}_i,r,i}^{(t)} - 1/C_1 \geq (1 - 1/c') 0.5\alpha - 0.1.$$

Then we have

$$-\ell_i'^{(t)} \leq \exp(-\tilde{y}_i f(\mathbf{W}^{(t)}, \mathbf{x}_i)) \leq 2\exp\big(-(1-1/c')0.5\alpha\big) \leq 2/T^*$$

where the last inequality is by choosing $C_t$ sufficiently large.

In both cases, (7) can be further bounded as

$$\overline{\rho}_{\tilde{y}_i,r,i}^{(\widetilde{T})} \leq 0.75\alpha + \frac{3\eta\sigma_\xi^2 dT^*}{2nm} \cdot \frac{2}{T^*} \leq \alpha$$

where the first inequality is by upper bound on the loss derivatives and the last inequality is by the condition on Condition 4.1 where $\eta \leq C^{-1}n\sigma_\xi^{-2}d^{-1} \leq nm\sigma_\xi^{-2}d^{-1}/3$.

(4) Finally for the upper bound on $\gamma_{j,r}^{(t)}$, we can verify that by the update of $\gamma_{j,r}^{(t)}$

$$\gamma_{j,r}^{(t+1)} = \gamma_{j,r}^{(t)} - \frac{\eta}{nm}\Big(\sum_{i\in\mathcal{S}_t} \ell_i'^{(t)} \mathbb{1}(\langle \mathbf{w}_{j,r}^{(t)}, y_i\boldsymbol{\mu}\rangle \geq 0) - \sum_{i\in\mathcal{S}_f} \ell_i'^{(t)} \mathbb{1}(\langle \mathbf{w}_{j,r}^{(t)}, y_i\boldsymbol{\mu}\rangle \geq 0)\Big)\|\boldsymbol{\mu}\|_2^2$$

$$\leq \gamma_{j,r}^{(t)} + \frac{\eta}{nm} \sum_{i\in\mathcal{S}_t} |\ell_i'^{(t)}| \|\boldsymbol{\mu}\|_2^2$$

$$\leq \gamma_{j,r}^{(t)} + \frac{\eta\|\boldsymbol{\mu}\|_2^2}{nm} \sum_{i\in\mathcal{S}_t} \frac{1}{1 + \exp\big(\frac{1}{m}\sum_{r=1}^{m}(\overline{\rho}_{\tilde{y}_i,r,i}^{(t)} + \gamma_{\tilde{y}_i,r}^{(t)}) - 1/C_1\big)}$$

Suppose $t_{j,r}$ be the last time $t < T^*$ such that $\gamma_{j,r}^{(t)} \leq 0.5C_\gamma\alpha$. Then

$$\gamma_{j,r}^{(\widetilde{T})} \leq \gamma_{j,r}^{(t_{j,r})} + \frac{\eta(2-\tau_+-\tau_-)\|\boldsymbol{\mu}\|_2^2}{2m} T^* \frac{1}{1 + \exp(0.5C_\gamma C_t \log(T^*) - 0.1)}$$

$$\leq \gamma_{j,r}^{(t_{j,r})} + \frac{\eta(2-\tau_+-\tau_-)\|\boldsymbol{\mu}\|_2^2}{2m} T^* \cdot 2\exp(-0.5C_\gamma C_t \log(T^*))$$

$$\leq \gamma_{j,r}^{(t_{j,r})} + \frac{\eta(2-\tau_+-\tau_-)\|\boldsymbol{\mu}\|_2^2}{m}$$

$$\leq C_\gamma\alpha$$

where we notice $\overline{\rho}_{j,r,i}^{(t)} \geq 0$ and we let $C_1 \geq 10, C_\gamma \geq 1$. The last inequality follows from Condition 4.1 where $\eta \leq C^{-1}n\sigma_\xi^{-2}d^{-1} \leq 0.5(2-\tau_+-\tau_-)^{-1}m\|\boldsymbol{\mu}\|_2^{-2}$, where the last inequality is by condition on $\|\boldsymbol{\mu}\|_2^2$ and $d$. Thus the proof is now complete. $\qquad\square$

## C.3. First Stage

Next we consider first stage of the training dynamics. In this stage, before the coefficients $\gamma_{j,r}^{(t)}, \overline{\rho}_{j,r,i}^{(t)}$ reach a constant order, we can both lower and upper bound the loss derivatives by an absolute constant, i.e., $\underline{C}_\ell \leq |\ell_i'^{(t)}| \leq \overline{C}_\ell$. Here we suggests $\underline{C}_\ell = 0.49$ and $\overline{C}_\ell = 0.51$ is sufficient to show the desired result.

Before we proceed to prove Theorem 4.1, we provide a tighter bound on $\|\boldsymbol{\xi}_i\|_2^2$ and $|\mathcal{S}_i^{(0)}|$ compared to Lemma B.1 and Lemma B.3 respectively.

**Lemma C.7.** *With probability at least $1 - \delta$, we can bound*

$$\sigma_\xi^2 d(1 - \widetilde{O}(1/\sqrt{d})) \leq \|\boldsymbol{\xi}_i\|_2^2 \leq \sigma_\xi^2 d(1 + \widetilde{O}(1/\sqrt{d}))$$

*Proof.* By Bernstein inequality, with probability at least $1 - \delta/n$, we have $\left|\|\boldsymbol{\xi}_i\|_2^2 - \sigma_p^2 d\right| = O\left(\sigma_p^2 \cdot \sqrt{d \log(6n/\delta)}\right)$. Then taking the union bound gives the desired result. $\square$

**Lemma C.8.** *With probability at least $1 - \delta$, we can bound*

$$\frac{m}{2}\left(1 - \widetilde{O}(1/\sqrt{m})\right) \leq |\mathcal{S}_i^{(0)}| \leq \frac{m}{2}\left(1 + \widetilde{O}(1/\sqrt{m})\right)$$

*Proof.* Because $\mathbb{P}(\langle \mathbf{w}_{\tilde{y}_i,r}^{(0)}, \boldsymbol{\xi}_i \rangle > 0) = 0.5$, by Hoeffding inequality, with probability at least $1 - \delta/n$, we can bound $\left|\frac{|\mathcal{S}_i^{(0)}|}{m} - \frac{1}{2}\right| \leq \sqrt{\frac{\log(2n/\delta)}{2m}}$. Then taking union bound gives the desired result. $\square$

**Theorem C.9** (Restatement of Theorem 4.1). *Under Condition 4.1, there exists $T_1 = \Theta\left(\eta^{-1}nm\sigma_\xi^{-2}d^{-1}\right)$ such that*

1. $\overline{\rho}_{\tilde{y}_i,r,i}^{(T_1)} = \Theta(1)$ *for all $i \in [n], r \in [m]$ such that $\langle \mathbf{w}_{\tilde{y}_i,r}^{(0)}, \mathbf{x}_i \rangle \geq 0$.*

2. $\gamma_{j,r}^{(T_1)} = \Theta(1)$ *for all $j = \pm 1, r \in [m]$.*

3. $\gamma_{j,r}^{(T_1)} > \overline{\rho}_{\tilde{y}_i,r,i}^{(T_1)}$ *for all $j = \pm 1, r \in [m], i \in [n]$.*

4. *All clean samples $i \in \mathcal{S}_t$ satisfy that $\tilde{y}_i f(\mathbf{W}^{(T_1)}, \mathbf{x}_i) \geq 0$.*

5. *All noisy samples $i \in \mathcal{S}_f$ satisfy that $\tilde{y}_i f(\mathbf{W}^{(T_1)}, \mathbf{x}_i) \leq 0$.*

*Proof of Theorem C.9.* We first show the lower and upper bound for noise dynamics. For any $i \in [n]$ and $r \in \mathcal{S}_i^{(0)}$, by Lemma C.6, we know that $r \in \mathcal{S}_i^{(t)}$ for all $t \leq T_1$. Hence, by the update of noise coefficients,

$$\overline{\rho}_{\tilde{y}_i,r,i}^{(t)} = \overline{\rho}_{\tilde{y}_i,r,i}^{(t-1)} - \frac{\eta}{nm}\ell_i'^{(t-1)}\|\boldsymbol{\xi}_i\|_2^2 \geq \overline{\rho}_{\tilde{y}_i,r,i}^{(t-1)} + 0.99 \cdot \frac{\eta C_\ell \sigma_\xi^2 d}{nm} = 0.99 \cdot \frac{\eta C_\ell \sigma_\xi^2 d}{nm}t$$

where the first inequality is by Lemma C.7 where we choose $d = \Omega(\log(n/\delta))$ sufficiently large and the loss derivative lower bound in this stage. The second inequality is by iterating the first inequality to $t = 0$ and by noticing $\overline{\rho}_{\tilde{y}_i,r,i}^{(0)} = 0$.

For the upper bound, for $i \in [n]$,

$$\overline{\rho}_{\tilde{y}_i,r,i}^{(t)} = \overline{\rho}_{\tilde{y}_i,r,i}^{(t-1)} + \frac{\eta}{nm}|\ell_i'^{(t-1)}|\|\boldsymbol{\xi}_i\|_2^2 \leq \overline{\rho}_{\tilde{y}_i,r,i}^{(t-1)} + 1.01 \cdot \frac{\eta\sigma_\xi^2 d}{nm} \leq 1.01 \cdot \frac{\eta\sigma_\xi^2 d}{nm}t$$

where we use Lemma C.7 and $|\ell_i'^{(t)}| \leq 1$.

Next, we lower and upper bound the signal dynamics. Recall the update rule for $\gamma_{j,r}^{(t)}$ as

$$\gamma_{j,r}^{(t)} = \gamma_{j,r}^{(t-1)} - \frac{\eta}{nm}\sum_{i=1}^n \ell_i'^{(t)}\sigma'(\langle \mathbf{w}_{j,r}^{(t-1)}, y_i\boldsymbol{\mu}\rangle)y_i\tilde{y}_i\|\boldsymbol{\mu}\|_2^2$$

$$= \gamma_{j,r}^{(t-1)} - \frac{\eta}{nm} \left( \sum_{i \in \mathcal{S}_t} \ell_i'^{(t)} \sigma'(\langle \mathbf{w}_{j,r}^{(t-1)}, y_i \boldsymbol{\mu} \rangle) - \sum_{i \in \mathcal{S}_f} \ell_i'^{(t)} \sigma'(\langle \mathbf{w}_{j,r}^{(t-1)}, y_i \boldsymbol{\mu} \rangle) \right) \|\boldsymbol{\mu}\|_2^2$$

When $\langle \mathbf{w}_{j,r}^{(t-1)}, \boldsymbol{\mu} \rangle \geq 0$,

$$\gamma_{j,r}^{(t)} = \gamma_{j,r}^{(t-1)} - \frac{\eta}{nm} \left( \sum_{i \in \mathcal{S}_t \cap \mathcal{S}_1} \ell_i'^{(t-1)} - \sum_{i \in \mathcal{S}_f \cap \mathcal{S}_1} \ell_i'^{(t-1)} \right) \|\boldsymbol{\mu}\|_2^2$$

$$\geq \gamma_{j,r}^{(t-1)} + \frac{\eta}{nm} \left( \frac{(1-\tau_+)nC_\ell}{2} - \frac{\tau_+ n}{2} \right) \|\boldsymbol{\mu}\|_2^2$$

$$\geq \gamma_{j,r}^{(t-1)} + 0.49 \cdot \frac{\eta \|\boldsymbol{\mu}\|_2^2 C_\ell}{m}$$

where the first inequality uses the lower bound and upper bound on loss derivatives, i.e., $C_\ell \leq |\ell_i'^{(t)}| \leq 1$. The second inequality follows by letting $\tau_+ \leq \frac{0.02 C_\ell}{1+C_\ell}$.

Similarly, when $\langle \mathbf{w}_{j,r}^{(t-1)}, \boldsymbol{\mu} \rangle < 0$,

$$\gamma_{j,r}^{(t)} = \gamma_{j,r}^{(t-1)} - \frac{\eta}{nm} \left( \sum_{i \in \mathcal{S}_t \cap \mathcal{S}_{-1}} \ell_i'^{(t-1)} - \sum_{i \in \mathcal{S}_f \cap \mathcal{S}_{-1}} \ell_i'^{(t-1)} \right) \|\boldsymbol{\mu}\|_2^2$$

$$\geq \gamma_{j,r}^{(t-1)} + \frac{\eta}{nm} \left( \frac{(1-\tau_-)nC_\ell}{2} - \frac{\tau_- n}{2} \right) \|\boldsymbol{\mu}\|_2^2$$

$$\geq \gamma_{j,r}^{(t-1)} + 0.49 \cdot \frac{\eta \|\boldsymbol{\mu}\|_2^2 C_\ell}{m}$$

where we let $\tau_- \leq \frac{0.02 C_\ell}{1+C_\ell}$. Combining both cases, we can iterate the inequality, which gives

$$\gamma_{j,r}^{(t)} \geq \gamma_{j,r}^{(0)} + \frac{\eta \|\boldsymbol{\mu}\|_2^2 C_\ell}{4m} t = 0.49 \cdot \frac{\eta \|\boldsymbol{\mu}\|_2^2 C_\ell}{m} t$$

We first show the claim that $\tilde{y}_i f(\mathbf{W}^{(T_1)}, \mathbf{x}_i) \geq 0$ for $i \in \mathcal{S}_t$. By the update of signal and noise coefficients, we have for all any $i \in \mathcal{S}_t$, and $r \in \mathcal{S}_i^{(0)}$, by , we have $r \in \mathcal{S}_i^{(t)}$ and thus for all $t \leq T_1$,

For the upper bound, we obtain from the signal dynamics that

$$\gamma_{j,r}^{(t)} = \gamma_{j,r}^{(t-1)} - \frac{\eta}{nm} \left( \sum_{i \in \mathcal{S}_t} \ell_i'^{(t)} \sigma'(\langle \mathbf{w}_{j,r}^{(t-1)}, y_i \boldsymbol{\mu} \rangle) - \sum_{i \in \mathcal{S}_f} \ell_i'^{(t)} \sigma'(\langle \mathbf{w}_{j,r}^{(t-1)}, y_i \boldsymbol{\mu} \rangle) \right) \|\boldsymbol{\mu}\|_2^2$$

$$\leq \gamma_{j,r}^{(t-1)} + \frac{\eta \|\boldsymbol{\mu}\|_2^2}{m} \frac{1-\tau_\pm}{2}$$

$$\leq \gamma_{j,r}^{(t-1)} + \frac{\eta \|\boldsymbol{\mu}\|_2^2}{2m}$$

$$\leq \frac{\eta \|\boldsymbol{\mu}\|_2^2}{2m} t$$

where we use the upper bound on loss derivative in the first inequality.

Now we verify the conditions such that the claims are satisfied. First, we set a termination time for the first stage as $T_1$, where

$$T_1 = C_2 \eta^{-1} C_\ell^{-1} nm \sigma_\xi^{-2} d^{-1}$$

for some constant $C_2 > 0$ to be chosen later. This suggests at the end of first stage, we have

- $\overline{\rho}_{\tilde{y}_i, r, i}^{(T_1)} \geq 0.99 \cdot C_2$, for all $i \in [n]$ and $r \in \mathcal{S}_i^{(0)}$

- $\overline{\rho}_{\tilde{y}_i,r,i}^{(T_1)} \leq 1.01 \cdot C_2 C_\ell^{-1}$ for all $i \in [n]$.

- $\gamma_{j,r}^{(T_1)} \geq 0.49 C_2 n \cdot \mathrm{SNR}^2$ for all $j = \pm 1, r \in [m]$.

- $\gamma_{j,r}^{(T_1)} \leq 0.5 C_2 n \cdot \mathrm{SNR}^2$ for all $j = \pm 1, r \in [m]$.

Then for all $i \in \mathcal{S}_t$, we have by Lemma C.4,

$$\tilde{y}_i f(\mathbf{W}^{(t)}, \mathbf{x}_i) \geq \frac{1}{m} \sum_{r=1}^m \left( \gamma_{\tilde{y}_i,r}^{(t)} + \overline{\rho}_{\tilde{y}_i,r,i}^{(t)} \right) - 1/C_1 \geq 0.49 C_2 n \cdot \mathrm{SNR}^2 + 0.99 C_2 - 1/C_1 > 0$$

where the last inequality is by choosing $C_1$ sufficiently large, e.g., $C_1 \geq (0.99 C_2)^{-1}$. This verifies the first claim of Theorem 4.1.

Second, for all $i \in \mathcal{S}_f$, we have by Lemma C.4,

$$\tilde{y}_i f(\mathbf{W}^{(t)}, \mathbf{x}_i) \leq \frac{1}{m} \sum_{r=1}^m \left( - \gamma_{-\tilde{y}_i,r}^{(t)} + \overline{\rho}_{\tilde{y}_i,r,i}^{(t)} \right) + 1/C_1 \leq -0.49 C_2 n \cdot \mathrm{SNR}^2 + 1.01 C_2 C_\ell^{-1} + 1/C_1$$

$$< 0$$

where the last inequality is by the choice that $C_1 \geq 1000$ and

$$n \cdot \mathrm{SNR}^2 \geq 2.07 C_\ell^{-1} + 0.002 C_2^{-1} \tag{8}$$

Under such condition, we also verify the third claim of Theorem C.9.

It remains to analyze the condition under which the loss derivative is lower bounded by $C_\ell$. In particular, we require $\min_{i \in [n], t \leq T_1} |\ell_i'^{(t)}| \geq C_\ell$, where

$$
\begin{aligned}
\min_{i \in [n], t \leq T_1} |\ell_i'^{(t)}| = \min_{i \in \mathcal{S}_t} |\ell_i'^{(T_1)}| &= \frac{1}{1 + \exp(\max_{i \in \mathcal{S}_t} \tilde{y}_i f(\mathbf{W}^{(T_1)}, \mathbf{x}_i))} \\
&\geq \frac{1}{1 + \exp\left( \frac{1}{m} \sum_{r=1}^m (\gamma_{\tilde{y}_{i^*},r}^{(T_1)} + \overline{\rho}_{\tilde{y}_{i^*},r,i^*}^{(T_1)}) + 1/C_1 \right)} \\
&\geq \frac{1}{1 + \exp\left( 1.01 C_2 C_\ell^{-1} + 0.5 C_2 n \cdot \mathrm{SNR}^2 + 1/C_1 \right)}
\end{aligned}
$$

where we denote $i^* = \arg\max_{i \in \mathcal{S}_t} \tilde{y}_i f(\mathbf{W}^{(T_1)}, \mathbf{x}_i)$ and apply Lemma C.4.

Thus to ensure $\min_{i \in [n], t \leq T_1} |\ell_i'^{(t)}| \geq C_\ell$, we require $1.01 C_2 C_\ell^{-1} + 0.5 C_2 n \cdot \mathrm{SNR}^2 + 1/C_1 \leq \log(C_\ell^{-1} - 1)$, which translates to

$$n \cdot \mathrm{SNR}^2 \leq 2 \log(C_\ell^{-1} - 1) C_2^{-1} - 2.02 C_\ell^{-1} - 0.002 C_2^{-1} \tag{9}$$

where we choose $C_1 \geq 1000$.

The final step is to show there exists a combination of $C_2$, $n \cdot \mathrm{SNR}^2$ and $C_\ell$ such that conditions (8) and (9) are satisfied. For this, we can fix $C_\ell = 0.4$ for example and thus

$$5.175 + 0.002 C_2^{-1} \leq n \cdot \mathrm{SNR}^2 \leq 0.34 C_2^{-1} - 5.05$$

Then let $C_2 = 1/31$, we have $5.237 \leq n \cdot \mathrm{SNR}^2 \leq 5.49$. Thus the proof is complete. $\qquad \square$

### C.4. Second Stage

The second stage aims to show convergence of the training dynamics. By the end of the first stage, without loss of generality, we set $C_2 = 2.1$ and we can see

- For all $i \in [n]$, $r \in \mathcal{S}_i^{(0)}$, we have $\overline{\rho}_{\tilde{y}_i, r, i}^{(T_1)} \geq 2$;

- For all $j = \pm 1$, $r \in [m]$, we have $\gamma_{j,r}^{(t)} = \Omega(n \cdot \mathrm{SNR}^2)$, where $n \cdot \mathrm{SNR}^2 = \Theta(1)$.

- $\max_{j,r,i} |\underline{\rho}_{j,r,i}^{(T_1)}| \leq 1/C$ for some sufficiently large constant $C > 0$.

In the second stage, we show that in order to achieve convergence in loss to arbitrary tolerance, noise coefficients for noisy samples would first surpass signal coefficients by a large margin. To this end, we first show convergence in the loss function.

First, we let $\mathbf{w}_{j,r}^* = \mathbf{w}_{j,r}^{(0)} + 5 \log(2/\epsilon) \sum_{i=1}^n \frac{\boldsymbol{\xi}_i}{\|\boldsymbol{\xi}_i\|_2^2} \mathbb{1}(\tilde{y}_i = j)$.

**Lemma C.10.** *Under Condition 4.1, we can show* $\|\mathbf{W}^{(T_1)} - \mathbf{W}^*\|_F \leq \widetilde{O}(m^{1/2} n^{1/2} \sigma_\xi^{-1} d^{-1/2})$.

*Proof of Lemma C.10.* From the decomposition at $T_1$, we can show

$$\|\mathbf{W}^{(T_1)} - \mathbf{W}^*\|_F \leq \|\mathbf{W}^{(T_1)} - \mathbf{W}^{(0)}\|_F + \|\mathbf{W}^* - \mathbf{W}^{(0)}\|_F$$

$$\leq O(\sqrt{m}) \max_{j,r} \gamma_{j,r}^{(T_1)} \|\boldsymbol{\mu}\|_2^{-1} + O(\sqrt{m}) \max_{j,r} \left\| \sum_{i=1}^n \overline{\rho}_{j,r,i}^{(T_1)} \cdot \frac{\boldsymbol{\xi}_i}{\|\boldsymbol{\xi}_i\|_2^2} + \sum_{i=1}^n \underline{\rho}_{j,r,i}^{(T_1)} \cdot \frac{\boldsymbol{\xi}_i}{\|\boldsymbol{\xi}_i\|_2^2} \right\|_2$$

$$+ O(m^{1/2} n^{1/2} \log(1/\epsilon) \sigma_\xi^{-1} d^{-1/2})$$

$$= O(m^{1/2} n \cdot \mathrm{SNR}^2 \|\boldsymbol{\mu}\|_2^{-1}) + \widetilde{O}(m^{1/2} n^{1/2} \sigma_\xi^{-1} d^{-1/2})$$

$$= \widetilde{O}(m^{1/2} n^{1/2} \sigma_\xi^{-1} d^{-1/2})$$

where the last inequality is by $n \cdot \mathrm{SNR}^2 = \Theta(1)$. $\qquad\square$

**Lemma C.11.** *Under Condition 4.1, we can show that for all* $T_1 \leq t \leq T^*$,

$$\tilde{y}_i \langle \nabla f(\mathbf{W}^{(t)}), \mathbf{W}^* \rangle \geq \log(2/\epsilon)$$

*Proof of Lemma C.11.* By the gradient decomposition, we can write

$$\tilde{y}_i \langle \nabla f(\mathbf{W}^{(t)}, \mathbf{x}_i), \mathbf{W}^* \rangle$$

$$= \frac{1}{m} \sum_{j,r} \sigma'(\langle \mathbf{w}_{j,r}^{(t)}, y_i \boldsymbol{\mu} \rangle) \langle \boldsymbol{\mu}, j \cdot \mathbf{w}_{j,r}^* \rangle + \frac{1}{m} \sum_{j,r} \sigma'(\langle \mathbf{w}_{j,r}^{(t)}, \boldsymbol{\xi}_i \rangle) \langle y_i \boldsymbol{\xi}_i, j \cdot \mathbf{w}_{j,r}^* \rangle$$

$$\geq \frac{1}{m} \sum_{j=\tilde{y}_i, r} \sigma'(\langle \mathbf{w}_{j,r}^{(t)}, \boldsymbol{\xi}_i \rangle) 5 \log(2/\epsilon) - \frac{1}{m} \sum_{j,r} \sum_{i' \neq i} \sigma'(\langle \mathbf{w}_{j,r}^{(t)}, \boldsymbol{\xi}_i \rangle) 5 \log(2/\epsilon) \widetilde{O}(d^{-1/2})$$

$$- \frac{1}{m} \sum_{j,r} \sum_{i'=1}^n \sigma'(\langle \mathbf{w}_{j,r}^{(t)}, y_i \boldsymbol{\mu} \rangle) 5 \log(2/\epsilon) \widetilde{O}(n^{-1} \|\boldsymbol{\mu}\|_2^{-1}) - \frac{1}{m} \sum_{j,r} \sigma'(\langle \mathbf{w}_{j,r}^{(t)}, y_i \boldsymbol{\mu} \rangle) \widetilde{O}(\sigma_0 \|\boldsymbol{\mu}\|_2)$$

$$- \frac{1}{m} \sum_{j,r} \sigma'(\langle \mathbf{w}_{j,r}^{(t)}, \boldsymbol{\xi}_i \rangle) \widetilde{O}\left(\sigma_0 \sigma_\xi \sqrt{d}\right)$$

$$\geq 2 \log(2/\epsilon) - \log(2/\epsilon)$$

$$= \log(2/\epsilon)$$

where in the first inequality, we use the expression of $\mathbf{w}_{j,r}^*$ and Lemma B.1. The second inequality is by

$$\frac{1}{m} \sum_{j=\tilde{y}_i, r} \sigma'(\langle \mathbf{w}_{j,r}^{(t)}, \boldsymbol{\xi}_i \rangle) 5 \log(2/\epsilon) \geq \frac{1}{m} |\mathcal{S}_i^{(t)}| 5 \log(2/\epsilon) \geq 2 \log(2/\epsilon)$$

where we use Lemma C.6 and Lemma C.8 that $|\mathcal{S}_i^{(t)}| \geq 0.4m$. Further the other terms can be bounded by arbitrarily small constant. This completes the proof. $\qquad\square$

**Theorem C.12** (Restatement of Theorem 4.2). *Under Condition 4.1, for arbitrary $\epsilon > 0$, there exists $t^* \in [T_1, T^*]$, where $T^* = \widetilde{\Theta}(\eta^{-1}\epsilon^{-1}nm\sigma_\xi^{-2}d^{-1})$, such that*

1. *Training loss converges, i.e., $L_S(\mathbf{W}^{(t^*)}) \leq \epsilon$*

2. *All clean samples, i.e., $i \in \mathcal{S}_t$, it holds that $\tilde{y}_i f(\mathbf{W}^{(t^*)}, \mathbf{x}_i) \geq 0$*

3. *There exists a constant $0 < \tau' \leq \frac{\tau_+ + \tau_-}{2}$ such that there are $\tau'n$ noisy samples, i.e., $i \in \mathcal{S}_f$ satisfy $\tilde{y}_i f(\mathbf{W}^{(t^*)}, \mathbf{x}_i) \geq 0$.*

4. *The test error $L_D^{0-1}(\mathbf{W}^{(t^*)}) \geq 0.5 \min\{\tau_+, \tau_-\}$.*

*Proof of Theorem C.12.* (1) First, we prove that the loss converges to arbitrarily small tolerance. Specifically, we use Lemma D.4 of (Kou et al., 2023) to bound for all $t \leq T^*$, we have

$$\|\nabla L_S(\mathbf{W}^{(t)})\|_F^2 = O(\max\{\|\boldsymbol{\mu}\|_2^2, \sigma_\xi^2 d\})L_S(\mathbf{W}^{(t)}) \tag{10}$$

Then we bound the difference in the distance to optimal solution as

$$
\begin{aligned}
&\|\mathbf{W}^{(t)} - \mathbf{W}^*\|_F^2 - \|\mathbf{W}^{(t+1)} - \mathbf{W}^*\|_F^2 \\
&= 2\eta\langle\nabla L_S(\mathbf{W}^{(t)}), \mathbf{W}^{(t)} - \mathbf{W}^*\rangle - \eta^2\|\nabla L_S(\mathbf{W}^{(t)})\|_F^2 \\
&= \frac{2\eta}{n}\sum_{i=1}^n \ell_i'^{(t)}\left[\tilde{y}_i f(\mathbf{W}^{(t)}, \mathbf{x}_i) - \langle\nabla f(\mathbf{W}^{(t)}, \mathbf{x}_i), \mathbf{W}^*\rangle\right] - \eta^2\|\nabla L_S(\mathbf{W}^{(t)})\|_F^2 \\
&\geq \frac{2\eta}{n}\sum_{i=1}^n \ell_i'^{(t)}\left[\tilde{y}_i f(\mathbf{W}^{(t)}, \mathbf{x}_i) - \log(2/\epsilon)\right] - \eta^2\|\nabla L_S(\mathbf{W}^{(t)})\|_F^2 \\
&\geq \frac{2\eta}{n}\sum_{i=1}^n \left[\ell\left(f(\mathbf{W}^{(t)}, \mathbf{x}_i), \tilde{y}_i\right) - \epsilon/2\right] - \eta^2\|\nabla L_S(\mathbf{W}^{(t)})\|_F^2 \\
&\geq 2\eta L_S(\mathbf{W}^{(t)}) - \eta\epsilon - \eta^2 O(\max\{\|\boldsymbol{\mu}\|_2^2, \sigma_\xi^2 d\})L_S(\mathbf{W}^{(t)}) \\
&\geq \eta L_S(\mathbf{W}^{(t)}) - \eta\epsilon
\end{aligned}
$$

where the first inequality is due to Lemma C.11 and the second inequality is by convexity of cross entropy function. The third inequality is by (10) and the last inequality is by choosing $\eta \leq C^{-1}\max\{\|\boldsymbol{\mu}\|_2^2, \sigma_\xi^2 d\}^{-1}$ to be sufficiently small.

Finally, we telescope the inequality from $t = T_1$ to $t = T^*$, which yields

$$\frac{1}{T^* - T_1 + 1}\sum_{s=T_1}^{T^*} L_S(\mathbf{W}^{(s)}) \leq \frac{\|\mathbf{W}^{(T_1)} - \mathbf{W}^*\|_F^2}{\eta(T^* - T_1 + 1)} + \epsilon \leq 2\epsilon$$

where the last inequality is due to the choice of $T^* = T_1 + \lfloor\eta^{-1}\epsilon^{-1}\|\mathbf{W}^{(T_1)} - \mathbf{W}^*\|_F^2\rfloor = T_1 + \widetilde{O}(\eta^{-1}\epsilon^{-1}mnd^{-1}\sigma_\xi^{-2})$.

This suggests there exists an iteration $t^* \in [T_1, T^*]$ where $L_S(\mathbf{W}^{(t^*)}) \leq 2\epsilon$ for any $\epsilon < 0$. By setting $\epsilon \leftarrow 2\epsilon$, we verify the first claim.

(2) For the second claim, it is easy to see by Lemma C.4, for all $i \in \mathcal{S}_t$

$$\tilde{y}_i f(\mathbf{W}^{(t^*)}, \mathbf{x}_i) \geq \frac{1}{m}\sum_{r=1}^m \left(\overline{\rho}_{\tilde{y}_i, r, i}^{(t^*)} + \gamma_{\tilde{y}_i, r}^{(t^*)}\right) - 1/C_1 \geq 0$$

where the second inequality is by $\gamma_{\tilde{y}_i, r} \geq 0$ and $\overline{\rho}_{\tilde{y}_i, r, i}^{(t^*)} \geq \overline{\rho}_{\tilde{y}_i, r, i}^{(T_1)} = \Omega(1)$ for $i \in [n], r \in \mathcal{S}_i^{(0)}$.

(3) For the third claim, we prove by contradiction that there exists a sufficiently large gap $C_\epsilon > 0$ such that for noisy samples $i \in \mathcal{S}_f$, there exists a constant fraction that satisfies $\frac{1}{m}\sum_{r=1}^m \left(\overline{\rho}_{\tilde{y}_i, r, i}^{(t^*)} - \gamma_{-\tilde{y}_i, r}^{(t^*)}\right) \geq C_\epsilon$.

We prove this claim by contradiction. Suppose the claim does not hold. Then there must exist a constant fraction of samples such that $\frac{1}{m}\sum_{r=1}^{m}\left(\overline{\rho}_{\tilde{y}_i,r,i}^{(t^*)} - \gamma_{-\tilde{y}_i,r}^{(t^*)}\right) \leq C_\epsilon$. Formally, We denote the set of such samples as

$$\mathcal{I}' := \left\{ i \in \mathcal{S}_f : \frac{1}{m}\sum_{r=1}^{m}\left(\overline{\rho}_{\tilde{y}_i,r,i}^{(t^*)} - \gamma_{-\tilde{y}_i,r}^{(t^*)}\right) \leq C_\epsilon \right\}$$

with $|\mathcal{I}'| = \tau' n$ for some constant $\tau' > 0$ that satisfies $\tau' \leq \frac{\tau_+ + \tau_-}{2}$, i.e., upper bounded by the number of noisy samples in the dataset. Then we have

$$L_S(\mathbf{W}^{(t^*)}) = \frac{1}{n}\sum_{i=1}^{n}\ell(f(\mathbf{W}^{(t^*)},\mathbf{x}_i),\tilde{y}_i) \geq \frac{1}{n}\sum_{i\in\mathcal{I}'}\log(1+\exp(-\tilde{y}_i f(\mathbf{W}^{(t^*)},\mathbf{x}_i)))$$

$$\geq \frac{1}{n}\sum_{i\in\mathcal{I}'}\log\left(1+\exp\left(\frac{1}{m}\sum_{r=1}^{m}\left(\gamma_{-\tilde{y}_i,r}^{(t^*)} - \overline{\rho}_{\tilde{y}_i,r,i}^{(t^*)}\right) - 1/C_1\right)\right)$$

$$\geq \tau'\log(1+\exp(-C_\epsilon - 0.001)) > \tau'\log(2)$$

where we use Lemma C.4 in the second inequality. The third inequality is by the definition of $\mathcal{I}'$ and $C_1 \geq 1000$. Thus this raises a contradiction given that $L_S(\mathbf{W}^{(t^*)}) \leq 2\epsilon$ for any $\epsilon > 0$. This suggests there exists a constant fraction of noisy samples satisfy $\frac{1}{m}\sum_{r=1}^{m}\left(\overline{\rho}_{\tilde{y}_i,r,i}^{(t^*)} - \gamma_{-\tilde{y}_i,r}^{(t^*)}\right) \geq C_\epsilon$. This further indicates by Lemma C.4, for these samples

$$\tilde{y}_i f(\mathbf{W}^{(t^*)},\mathbf{x}_i) \geq C_\epsilon - 1/C_1 > 0.$$

(4) For the test error, we first derive the probability $\mathbb{P}(yf(\mathbf{W}^{(t^*)},\mathbf{x}) < 0)$ as

$$\mathbb{P}(yf(\mathbf{W}^{(T^*)},\mathbf{x}) < 0)$$

$$= \mathbb{P}\left(\sum_{r=1}^{m}\sigma(\langle\mathbf{w}_{-y,r}^{(t^*)},\boldsymbol{\xi}+\boldsymbol{\zeta}\rangle) - \sum_{r=1}^{m}\sigma(\langle\mathbf{w}_{y,r}^{(t^*)},\boldsymbol{\xi}+\boldsymbol{\zeta}\rangle) \geq \sum_{r=1}^{m}\sigma(\langle\mathbf{w}_{y,r}^{(t^*)},y\boldsymbol{\mu}\rangle) - \sum_{r=1}^{m}\sigma(\langle\mathbf{w}_{-y,r}^{(t^*)},y\boldsymbol{\mu}\rangle)\right)$$

$$\geq \mathbb{P}\left(\frac{1}{m}\sum_{r=1}^{m}\sigma(\langle\mathbf{w}_{-y,r}^{(t^*)},\boldsymbol{\xi}+\boldsymbol{\zeta}\rangle) - \frac{1}{m}\sum_{r=1}^{m}\sigma(\langle\mathbf{w}_{y,r}^{(t^*)},\boldsymbol{\xi}+\boldsymbol{\zeta}\rangle) \geq \frac{1}{m}\sum_{r=1}^{m}\gamma_{y,r}^{(t^*)} + 1/C'\right)$$

$$= \frac{1}{n}\sum_{i=1}^{n}\mathbb{P}\left(\frac{1}{m}\sum_{r=1}^{m}\sigma(\langle\mathbf{w}_{-y,r}^{(t^*)},\boldsymbol{\xi}_i+\boldsymbol{\zeta}\rangle) - \frac{1}{m}\sum_{r=1}^{m}\sigma(\langle\mathbf{w}_{y,r}^{(t^*)},\boldsymbol{\xi}_i+\boldsymbol{\zeta}\rangle) \geq \frac{1}{m}\sum_{r=1}^{m}\gamma_{y,r}^{(t^*)} + 1/C'\right) \tag{11}$$

for some sufficiently large constant $C' > 0$ and the second equality is by uniform distribution of $\boldsymbol{\xi}$.

Next, we consider the following two cases separately, i.e., (a) When $y = 1$ and (b) when $y = -1$. When $y = 1$, (11) can be further bounded as

$$\mathbb{P}(yf(\mathbf{W}^{(t^*)},\mathbf{x}) < 0)$$

$$\geq 0.5\tau_+\mathbb{P}\left(\frac{1}{m}\sum_{r=1}^{m}\sigma(\langle\mathbf{w}_{-1,r}^{(t^*)},\boldsymbol{\xi}_{i:i\in\mathcal{S}_f\cap\mathcal{S}_1}+\boldsymbol{\zeta}\rangle) - \frac{1}{m}\sum_{r=1}^{m}\sigma(\langle\mathbf{w}_{1,r}^{(t^*)},\boldsymbol{\xi}_{i:i\in\mathcal{S}_f\cap\mathcal{S}_1}+\boldsymbol{\zeta}\rangle) \geq \frac{1}{m}\sum_{r=1}^{m}\gamma_{1,r}^{(t^*)} + 1/C'\right)$$

where we use the sample size of $|\mathcal{S}_f \cap \mathcal{S}_1| = \frac{\tau_+ n}{2}$.

Now we analyze the the magnitude of each term. Based on the decomposition, we obtain for any $i \in \mathcal{S}_f$, $j = \pm 1$ and any $r \in [m]$

$$\langle\mathbf{w}_{j,r}^{(t^*)},\boldsymbol{\xi}_i+\boldsymbol{\zeta}\rangle = \left\langle\mathbf{w}_{j,r}^{(0)} - \gamma_{j,r}^{(t^*)}\|\boldsymbol{\mu}\|_2^{-2}\boldsymbol{\mu} + \sum_{i'=1}^{n}\overline{\rho}_{j,r,i}^{(t^*)}\|\boldsymbol{\xi}_{i'}\|_2^{-2}\boldsymbol{\xi}_{i'} + \sum_{i'=1}^{n}\underline{\rho}_{j,r,i}^{(t^*)}\|\boldsymbol{\xi}_{i'}\|_2^{-2}\boldsymbol{\xi}_{i'},\boldsymbol{\xi}_i+\boldsymbol{\zeta}\right\rangle$$

$$= \overline{\rho}_{j,r,i}^{(t^*)} + \underline{\rho}_{j,r,i}^{(t^*)} + \langle\mathbf{w}_{j,r}^{(0)},\boldsymbol{\xi}_i+\boldsymbol{\zeta}\rangle - \langle\gamma_{j,r}^{(t^*)}\|\boldsymbol{\mu}\|_2^{-2}\boldsymbol{\mu},\boldsymbol{\xi}_i+\boldsymbol{\zeta}\rangle$$

$$+ \sum_{i'\neq i}\overline{\rho}_{j,r,i'}^{(t^*)}\frac{\langle\boldsymbol{\xi}_{i'},\boldsymbol{\xi}_i+\boldsymbol{\zeta}\rangle}{\|\boldsymbol{\xi}_{i'}\|_2^2} + \sum_{i'\neq i}\underline{\rho}_{j,r,i'}^{(t^*)}\frac{\langle\boldsymbol{\xi}_{i'},\boldsymbol{\xi}_i+\boldsymbol{\zeta}\rangle}{\|\boldsymbol{\xi}_{i'}\|_2^2}$$

Then we can bound particularly for $i \in \mathcal{S}_f \cap \mathcal{S}_1$, i.e., $\tilde{y}_i = -1$

$$\langle \mathbf{w}_{-1,r}^{(t^*)}, \boldsymbol{\xi}_i + \boldsymbol{\zeta} \rangle \geq \overline{\rho}_{-1,r,i}^{(t^*)} - \widetilde{O}(\sigma_0 \sigma_\xi \sqrt{d}) - \widetilde{O}(\|\boldsymbol{\mu}\|_2^{-1} \sigma_\xi) - \widetilde{O}(nd^{-1/2})$$
$$\geq \overline{\rho}_{-1,r,i}^{(t^*)} - 1/C_3$$

where we use Lemma B.1, B.2 and the upper bound on $\overline{\rho}_{j,r,i}^{(t^*)}, \gamma_{j,r}^{(t^*)} = \widetilde{O}(1)$ for the first inequality. The second inequality is by Condition 4.1 on $\|\boldsymbol{\mu}\|_2$ and $d$ for some sufficiently large constant $C_3$.

In addition, we can similarly show

$$\langle \mathbf{w}_{1,r}^{(t^*)}, \boldsymbol{\xi}_i + \boldsymbol{\zeta} \rangle \leq \underline{\rho}_{1,r,i}^{(t^*)} + \widetilde{O}(\sigma_0 \sigma_\xi \sqrt{d}) + \widetilde{O}(\|\boldsymbol{\mu}\|_2^{-1} \sigma_\xi) + \widetilde{O}(nd^{-1/2}) \leq 1/C_3$$

Then can show for any $i \in \mathcal{S}_f \cap \mathcal{S}_1$, i.e., $\tilde{y}_i = -1$

$$\frac{1}{m} \sum_{r=1}^m \sigma(\langle \mathbf{w}_{-1,r}^{(t^*)}, \boldsymbol{\xi}_i + \boldsymbol{\zeta} \rangle) - \frac{1}{m} \sum_{r=1}^m \sigma(\langle \mathbf{w}_{1,r}^{(t^*)}, \boldsymbol{\xi}_i + \boldsymbol{\zeta} \rangle) \geq \frac{1}{m} \sum_{r=1}^m \overline{\rho}_{-1,r,i}^{(t^*)} - 2/C_3$$
$$\geq \frac{1}{m} \sum_{r=1}^m \gamma_{1,r}^{(t^*)} + C_\epsilon - 2/C_3$$
$$> \frac{1}{m} \sum_{r=1}^m \gamma_{1,r}^{(t^*)} + 1/C'$$

where we choose $C_3, C'$ such that $C_\epsilon - 2/C_3 > 1/C'$. This suggests when $y = 1$, we have

$$\mathbb{P}(yf(\mathbf{W}^{(T^*)}, \mathbf{x}) < 0) \geq 0.5\tau_+$$

Similarly, we use the same argument to show when $y = -1$,

$$\mathbb{P}(yf(\mathbf{W}^{(T^*)}, \mathbf{x}) < 0) \geq 0.5\tau_-.$$

This completes the proof that $\mathbb{P}(yf(\mathbf{W}^{(T^*)}, \mathbf{x}) < 0) \geq 0.5 \min\{\tau_-, \tau_+\}$. $\qquad \square$

## D. Analysis without Label Noise

For the case of no label noise, i.e., $\tau_+, \tau_- = 0$. We still require the same assumption as in Condition 4.1. We reiterate the assumption for completeness here.

**Condition D.1.** We let $T^* = \widetilde{\Theta}(\eta^{-1} \epsilon^{-1} nm\sigma_\xi^{-2} d^{-1})$ to be the maximum number of iterations considered. Suppose that there exists a sufficiently large constant $C$ such that the following hold:

1. The signal-to-noise ratio is bounded by constants $n \cdot \mathrm{SNR}^2 = \Theta(1)$.

2. The dimension $d$ is sufficiently large, $d \geq C \max \left\{ n^2 \log(nm/\delta) \log(T^*)^2, n\|\boldsymbol{\mu}\|_2 \sigma_\xi^{-1} \sqrt{\log(n/\delta)} \right\}$.

3. The standard deviation of the Gaussian initialization $\sigma_0$ is chosen such that $\sigma_0 \leq C^{-1} \min \left\{ \sqrt{n}\sigma_\xi^{-1} d^{-1}, \|\boldsymbol{\mu}\|_2^{-1} \log(m/\delta)^{-1/2} \right\}$.

4. The size of training sample $n$ and width $m$ adhere to $m \geq C \log(n/\delta), n \geq C \log(m/\delta)$.

5. The signal strength satisfies $\|\boldsymbol{\mu}\|_2^2 \geq C\sigma_\xi^2 \log(n/\delta)$.

6. The learning rate $\eta$ satisfies $\eta \leq C^{-1} \min \left\{ \sigma_\xi^{-2} d^{-3/2} n^2 m \sqrt{\log(n/\delta)}, \sigma_\xi^{-2} d^{-1} n \right\}$.

With the label noise, the coefficient update equations are given by

$$\gamma_{j,r}^{(0)}, \overline{\rho}_{j,r,i}^{(0)}, \underline{\rho}_{j,r,i}^{(0)} = 0,$$

$$\gamma_{j,r}^{(t+1)} = \gamma_{j,r}^{(t)} - \frac{\eta}{nm} \sum_{i=1}^{n} \ell_i'^{(t)} \sigma'(\langle \mathbf{w}_{j,r}^{(t)}, y_i \boldsymbol{\mu} \rangle) \|\boldsymbol{\mu}\|_2^2,$$

$$\overline{\rho}_{j,r,i}^{(t+1)} = \overline{\rho}_{j,r,i}^{(t)} - \frac{\eta}{nm} \ell_i'^{(t)} \sigma'(\langle \mathbf{w}_{j,r}^{(t)}, \boldsymbol{\xi}_i \rangle) \|\boldsymbol{\xi}_i\|_2^2 \mathbb{1}(y_i = j),$$

$$\underline{\rho}_{j,r,i}^{(t+1)} = \underline{\rho}_{j,r,i}^{(t)} + \frac{\eta}{nm} \ell_i'^{(t)} \sigma'(\langle \mathbf{w}_{j,r}^{(t)}, \boldsymbol{\xi}_i \rangle) \|\boldsymbol{\xi}_i\|_2^2 \mathbb{1}(y_i = -j).$$

where we highlight that for all $i \in [n]$, $\tilde{y}_i = y_i$.

**Proposition D.1.** *Under Assumption 4.1 and the same definition as for the label noise case, for $0 \leq t \leq T^*$, we have*

$$0 \leq \overline{\rho}_{j,r,i}^{(t)} \leq \alpha, \tag{12}$$

$$0 \geq \underline{\rho}_{j,r,i}^{(t)} \geq -\beta - 10\sqrt{\frac{\log(6n^2/\delta)}{d}} n\alpha \geq -\alpha, \tag{13}$$

$$0 \leq \gamma_{j,r}^{(t)} \leq C_\gamma \alpha \tag{14}$$

In order to prove such results, we use the same induction strategy as for the label noise case. We first notice that if (12), (13), (14) hold at iteration $t$, then bounds in Lemma C.3 and Lemma C.4 hold at iteration $t$. We include the results here for the purpose of completeness.

**Lemma D.2.** *Under Condition D.1, suppose (12), (13), (14) hold at iteration t,*

$$|\langle \mathbf{w}_{j,r}^{(t)} - \mathbf{w}_{j,r}^{(0)}, \boldsymbol{\mu} \rangle - j \cdot \gamma_{j,r}^{(t)}| \leq \text{SNR}\sqrt{\frac{8\log(6n/\delta)}{d}} n\alpha,$$

$$|\langle \mathbf{w}_{j,r}^{(t)} - \mathbf{w}_{j,r}^{(0)}, \boldsymbol{\xi}_i \rangle - \overline{\rho}_{j,r,i}^{(t)}| \leq 5\sqrt{\frac{\log(6n^2/\delta)}{d}} n\alpha, \quad y_i = j$$

$$|\langle \mathbf{w}_{j,r}^{(t)} - \mathbf{w}_{j,r}^{(0)}, \boldsymbol{\xi}_i \rangle - \underline{\rho}_{j,r,i}^{(t)}| \leq 5\sqrt{\frac{\log(6n^2/\delta)}{d}} n\alpha, \quad y_i = -j$$

*for all $r \in [m], j = \pm 1, i \in [n]$. Further, there exists a sufficiently large constant $C_1$ such that*

$$\frac{1}{m} \sum_{r=1}^{m} (\gamma_{y_i,r}^{(t)} + \overline{\rho}_{y_i,r,i}^{(t)}) - 1/C_1 \leq y_i f(\mathbf{W}^{(t)}, \mathbf{x}_i) \leq \frac{1}{m} \sum_{r=1}^{m} (\gamma_{y_i,r}^{(t)} + \overline{\rho}_{y_i,r,i}^{(t)}) + 1/C_1$$

*for all $i \in [n]$.*

*Proof of Lemma D.2.* The proof follows directly from Lemma C.3 and Lemma C.4. $\square$

Next we prove a stronger lemma that only holds under the condition $n \cdot \text{SNR}^2 = \Theta(1)$ and without the presence of label noise.

First we require a lemma that allows us to bound the loss derivative ratios.

**Lemma D.3** ((Kou et al., 2023)). *Let $g(z) = -1/(1 + \exp(z))$, then for all $z_2 - c \geq z_1 \geq -1$, for $c \geq 0$, we have*

$$\frac{\exp(c)}{4} \leq \frac{g(z_1)}{g(z_2)} \leq \exp(c)$$

**Lemma D.4.** *Under Condition D.1, and for any given $t \leq T^*$, suppose (12), (13), (14) hold for all iterations $s \leq t$. Then we can prove for some constant $\kappa \geq 0$*

*(1) $\frac{1}{m} \sum_{r=1}^{m} (\overline{\rho}_{y_i,r,i}^{(s)} + \gamma_{y_i,r}^{(s)} - \overline{\rho}_{y_k,r,k}^{(s)} - \gamma_{y_k,r}^{(s)}) \leq \kappa$ for all $i, k \in [n]$.*

*(2) $\ell_i'^{(s)} / \ell_k'^{(s)} \leq \tilde{C}_\ell$ for all $i, k \in [n]$.*

*(3) $\mathcal{S}_i^{(0)} \subseteq \mathcal{S}_i^{(s)}, \mathcal{S}_{j,r}^{(0)} \subseteq \mathcal{S}_{j,r}^{(s)}$, for all $i \in [n]$ and $j = \pm 1, r \in [m]$.*

*Proof of Lemma D.4.* We prove the results by induction. It is clear at $s = 0$, claims (1) and (3) are satisfied trivially. Then for claim (2), we use Lemma D.3 to bound

$$\frac{\ell_i'^{(0)}}{\ell_k'^{(0)}} \le \exp(y_k f(\mathbf{W}^{(0)}, \mathbf{x}_k) - y_i f(\mathbf{W}^{(0)}, \mathbf{x}_i))$$

$$\le \exp\Big(\frac{1}{m}\sum_{r=1}^m \big(\overline{\rho}_{y_k,r,k}^{(0)} + \gamma_{y_k,r}^{(0)}\big) - \frac{1}{m}\sum_{r=1}^m \big(\overline{\rho}_{y_i,r,i}^{(0)} + \gamma_{y_i,r}^{(0)}\big) + 2/C_1\Big)$$

$$= \exp(2/C_1),$$

which shows a constant upper bound.

Next suppose at $t = \tilde{t}$, (1)-(3) hold for all $s \le \tilde{t} - 1$, then we show they also hold at $\tilde{t}$. For (1), according to the update rule of the coefficients,

$$\frac{1}{m}\sum_{r=1}^m \big(\overline{\rho}_{y_i,r,i}^{(\tilde{t})} - \overline{\rho}_{y_k,r,k}^{(\tilde{t})}\big) = \frac{1}{m}\sum_{r=1}^m \big(\overline{\rho}_{y_i,r,i}^{(\tilde{t}-1)} - \overline{\rho}_{y_k,r,k}^{(\tilde{t}-1)}\big)$$

$$- \frac{\eta}{nm^2}\Bigg(\sum_{r\in\mathcal{S}_i^{(\tilde{t}-1)}} \ell_i'^{(\tilde{t}-1)}\|\boldsymbol{\xi}_i\|_2^2 - \sum_{r\in\mathcal{S}_k^{(\tilde{t}-1)}} \ell_k'^{(\tilde{t}-1)}\|\boldsymbol{\xi}_k\|_2^2\Bigg) \tag{15}$$

$$\frac{1}{m}\sum_{r=1}^m \big(\gamma_{y_i,r}^{(\tilde{t})} - \gamma_{y_k,r}^{(\tilde{t})}\big) = \frac{1}{m}\sum_{r=1}^m \big(\gamma_{y_i,r}^{(\tilde{t}-1)} - \gamma_{y_k,r}^{(\tilde{t}-1)}\big)$$

$$- \frac{\eta\|\boldsymbol{\mu}\|_2^2}{nm^2}\sum_{r=1}^m \underbrace{\Bigg(\sum_{i'=1}^n \ell_{i'}'^{(t-1)}\mathbb{1}(\langle\mathbf{w}_{y_i,r}^{(\tilde{t}-1)}, y_{i'}\boldsymbol{\mu}\rangle) - \sum_{i'=1}^n \ell_{i'}'^{(t-1)}\mathbb{1}(\langle\mathbf{w}_{y_k,r}^{(\tilde{t}-1)}, y_{i'}\boldsymbol{\mu}\rangle)\Bigg)}_{A_5} \tag{16}$$

We first analyze $A_5$ depending on the following four cases.

- When $\langle\mathbf{w}_{y_i,r}^{(\tilde{t}-1)}, \boldsymbol{\mu}\rangle \ge 0, \langle\mathbf{w}_{y_k,r}^{(\tilde{t}-1)}, \boldsymbol{\mu}\rangle \ge 0$, we have $A_5 = \sum_{i'\in\mathcal{S}_1} \ell_{i'}'^{(t-1)} - \sum_{i'\in\mathcal{S}_1} \ell_{i'}'^{(t-1)} = 0$.

- When $\langle\mathbf{w}_{y_i,r}^{(\tilde{t}-1)}, \boldsymbol{\mu}\rangle \le 0, \langle\mathbf{w}_{y_k,r}^{(\tilde{t}-1)}, \boldsymbol{\mu}\rangle \le 0$, we have $A_5 = \sum_{i'\in\mathcal{S}_{-1}} \ell_{i'}'^{(t-1)} - \sum_{i'\in\mathcal{S}_{-1}} \ell_{i'}'^{(t-1)} = 0$.

- When $\langle\mathbf{w}_{y_i,r}^{(\tilde{t}-1)}, \boldsymbol{\mu}\rangle \ge 0, \langle\mathbf{w}_{y_k,r}^{(\tilde{t}-1)}, \boldsymbol{\mu}\rangle \le 0$, we have $A_5 = \sum_{i'\in\mathcal{S}_1} \ell_{i'}'^{(t-1)} - \sum_{i'\in\mathcal{S}_{-1}} \ell_{i'}'^{(t-1)}$.

- When $\langle\mathbf{w}_{y_i,r}^{(\tilde{t}-1)}, \boldsymbol{\mu}\rangle \le 0, \langle\mathbf{w}_{y_k,r}^{(\tilde{t}-1)}, \boldsymbol{\mu}\rangle \ge 0$, we have $A_5 = \sum_{i'\in\mathcal{S}_{-1}} \ell_{i'}'^{(t-1)} - \sum_{i'\in\mathcal{S}_1} \ell_{i'}'^{(t-1)}$.

Now we would like to bound the combination of (15) and (16).

When $\frac{1}{m}\sum_{r=1}^m \big(\overline{\rho}_{y_i,r,i}^{(\tilde{t}-1)} + \gamma_{y_i,r}^{(\tilde{t}-1)} - \overline{\rho}_{y_k,r,k}^{(\tilde{t}-1)} - \gamma_{y_k,r}^{(\tilde{t}-1)}\big) \le 0.5\kappa$, then (15) can be bounded as

$$\frac{1}{m}\sum_{r=1}^m \big(\overline{\rho}_{y_i,r,i}^{(\tilde{t})} + \gamma_{y_i,r}^{(\tilde{t})} - \overline{\rho}_{y_k,r,k}^{(\tilde{t})} - \gamma_{y_k,r}^{(\tilde{t})}\big)$$

$$\le \frac{1}{m}\sum_{r=1}^m \big(\overline{\rho}_{y_i,r,i}^{(\tilde{t}-1)} + \gamma_{y_i,r}^{(\tilde{t}-1)} - \overline{\rho}_{y_k,r,k}^{(\tilde{t}-1)} - \gamma_{y_k,r}^{(\tilde{t}-1)}\big) - \frac{\eta}{nm^2}\Bigg(\sum_{r\in\mathcal{S}_i^{(\tilde{t}-1)}} \ell_i'^{(\tilde{t}-1)}\|\boldsymbol{\xi}_i\|_2^2 - \sum_{r\in\mathcal{S}_k^{(\tilde{t}-1)}} \ell_k'^{(\tilde{t}-1)}\|\boldsymbol{\xi}_k\|_2^2\Bigg)$$

$$- \frac{\eta\|\boldsymbol{\mu}\|_2^2}{nm^2}\sum_{r=1}^m \Bigg(\sum_{i'=1}^n \ell_{i'}'^{(t-1)}\mathbb{1}(\langle\mathbf{w}_{y_i,r}^{(\tilde{t}-1)}, y_{i'}\boldsymbol{\mu}\rangle) - \sum_{i'=1}^n \ell_{i'}'^{(t-1)}\mathbb{1}(\langle\mathbf{w}_{y_k,r}^{(\tilde{t}-1)}, y_{i'}\boldsymbol{\mu}\rangle)\Bigg)$$

$$\le 0.5\kappa - \frac{\eta}{nm}|\mathcal{S}_i^{(\tilde{t}-1)}|\ell_i'^{(\tilde{t}-1)}\|\boldsymbol{\xi}_i\|_2^2 - \frac{\eta}{nm}\sum_{i'=1}^n \ell_i'^{(\tilde{t})}\|\boldsymbol{\mu}\|_2^2$$

$$\leq 0.5\kappa + 1.01\frac{\eta\sigma_\xi^2 d}{n} + \frac{\eta\|\boldsymbol{\mu}\|_2^2}{m}$$

$$\leq \kappa$$

where the second inequality is by $\ell_i'^{(t)} \leq 0$ for all $i, t$. The third inequality is by $|\mathcal{S}_i^{(\tilde{t}-1)}| \leq m$, $|\ell_i'^{(\tilde{t}-1)}| \leq 1$ and Lemma C.7. The last inequality us by Condition D.1 for sufficiently small stepsize $\eta$.

When $\frac{1}{m}\sum_{r=1}^m \big(\overline{\rho}_{y_i,r,i}^{(\tilde{t}-1)} + \gamma_{y_i,r}^{(\tilde{t}-1)} - \overline{\rho}_{y_k,r,k}^{(\tilde{t}-1)} - \gamma_{y_k,r}^{(\tilde{t}-1)}\big) \geq 0.5\kappa$, then by Lemma D.2,

$$y_i f(\mathbf{W}^{(\tilde{t}-1)}, \mathbf{x}_i) - y_k f(\mathbf{W}^{(\tilde{t}-1)}, \mathbf{x}_k) \geq \frac{1}{m}\sum_{r=1}^m \big(\gamma_{y_i,r}^{(\tilde{t}-1)} + \overline{\rho}_{y_i,r,i}^{(\tilde{t}-1)} - \gamma_{y_k,r}^{(\tilde{t}-1)} - \overline{\rho}_{y_k,r,k}^{(\tilde{t}-1)}\big) - 2/C_1$$

$$\geq 0.5\kappa - 2/C_1$$

$$\geq 0.4\kappa$$

where we choose $C_1 \geq 20/\kappa$. Then by Lemma D.3

$$\frac{\ell_i'^{(\tilde{t}-1)}}{\ell_k'^{(\tilde{t}-1)}} \leq \exp(y_k f(\mathbf{W}^{(\tilde{t}-1)}, \mathbf{x}_k) - y_i f(\mathbf{W}^{(\tilde{t}-1)}, \mathbf{x}_i)) \leq \exp(-0.4\kappa).$$

Then we can show

$$\frac{|\mathcal{S}_i^{(\tilde{t}-1)}| \cdot |\ell_i'^{(\tilde{t}-1)}| \cdot \|\boldsymbol{\xi}_i\|_2^2}{|\mathcal{S}_k^{(\tilde{t}-1)}| \cdot |\ell_k'^{(\tilde{t}-1)}| \cdot \|\boldsymbol{\xi}_k\|_2^2} \leq 1.01 \cdot \exp(-0.4\kappa) \tag{17}$$

where we use Lemma C.7 and C.8 by choosing sufficiently large $d$ and $m$.

Then we obtain

$$\frac{1}{m}\sum_{r=1}^m \big(\overline{\rho}_{y_i,r,i}^{(\tilde{t})} + \gamma_{y_i,r}^{(\tilde{t})} - \overline{\rho}_{y_k,r,k}^{(\tilde{t})} - \gamma_{y_k,r}^{(\tilde{t})}\big)$$

$$\leq \frac{1}{m}\sum_{r=1}^m \big(\overline{\rho}_{y_i,r,i}^{(\tilde{t}-1)} + \gamma_{y_i,r}^{(\tilde{t}-1)} - \overline{\rho}_{y_k,r,k}^{(\tilde{t}-1)} - \gamma_{y_k,r}^{(\tilde{t}-1)}\big) - \frac{\eta}{nm^2}\Big(|\mathcal{S}_i^{(\tilde{t}-1)}|\ell_i'^{(\tilde{t}-1)}\|\boldsymbol{\xi}_i\|_2^2 - |\mathcal{S}_k^{(\tilde{t}-1)}|\ell_k'^{(\tilde{t}-1)}\|\boldsymbol{\xi}_k\|_2^2\Big)$$

$$\quad - \frac{\eta}{nm}\bigg(\sum_{i'\in\mathcal{S}_{\pm 1}} \ell_{i'}'^{(\tilde{t}-1)} - \sum_{i'\in\mathcal{S}_{\mp 1}} \ell_{i'}'^{(\tilde{t}-1)}\bigg)\|\boldsymbol{\mu}\|_2^2$$

$$\leq \frac{1}{m}\sum_{r=1}^m \big(\overline{\rho}_{y_i,r,i}^{(\tilde{t}-1)} + \gamma_{y_i,r}^{(\tilde{t}-1)} - \overline{\rho}_{y_k,r,k}^{(\tilde{t}-1)} - \gamma_{y_k,r}^{(\tilde{t}-1)}\big) + \frac{\eta}{nm^2}\big(1.01\exp(-0.4\kappa) - 1\big)|\mathcal{S}_k^{(\tilde{t}-1)}||\ell_k'^{(\tilde{t}-1)}| \cdot \|\boldsymbol{\xi}_i\|_2^2$$

$$\quad + \frac{\eta}{2m}(\tilde{C}_\ell - 1)\min_{i\in[n]}|\ell_i'^{(\tilde{t}-1)}|\|\boldsymbol{\mu}\|_2^2$$

$$\leq \frac{1}{m}\sum_{r=1}^m \big(\overline{\rho}_{y_i,r,i}^{(\tilde{t}-1)} + \gamma_{y_i,r}^{(\tilde{t}-1)} - \overline{\rho}_{y_k,r,k}^{(\tilde{t}-1)} - \gamma_{y_k,r}^{(\tilde{t}-1)}\big) + \frac{\eta}{nm}\Big(\big(0.49\exp(-0.4\kappa) - 0.48\big)|\ell_k'^{(\tilde{t}-1)}|\sigma_\xi^2 d$$

$$\quad + 0.5n(\tilde{C}_\ell - 1)\tilde{C}_\ell|\ell_k'^{(\tilde{t}-1)}|\|\boldsymbol{\mu}\|_2^2\Big)$$

$$\leq \frac{1}{m}\sum_{r=1}^m \big(\overline{\rho}_{y_i,r,i}^{(\tilde{t}-1)} + \gamma_{y_i,r}^{(\tilde{t}-1)} - \overline{\rho}_{y_k,r,k}^{(\tilde{t}-1)} - \gamma_{y_k,r}^{(\tilde{t}-1)}\big)$$

$$\leq \kappa$$

where the second inequality is by applying (17) and also $\sum_{i'\in\mathcal{S}_{\pm 1}} \ell_{i'}'^{(\tilde{t}-1)} - \sum_{i'\in\mathcal{S}_{\mp 1}} \ell_{i'}'^{(\tilde{t}-1)} \leq \frac{n}{2}(\max_{i\in[n]}|\ell_i'^{(\tilde{t}-1)}| - \min_{i\in[n]}|\ell_i'^{(\tilde{t}-1)}|) \leq \frac{n}{2}(\tilde{C}_\ell - 1)\min_{i\in[n]}|\ell_i'^{(\tilde{t}-1)}|)$ by induction. The third inequality is by $\kappa \geq 1$ and Lemma C.7, Lemma C.8. The fourth inequality follows from the conditions on $n \cdot \text{SNR}^2 \leq \frac{2(0.48 - 0.49\exp(-0.4\kappa))}{(\tilde{C}_\ell - 1)\tilde{C}_\ell} = O(1)$. This verifies the claim (1) for $t = \tilde{t}$.

Now by Lemma D.3 and Lemma D.2, we can show

$$\frac{\ell_i'^{(\tilde{t})}}{\ell_k'^{(\tilde{t})}} \leq \exp(y_k f(\mathbf{W}^{(\tilde{t})}, \mathbf{x}_k) - y_i f(\mathbf{W}^{(\tilde{t})}, \mathbf{x}_i))$$

$$\leq \exp\left(\frac{1}{m} \sum_{r=1}^{m} \left(\overline{\rho}_{y_k,r,i}^{(\tilde{t})} + \gamma_{y_k,r}^{(\tilde{t})} - \overline{\rho}_{y_i,r,k}^{(\tilde{t})} - \gamma_{y_i,r}^{(\tilde{t})}\right) + 2/C_1\right)$$

$$\leq \exp(\kappa + 2/C_1)$$

Hence for a given $\kappa$, we can take $\tilde{C}_\ell = \exp(\kappa + 2/C_1)$. This verifies that claim (2) is satisfied.

To verify the claim (3), we can show

$$\langle \mathbf{w}_{\tilde{y}_i,r}^{(\tilde{t})}, \boldsymbol{\xi}_i \rangle = \langle \mathbf{w}_{\tilde{y}_i,r}^{(\tilde{t}-1)}, \boldsymbol{\xi}_i \rangle - \frac{\eta}{nm} \sum_{i'=1}^{n} \ell_{i'}'^{(\tilde{t}-1)} \cdot \sigma'(\langle \mathbf{w}_{\tilde{y}_i,r}^{(\tilde{t}-1)}, \boldsymbol{\xi}_{i'} \rangle) \cdot \langle \boldsymbol{\xi}_i, \boldsymbol{\xi}_{i'} \rangle$$

$$- \frac{\eta}{nm} \sum_{i'=1}^{n} \ell_{i'}'^{(\tilde{t}-1)} \cdot \sigma'(\langle \mathbf{w}_{\tilde{y}_i,r}^{(\tilde{t}-1)}, y_{i'}\boldsymbol{\mu} \rangle) \cdot \langle y_{i'}\boldsymbol{\mu}, \boldsymbol{\xi}_i \rangle$$

$$= \langle \mathbf{w}_{\tilde{y}_i,r}^{(\tilde{t}-1)}, \boldsymbol{\xi}_i \rangle \underbrace{- \frac{\eta}{nm} \ell_i'^{(\tilde{t}-1)} \|\boldsymbol{\xi}_i\|_2^2}_{A_6} \underbrace{- \frac{\eta}{nm} \sum_{i'\neq i} \ell_{i'}'^{(\tilde{t}-1)} \sigma'(\langle \mathbf{w}_{\tilde{y}_i,r}^{(\tilde{t}-1)}, \boldsymbol{\xi}_{i'} \rangle) \cdot \langle \boldsymbol{\xi}_i, \boldsymbol{\xi}_{i'} \rangle}_{A_7}$$

$$\underbrace{- \frac{\eta}{nm} \sum_{i'=1}^{n} \ell_{i'}'^{(\tilde{t}-1)} \cdot \sigma'(\langle \mathbf{w}_{\tilde{y}_i,r}^{(\tilde{t}-1)}, y_{i'}\boldsymbol{\mu} \rangle) \cdot \langle y_{i'}\boldsymbol{\mu}, \boldsymbol{\xi}_i \rangle}_{A_8}.$$

We respectively bound the three terms as

$$A_6 \geq 0.99 \frac{\sigma_\xi^2 d\eta}{nm} |\ell_i'^{(\tilde{t}-1)}|,$$

where we use C.7. For $A_7$, we can bound

$$|A_7| \leq 2n\tilde{C}_\ell |\ell_i'^{(\tilde{t}-1)}| \sigma_\xi^2 \sqrt{d \log(6n^2/\delta)},$$

where we use Lemma B.1 and claim (2). Similarly,

$$|A_8| \leq n\tilde{C}_\ell |\ell_i'^{(\tilde{t}-1)}| \|\boldsymbol{\mu}\|_2 \sigma_\xi \sqrt{2 \log(6n/\delta)}$$

where we use Lemma B.1 and claim (2). By the Condition D.1 where $d \geq C \max\{\tilde{C}_\ell^2 n^2 \log(6n^2/\delta), \tilde{C}_\ell n \|\boldsymbol{\mu}\|_2 \sigma_\xi^{-1} \sqrt{2 \log(6n/\delta)}\}$ for sufficiently large $C$. This ensures $A_6 \geq |A_7| + |A_8|$, which leads to $\langle \mathbf{w}_{\tilde{y}_i,r}^{(\tilde{t})}, \boldsymbol{\xi}_i \rangle \geq \langle \mathbf{w}_{\tilde{y}_i,r}^{(\tilde{t}-1)}, \boldsymbol{\xi}_i \rangle > 0$ and thus $\mathcal{S}_i^{(0)} \subseteq \mathcal{S}_i^{(\tilde{t}-1)} \subseteq \mathcal{S}_i^{(\tilde{t})}$. Similarly, we can use the same argument to prove $\mathcal{S}_{j,r}^{(0)} \subseteq \mathcal{S}_{j,r}^{(\tilde{t}-1)} \subseteq \mathcal{S}_{j,r}^{(\tilde{t})}$. $\square$

*Proof of Proposition D.1.* We prove the results by induction. At $t = 0$, it is clear that the claims hold trivially. Suppose there exists $\tilde{T} \leq T^*$ such that the the claims hold for all $t \leq \tilde{T} - 1$. We aim to show they also hold at $t = \tilde{T}$. By Lemma D.4, we know for all $t \leq \tilde{T} - 1$, we have $\ell_i'^{(t)}/\ell_k'^{(t)} \leq \tilde{C}_\ell$ for all $i, k \in [n]$ and $\mathcal{S}_i^{(0)} \subseteq \mathcal{S}_i^{(t)}, \mathcal{S}_{j,r}^{(0)} \subseteq \mathcal{S}_{j,r}^{(t)}$.

First, we follow the same proof strategy to show $0 \geq \underline{\rho}_{j,r,i}^{(\tilde{T})} \geq -\beta - 10\sqrt{\log(6n^2/\delta)/d}$.

Next we show the upper bound for $\overline{\rho}_{j,r,i}^{(\tilde{T})}$. Let $t_{r,i}$ be the last time $t < T^*$ such that $\overline{\rho}_{j,r,i}^{(t)} \leq 0.5\alpha$. Then

$$\overline{\rho}_{y_i,r,i}^{(\tilde{T})} = \overline{\rho}_{y_i,r,i}^{(t_{r,i})} - \frac{\eta}{nm} \ell_i'^{(t_{r,i})} \mathbb{1}(\langle \mathbf{w}_{y_i,r}^{(t_{r,i})}, \boldsymbol{\xi}_i \rangle \geq 0) \|\boldsymbol{\xi}_i\|_2^2 - \sum_{t\in(t_{r,i},\tilde{T})} \frac{\eta}{nm} \ell_i'^{(t)} \mathbb{1}(\langle \mathbf{w}_{y_i,r}^{(t)}, \boldsymbol{\xi}_i \rangle \geq 0) \|\boldsymbol{\xi}_i\|_2^2$$

$$\leq 0.5\alpha + 1.01\frac{\eta\sigma_\xi^2 d}{nm} + 1.01\frac{\eta\sigma_\xi^2 d}{nm}\sum_{t\in(t_{r,i},\widetilde{T})}\frac{1}{1+\exp(\frac{1}{m}\sum_{r=1}^m(\overline\rho_{y_i,r,i}^{(t)}+\gamma_{y_i,r}^{(t)})-1/C_1)}$$

$$\leq 0.75\alpha + 2.02\frac{\eta\sigma_\xi^2 d}{nm}$$

$$\leq \alpha$$

where the first inequality is by Lemma C.7, Lemma D.2 and $|\ell_i'^{(t_{r,i})}| \leq 1$. The second and third inequality is by choosing $\eta \leq C^{-1}nm\sigma_\xi^{-2}d^{-1}$ for sufficiently large $C$.

Then we proceed to show upper bound for $\gamma_{j,r}^{(\widetilde{T})}$. From the update rule, it is clear that $\gamma_{j,r}^{(\widetilde{T})} \geq \gamma_{j,r}^{(\widetilde{T}-1)} \geq 0$. To prove the upper bound on $\gamma_{j,r}^{(\widetilde{T})}$, we aim to show there exists $i^* \in [n]$ such that for all $t \leq T^*$ that

$$\frac{\gamma_{j,r}^{(t)}}{\overline\rho_{y_{i^*},r,i^*}^{(t)}} \leq C_\gamma n \cdot \mathrm{SNR}^2.$$

where we take $C_\gamma = 1.1\tilde{C}_\ell$. We prove the claim by induction. We first lower bound $\overline\rho_{y_i,r,i}^{(\widetilde{T})}$. In particular, we can lower bound for any $i^* \in [n], r \in \mathcal{S}_i^{(0)}$ that

$$\overline\rho_{y_{i^*},r,i^*}^{(\widetilde{T})} = \overline\rho_{y_{i^*},r,i^*}^{(\widetilde{T}-1)} - \frac{\eta}{nm}\ell_{i^*}'^{(\widetilde{T}-1)}\|\boldsymbol\xi_{i^*}\|_2^2 \geq \overline\rho_{y_{i^*},r,i^*}^{(\widetilde{T}-1)} + 0.49\frac{\eta\sigma_\xi^2 d}{nm}|\ell_{i^*}'^{(\widetilde{T}-1)}|,$$

where we use Lemma C.7 and Lemma D.2. Then for $\gamma_{j,r}^{(\widetilde{T})}$, we have

$$\gamma_{j,r}^{(\widetilde{T})} = \gamma_{j,r}^{(\widetilde{T}-1)} - \frac{\eta}{nm}\sum_{i=1}^n \ell_i'^{(\widetilde{T}-1)}\sigma'(\langle\mathbf{w}_{j,r}^{(\widetilde{T}-1)}, y_i\boldsymbol\mu\rangle)\|\boldsymbol\mu\|_2^2 \leq \gamma_{j,r}^{(\widetilde{T}-1)} + \frac{\eta\|\boldsymbol\mu\|_2^2\tilde{C}_\ell|\ell_{i^*}'^{(\widetilde{T}-1)}|}{m}$$

where we use the second claim of Lemma D.4.

Then at iteration $t = 1$, we can show

$$\frac{\gamma_{j,r}^{(1)}}{\overline\rho_{y_{i^*},r,i^*}^{(1)}} \leq \frac{n\|\boldsymbol\mu\|_2^2\tilde{C}_\ell}{0.49\sigma_\xi^2 d} \leq 1.1\tilde{C}_\ell n \cdot \mathrm{SNR}^2 = C_\gamma n \cdot \mathrm{SNR}^2.$$

Now suppose for all $t \leq \widetilde{T} - 1$, we have $\gamma_{j,r}^{(t)}/\overline\rho_{y_{i^*},r,i^*}^{(t)} \leq C_\gamma n \cdot \mathrm{SNR}^2$, then we

Then we can bound

$$\frac{\gamma_{j,r}^{(\widetilde{T})}}{\overline\rho_{y_{i^*},r,i^*}^{(\widetilde{T})}} \leq \max\left\{\frac{\gamma_{j,r}^{(\widetilde{T}-1)}}{\overline\rho_{y_{i^*},r,i^*}^{(\widetilde{T}-1)}}, \frac{n\|\boldsymbol\mu\|_2^2\tilde{C}_\ell}{0.49\sigma_\xi^2 d}\right\} \leq C_\gamma n \cdot \mathrm{SNR}^2.$$

This shows $\gamma_{j,r}^{(\widetilde{T})} \leq C_\gamma n \cdot \mathrm{SNR}^2 \overline\rho_{y_{i^*},r,i^*}^{(\widetilde{T}-1)} \leq C_\gamma\alpha.$ □

## D.1. First Stage

In the first stage, we can lower bound the loss derivatives by a constant $C_\ell$ (by Lemma C.5) and we show both $\rho_{y_i,r,i}^{(t)}$, $r \in \mathcal{S}_i^{(0)}$ and $\gamma_{j,r}^{(t)}$ can grow to a constant order.

**Theorem D.5.** *Under Condition D.1, there exists $T_1 = \Theta(\eta^{-1}nm\sigma_\xi^{-2}d^{-1})$ such that*

- $\overline\rho_{y_i,r,i}^{(T_1)} = \Theta(1)$ *for all $i \in [n]$ and $r \in \mathcal{S}_i^{(0)}$.*

- $\gamma_{j,r}^{(T_1)} = \Theta(n \cdot \text{SNR}^2) = \Theta(1)$ *for all* $j = \pm 1, r \in [m]$.

- $y_i f(\mathbf{W}^{(T_1)}, \mathbf{x}_i) \geq 0$, *for all* $i \in [n]$.

*Proof.* By the update rule of the coefficients, for $r \in \mathcal{S}_i^{(0)}$,

$$\overline{\rho}_{y_i,r,i}^{(t)} = \overline{\rho}_{y_i,r,i}^{(t-1)} - \frac{\eta}{nm} \ell_i'^{(t-1)} \|\boldsymbol{\xi}_i\|_2^2 \geq \overline{\rho}_{y_i,r,i}^{(t-1)} + 0.99 \frac{\eta C_\ell \sigma_\xi^2 d}{nm} \geq 0.99 \frac{\eta C_\ell \sigma_\xi^2 d}{nm} t,$$

where we use the lower bound on loss derivatives and Lemma C.7.

Then with

$$T_1 = 2.1 \eta^{-1} nm C_\ell^{-1} \sigma_\xi^{-2} d^{-1}$$

we can show $\overline{\rho}_{y_i,r,i}^{(t)} \geq 2$. Further we can obtain the upper bound as for all $i \in [n], r \in [m], j = \pm 1$

$$\overline{\rho}_{j,r,i}^{(t)} = \overline{\rho}_{j,r,i}^{(t-1)} - \frac{\eta}{nm} \ell_i'^{(t-1)} \|\boldsymbol{\xi}_i\|_2^2 \leq \overline{\rho}_{j,r,i}^{(t-1)} + 1.01 \frac{\eta \sigma_\xi^2 d}{nm} \leq 1.01 \frac{\eta \sigma_\xi^2 d}{nm} t$$

where we use the upper bound on loss derivatives and Lemma C.7. Under the definition of $T_1$, for all $i, r, j$, we upper bound $\overline{\rho}_{j,r,i}^{(t)} \leq 3 C_\ell^{-1}$.

Next for lower and upper bound for $\gamma_{j,r}^{(t)}$, we first recall the update as for any $j = \pm 1, r \in [m]$,

$$\gamma_{j,r}^{(t)} = \gamma_{j,r}^{(t)} - \frac{\eta}{nm} \sum_{i=1}^n \ell_i'^{(t-1)} \sigma'(\langle \mathbf{w}_{j,r}^{(t-1)}, y_i \boldsymbol{\mu} \rangle) \|\boldsymbol{\mu}\|_2^2.$$

When $\langle \mathbf{w}_{j,r}^{(t-1)}, \boldsymbol{\mu} \rangle \geq 0$,

$$\gamma_{j,r}^{(t)} = \gamma_{j,r}^{(t-1)} + \frac{\eta}{nm} \sum_{i \in \mathcal{S}_1} |\ell_i'^{(t-1)}| \|\boldsymbol{\mu}\|_2^2 \geq \gamma_{j,r}^{(t-1)} + \frac{\eta C_\ell \|\boldsymbol{\mu}\|_2^2}{2m} \geq \frac{\eta C_\ell \|\boldsymbol{\mu}\|_2^2}{2m} t$$

where we use the lower bound on $|\ell_i'^{(t-1)}|$. Similarly, when $\langle \mathbf{w}_{j,r}^{(t-1)}, \boldsymbol{\mu} \rangle \leq 0$, we can obtain the same lower bound as $\gamma_{j,r}^{(t)} \geq \frac{\eta C_\ell \|\boldsymbol{\mu}\|_2^2}{2m}$.

Then at $t = T_1$, we can bound for all $j = \pm 1, r \in [m]$ as

$$\gamma_{j,r}^{(t)} \geq \frac{n \|\boldsymbol{\mu}\|_2^2}{\sigma_\xi^2 d} = n \cdot \text{SNR}^2 = \Omega(1)$$

The upper bound follows from

$$\gamma_{j,r}^{(t)} \leq \gamma_{j,r}^{(t-1)} + \frac{\eta \|\boldsymbol{\mu}\|_2^2}{2m} \leq \frac{\eta \|\boldsymbol{\mu}\|_2^2}{2m} t$$

where we apply the upper bound on $|\gamma_{j,r}^{(t-1)}| \leq 1$. This verifies that at $t = T_1$,

$$\gamma_{j,r}^{(t)} \leq 1.1 C_\ell^{-1} n \cdot \text{SNR}^2 = O(1),$$

which shows at $t = T_1$, both $\overline{\rho}_{y_i,r,i}^{(T_1)}, \gamma_{j,r}^{(T_1)} = \Theta(1)$.

For all samples, by Lemma D.2,

$$y_i f(\mathbf{W}^{(T_1)}, \mathbf{x}_i) \geq \frac{1}{m} \sum_{r=1}^m \left( \overline{\rho}_{y_i,r,i}^{(T_1)} + \gamma_{j,r}^{(T_1)} \right) - 1/C_1 \geq 0,$$

where we let $C_1$ to be sufficiently large. □

### D.2. Second Stage

In the second stage, we show the loss converges and under constant signal-to-noise ratio, we can show the test error can be arbitrarily small at convergence, while for all $T_1 \leq t \leq T^*$, $y_i f(\mathbf{W}^{(t)}, \mathbf{x}_i) \geq 0$, for all $i \in [n]$.

**Theorem D.6.** *Under Condition D.1, there exists a time $t^* \in [T_1, T^*]$ where $T^* = \widetilde{\Theta}(\eta^{-1}\epsilon^{-1}nm\sigma_\xi^{-2}d^{-1})$ such that*

- *Training loss converges, i.e., $L_S(\mathbf{W}^{(t^*)}) \leq \epsilon$.*

- *For all samples, i.e., $i \in [n]$, it satisfies $y_i f(\mathbf{W}^{(t)}, \mathbf{x}_i) \geq 0$.*

- *Test error is small, i.e., $L_D^{0-1}(\mathbf{W}^{(t^*)}) \leq \exp\left(\frac{d}{n} - \frac{n\|\boldsymbol{\mu}\|_2^4}{C_D\sigma_\xi^4 d}\right)$.*

*Proof of Theorem D.6.* The proof of convergence is exactly the same as for the case with label noise, and thus we omit it here. The second claim is also easy to verify given that both $\gamma_{j,r}^{(t)}, \overline{\rho}_{y_i,r,i}^{(t)}$ are monotonically increasing. By Lemma D.2, we can obtain the desired result.

Now we prove the third claim regarding the test error. To this end, we first show for all $T_1 \leq t \leq T^*$ $\gamma_{j,r}^{(t)}/\sum_{i=1}^n \overline{\rho}_{j,r,i}^{(t)} = \Theta(\text{SNR}^2)$ for all $j = \pm 1, r \in [m]$. We prove such a claim by induction. It is clear at $t = T_1$, we have $\sum_{i=1}^n \overline{\rho}_{j,r,i}^{(T_1)} = \Theta(n)$ and $\gamma_{j,r}^{(T_1)} = \Theta(n \cdot \text{SNR}^2)$ for all $j = \pm 1, r \in [m]$. Thus, we can verify the $\gamma_{j,r}^{(T_1)}/\sum_{i=1}^n \overline{\rho}_{j,r,i}^{(T_1)} = \Theta(\text{SNR}^2)$. Now suppose for a given $\widetilde{T} \in [T_1, T^*]$ such that $\gamma_{j,r}^{(t)}/\sum_{i=1}^n \overline{\rho}_{j,r,i}^{(t)} = \Theta(\text{SNR}^2)$ holds for all $T_1 \leq t \leq \widetilde{T} - 1$. Then according tor the update,

$$\sum_{i=1}^n \overline{\rho}_{j,r,i}^{(\widetilde{T})} = \sum_{i:y_i=j}^n \overline{\rho}_{j,r,i}^{(\widetilde{T})} = \sum_{i:y_i=j} \overline{\rho}_{j,r,i}^{(\widetilde{T}-1)} - \frac{\eta}{nm} \sum_{i \in \mathcal{S}_{j,r}^{(\widetilde{T}-1)}} \ell_i'^{(\widetilde{T}-1)}\|\boldsymbol{\xi}_i\|_2^2$$

$$\geq \sum_{i=1}^n \overline{\rho}_{j,r,i}^{(\widetilde{T}-1)} + 0.12 \frac{\eta\sigma_\xi^2 d}{m} \min_{i \in [n]} |\ell_i'^{(\widetilde{T}-1)}|$$

where the second inequality is by $\mathcal{S}_{j,r}^{(0)} \subseteq \mathcal{S}_{j,r}^{(\widetilde{T}-1)}$ and Lemma B.3, Lemma C.7. Similarly, we can upper bound

$$\sum_{i=1}^n \overline{\rho}_{j,r,i}^{(\widetilde{T})} \leq \sum_{i=1}^n \overline{\rho}_{j,r,i}^{(\widetilde{T}-1)} + 1.01 \frac{\eta\sigma_\xi^2 d}{m} \max_{i \in [n]} |\ell_i'^{(\widetilde{T}-1)}|$$

where we Lemma C.7.

On the other hand, we can lower and upper bound

$$\gamma_{j,r}^{(\widetilde{T})} \geq \gamma_{j,r}^{(\widetilde{T}-1)} + \frac{\eta}{2m} \min_{i \in \mathcal{S}_n} |\ell_i'^{(\widetilde{T}-1)}|\|\boldsymbol{\mu}\|_2^2$$

$$\gamma_{j,r}^{(\widetilde{T})} \leq \gamma_{j,r}^{(\widetilde{T}-1)} + \frac{\eta}{2m} \max_{i \in [n]} |\ell_i'^{(\widetilde{T}-1)}|\|\boldsymbol{\mu}\|_2^2$$

This suggests

$$\frac{\gamma_{j,r}^{(\widetilde{T})}}{\sum_{i=1}^n \overline{\rho}_{j,r,i}^{(\widetilde{T})}} \geq \min\left\{\frac{\gamma_{j,r}^{(\widetilde{T}-1)}}{\sum_{i=1}^n \overline{\rho}_{j,r,i}^{(\widetilde{T}-1)}}, \frac{\min_{i \in [n]}|\ell_i'^{(\widetilde{T}-1)}|\|\boldsymbol{\mu}\|_2^2}{2.02\max_{i \in [n]}|\ell_i'^{(\widetilde{T}-1)}|\sigma_\xi^2 d}\right\} \geq \min\left\{\frac{\gamma_{j,r}^{(\widetilde{T}-1)}}{\sum_{i=1}^n \overline{\rho}_{j,r,i}^{(\widetilde{T}-1)}}, \frac{\text{SNR}^2}{2.02\tilde{C}_\ell}\right\}$$

$$= \Omega(\text{SNR}^2)$$

where we use $\max_{i \in [n]}|\ell_i'^{(\widetilde{T}-1)}| \leq \tilde{C}_\ell \min_{i \in [n]}|\ell_i'^{(\widetilde{T}-1)}|$ by second claim of Lemma D.4. Similarly,

$$\frac{\gamma_{j,r}^{(\widetilde{T})}}{\sum_{i=1}^n \overline{\rho}_{j,r,i}^{(\widetilde{T})}} \leq \max\left\{\frac{\gamma_{j,r}^{(\widetilde{T}-1)}}{\sum_{i=1}^n \overline{\rho}_{j,r,i}^{(\widetilde{T}-1)}}, \frac{\tilde{C}_\ell\text{SNR}^2}{0.24}\right\} = O(\text{SNR}^2).$$

This verifies for all $T_1 \leq t \leq T^*$,

$$\gamma_{j,r}^{(t)} / \sum_{i=1}^{n} \overline{\rho}_{j,r,i}^{(t)} = \Theta(\text{SNR}^2).\tag{18}$$

Finally, we prove the test error can be upper bounded. We first write for a test sample $(\mathbf{x}, y) \sim \mathcal{D}_{\text{test}}$,

$$yf(\mathbf{W}^{(t^*)}, \mathbf{x}) = \frac{1}{m} \sum_{r=1}^{m} \left( \sigma(\langle \mathbf{w}_{y,r}^{(t^*)}, y\boldsymbol{\mu} \rangle) + \sigma(\langle \mathbf{w}_{y,r}^{(t^*)}, \boldsymbol{\xi} + \boldsymbol{\zeta} \rangle) \right)$$

$$- \frac{1}{m} \sum_{r=1}^{m} \left( \sigma(\mathbf{w}_{-y,r}^{(t^*)}, y\boldsymbol{\mu}) + \sigma(\langle \mathbf{w}_{-y,r}^{(t^*)}, \boldsymbol{\xi} + \boldsymbol{\zeta} \rangle) \right).$$

For $\langle \mathbf{w}_{y,r}^{(t^*)}, y\boldsymbol{\mu} \rangle$, we can bound

$$\langle \mathbf{w}_{y,r}^{(t^*)}, y\boldsymbol{\mu} \rangle = \gamma_{y,r}^{(t^*)} + \langle \mathbf{w}_{y,r}^{(0)}, y\boldsymbol{\mu} \rangle + \sum_{i=1}^{n} \frac{\langle \boldsymbol{\xi}_i, y\boldsymbol{\mu} \rangle}{\|\boldsymbol{\xi}_i\|_2^2} \overline{\rho}_{y,r,i}^{(t^*)} + \sum_{i=1}^{n} \frac{\langle \boldsymbol{\xi}_i, y\boldsymbol{\mu} \rangle}{\|\boldsymbol{\xi}_i\|_2^2} \underline{\rho}_{y,r,i}^{(t^*)}$$

$$\geq \gamma_{y,r}^{(t^*)} - \sqrt{2\log(12m/\delta)}\sigma_0\|\boldsymbol{\mu}\|_2 - (\|\boldsymbol{\mu}\|_2\sqrt{2\log(6n/\delta)}\sigma_\xi^{-1}d^{-1})\Theta(\text{SNR}^{-2})\gamma_{y,r}^{(t^*)}$$

$$\geq 0.99\gamma_{y,r}^{(t^*)}$$

where we use Lemma B.1 and Lemma B.2 and (18) in the second inequality. The last inequality follows from Condition D.1. With an similar argument, we can show

$$\langle \mathbf{w}_{-y,r}^{(t^*)}, y\boldsymbol{\mu} \rangle \leq -0.99\gamma_{-y,r}^{(t^*)}.$$

Further, we let $g(\boldsymbol{\zeta}) = \sum_{r=1}^{m} \sigma(\langle \mathbf{w}_{-y,r}^{(t^*)}, \boldsymbol{\zeta} + \boldsymbol{\xi} \rangle)$ and by Vershynin (2018, Theorem 5.2.2), we have for any $a \geq 0$,

$$\mathbb{P}(g(\boldsymbol{\zeta}) - \mathbb{E}g(\boldsymbol{\zeta}) > a) \leq \exp(-ca^2\sigma_\xi^{-2}\|g\|_{\text{Lip}}^{-2})$$

where expectation is taken with respect to $\boldsymbol{\zeta} \sim \mathcal{N}(0, \sigma_\xi^2\mathbf{I})$ and $c > 0$ is a constant. To compute the Lipschitz constant, we compute

$$|g(\boldsymbol{\zeta}) - g(\boldsymbol{\zeta}')| = \left| \sum_{r=1}^{m} \sigma(\langle \mathbf{w}_{-y,r}^{(t^*)}, \boldsymbol{\zeta} + \boldsymbol{\xi} \rangle) - \sum_{r=1}^{m} \sigma(\langle \mathbf{w}_{-y,r}^{(t^*)}, \boldsymbol{\zeta}' + \boldsymbol{\xi} \rangle) \right|$$

$$\leq \sum_{r=1}^{m} \left| \sigma(\langle \mathbf{w}_{-y,r}^{(t^*)}, \boldsymbol{\zeta} + \boldsymbol{\xi} \rangle) - \sigma(\langle \mathbf{w}_{-y,r}^{(t^*)}, \boldsymbol{\zeta}' + \boldsymbol{\xi} \rangle) \right|$$

$$\leq \sum_{r=1}^{m} |\langle \mathbf{w}_{-y,r}^{(t^*)}, \boldsymbol{\zeta} - \boldsymbol{\zeta}' \rangle| \leq \sum_{r=1}^{m} \|\mathbf{w}_{-y,r}^{(t^*)}\|_2\|\boldsymbol{\zeta} - \boldsymbol{\zeta}'\|_2$$

where the first inequality is by triangle inequality and the second is by the property of ReLU function. This suggests $\|g\|_{\text{Lip}} \leq \sum_{r=1}^{m} \|\mathbf{w}_{-y,r}^{(t^*)}\|_2$.

In addition, because conditioned on $\boldsymbol{\xi}$, $\langle \mathbf{w}_{-y,r}^{(t^*)}, \boldsymbol{\zeta} + \boldsymbol{\xi} \rangle \sim \mathcal{N}(\langle \mathbf{w}_{-y,r}^{(t^*)}, \boldsymbol{\xi} \rangle, \|\mathbf{w}_{-y,r}^{(t^*)}\|_2^2\sigma_\xi^2)$.

$$\mathbb{E}g(\boldsymbol{\zeta}) = \sum_{r=1}^{m} \mathbb{E}\sigma(\langle \mathbf{w}_{-y,r}^{(t^*)}, \boldsymbol{\zeta} + \boldsymbol{\xi} \rangle)$$

$$= \sum_{r=1}^{m} \left( \langle \mathbf{w}_{-y,r}^{(t^*)}, \boldsymbol{\xi} \rangle \left(1 - \Phi(-\frac{\langle \mathbf{w}_{-y,r}^{(t^*)}, \boldsymbol{\xi} \rangle}{\|\mathbf{w}_{-y,r}^{(t^*)}\|_2\sigma_\xi})\right) + \frac{\|\mathbf{w}_{-y,r}^{(t^*)}\|_2\sigma_\xi}{\sqrt{2\pi}}\exp(-\frac{\langle \mathbf{w}_{-y,r}^{(t^*)}, \boldsymbol{\xi} \rangle^2}{2\|\mathbf{w}_{-y,r}^{(t^*)}\|_2^2\sigma_\xi^2}) \right)$$

$$\leq \sum_{r=1}^{m} \left( \overline{\rho}_{-y,r,i^*}^{(t^*)} + \frac{\|\mathbf{w}_{-y,r}^{(t^*)}\|_2\sigma_\xi}{\sqrt{2\pi}} \right)$$

where the expectation is taken with respect to $\zeta$ and the last equality is due to (Beauchamp, 2018) on expectation of truncated Gaussian. The first inequality is by taking $\boldsymbol{\xi} = \boldsymbol{\xi}_{i^*}$ where $i^* = \arg\max_{i\in[n],r} \overline{\rho}_{-y,r,i}^{(t^*)}$ and use Condition D.1 to remove the leading constant.

Further, we require to bound $\|\mathbf{w}_{-y,r}^{(t^*)}\|_2$ by first bounding

$$\left\|\sum_{i=1}^n \rho_{j,r,i}^{(t^*)}\|\boldsymbol{\xi}_i\|_2^{-2}\cdot\boldsymbol{\xi}_i\right\|_2^2 = \sum_{i=1}^n \rho_{j,r,i}^{(t^*)2}\|\boldsymbol{\xi}_i\|_2^{-2} + 2\sum_{1\le i<i'\le n}\rho_{j,r,i}^{(t^*)}\rho_{j,r,i'}^{(t^*)}\frac{\langle\boldsymbol{\xi}_i,\boldsymbol{\xi}_{i'}\rangle}{\|\boldsymbol{\xi}_i\|_2^2\|\boldsymbol{\xi}_{i'}\|_2^2}$$

$$\le 1.01\sigma_\xi^{-2}d^{-1}\sum_{i=1}^n\rho_{j,r,i}^{(t^*)2} + 4.08\frac{\sqrt{\log(6n^2/\delta)}}{\sigma_\xi^2 d^{3/2}}\sum_{1\le i<i'\le n}|\rho_{j,r,i}^{(t^*)}\rho_{j,r,i'}^{(t^*)}|$$

$$= 1.01\sigma_\xi^{-2}d^{-1}\sum_{i=1}^n\rho_{j,r,i}^{(t^*)2} + 4.08\frac{\sqrt{\log(6n^2/\delta)}}{\sigma_\xi^2 d^{3/2}}\left((\sum_{i=1}^n\rho_{j,r,i}^{(t^*)})^2 - \sum_{i=1}^n\rho_{j,r,i}^{(t^*)2}\right)$$

$$= \Theta(\sigma_\xi^{-2}d^{-1})\sum_{i=1}^n\rho_{j,r,i}^{(t^*)2} + \widetilde{\Theta}(\sigma_\xi^{-2}d^{-3/2})(\sum_{i=1}^n\rho_{j,r,i}^{(t^*)})^2$$

$$\le \Theta(\sigma_\xi^{-2}d^{-1}n^{-1})(\sum_{i=1}^n\overline{\rho}_{j,r,i}^{(t^*)})^2$$

where the first inequality is by Lemma C.7 and Lemma B.1 and the last inequality is by the coefficient orders at $t^*$. Thus, we can bound

$$\|\mathbf{w}_{j,r}^{(t^*)}\|_2 \le \|\mathbf{w}_{j,r}^{(0)}\|_2 + \gamma_{j,r}^{(t^*)}\|\boldsymbol{\mu}\|_2^{-1} + \Theta(\sigma_\xi^{-1}d^{-1/2}n^{-1/2})\sum_{i=1}^n\overline{\rho}_{j,r,i}^{(t^*)}$$

$$= \Theta(\sigma_0\sqrt{d}) + \Theta(\mathrm{SNR}^2\|\boldsymbol{\mu}\|_2^{-1} + \sigma_\xi^{-1}d^{-1/2}n^{-1/2})\sum_{i=1}^n\overline{\rho}_{j,r,i}^{(t^*)}$$

$$\le \Theta(\sigma_\xi^{-1}d^{-1/2}n^{-1/2})\sum_{i=1}^n\overline{\rho}_{j,r,i}^{(t^*)},$$

where the first equality uses Lemma B.2 and (18). The second equality follows from $\sum_{i=1}^n\overline{\rho}_{j,r,i}^{(t^*)} = \Omega(n)$. The last inequality is by Condition D.1 where $\mathrm{SNR}^2\|\boldsymbol{\mu}\|_2^{-1}/(\sigma_\xi^{-1}d^{-1/2}n^{-1/2}) = \sqrt{n}\cdot\mathrm{SNR} = \Theta(1)$ and $\sigma_0\sqrt{d}/(\sigma_\xi^{-1}d^{-1/2}n^{-1/2}\sum_{i=1}^n\overline{\rho}_{j,r,i}^{(t^*)}) = O(\sigma_0\sigma_\xi dn^{-1/2}) = O(1)$ by condition on $\sigma_0$ in Condition D.1.

This gives

$$\frac{\sum_{r=1}^m\sigma(\langle\mathbf{w}_{y,r}^{(t^*)},y\boldsymbol{\mu}\rangle)}{\mathbb{E}g(\zeta)} \ge \frac{\Theta(\sum_{r=1}^m\gamma_{y,r}^{(t^*)})}{\Theta(d^{-1/2}n^{-1/2})\sum_{r,i}\overline{\rho}_{-y,r,i}^{(t^*)} + \sum_{r=1}^m\overline{\rho}_{-y,r^*,i^*}^{(t^*)}} = \Theta(n\cdot\mathrm{SNR}^2)$$

Suppose $n\cdot\mathrm{SNR}^2 \ge C_+$ for $C_+ > 0$ sufficiently large. Then we have $\sum_{r=1}^m\sigma(\langle\mathbf{w}_{y,r}^{(t^*)},y\boldsymbol{\mu}\rangle) - \mathbb{E}g(\zeta) > 0$.

We bound the test error as

$$\mathbb{P}(yf(\mathbf{W}^{(t^*)},\mathbf{x}) < 0)$$

$$\le \mathbb{P}\left(\sum_{r=1}^m\sigma(\langle\mathbf{w}_{-y,r}^{(t^*)},\boldsymbol{\xi}+\zeta\rangle) \ge \sum_{r=1}^m\sigma(\langle\mathbf{w}_{y,r}^{(t^*)},y\boldsymbol{\mu}\rangle)\right)$$

$$= \frac{1}{n}\sum_{i=1}^n\mathbb{P}\left(\sum_{r=1}^m\sigma(\langle\mathbf{w}_{-y,r}^{(t^*)},\boldsymbol{\xi}_i+\zeta\rangle) \ge \sum_{r=1}^m\sigma(\langle\mathbf{w}_{y,r}^{(t^*)},y\boldsymbol{\mu}\rangle)\right)$$

$$= \frac{1}{n}\sum_{i=1}^n\mathbb{P}\left(g_i(\zeta) - \mathbb{E}g_i(\zeta) \ge \sum_{r=1}^m\sigma(\langle\mathbf{w}_{y,r}^{(t^*)},y\boldsymbol{\mu}\rangle) - \mathbb{E}g_i(\zeta)\right)$$

$$\leq \frac{1}{n}\sum_{i=1}^{n}\exp\left(-\frac{c\big(\sum_{r=1}^{m}\sigma(\langle \mathbf{w}_{y,r}^{(t^*)}, y\boldsymbol{\mu}\rangle) - \sum_{r=1}^{m}\overline{\rho}_{-y,r,i}^{(t^*)} - \sigma_\xi/(2\pi)\sum_{r=1}^{m}\|\mathbf{w}_{-y,r}^{(t^*)}\|_2\big)^2}{\sigma_\xi^2(\sum_{r=1}^{m}\|\mathbf{w}_{y,r}^{(t^*)}\|_2)^2}\right)$$

$$\leq \frac{1}{n}\sum_{i=1}^{n}\exp\left(\frac{c}{2\pi} - 0.5c\Big(\frac{\sum_{r=1}^{m}\sigma(\langle \mathbf{w}_{y,r}^{(t^*)}, y\boldsymbol{\mu}\rangle) - \sum_{r=1}^{m}\overline{\rho}_{-y,r,i}^{(t^*)}}{\sigma_\xi \sum_{r=1}^{m}\|\mathbf{w}_{y,r}^{(t^*)}\|_2}\Big)^2\right)$$

$$= \frac{1}{n}\sum_{i=1}^{n}\exp\left(\frac{c}{2\pi} - 0.5c\Big(\frac{\sum_{r=1}^{m}\Theta(\gamma_{y,r}^{(t^*)}) - \sum_{r=1}^{m}\overline{\rho}_{-y,r,i}^{(t^*)}}{\Theta(d^{-1/2}n^{-1/2})\sum_{r=1}^{m}\sum_{i=1}^{n}\overline{\rho}_{j,r,i}^{(t^*)}}\Big)^2\right)$$

$$\leq \frac{1}{n}\sum_{i=1}^{n}\exp\left(\frac{c}{2\pi} + \Big(\frac{\sum_{r=1}^{m}\Theta(n^{-1}\mathrm{SNR}^{-2}\gamma_{y,r}^{(t^*)})}{\Theta(d^{-1/2}n^{-1/2})\sum_{r=1}^{m}\sum_{i=1}^{n}\overline{\rho}_{j,r,i}^{(t^*)}}\Big)^2 - 0.25c\Big(\frac{\sum_{r=1}^{m}\Theta(\gamma_{y,r}^{(t^*)})}{\Theta(d^{-1/2}n^{-1/2})\sum_{r=1}^{m}\sum_{i=1}^{n}\overline{\rho}_{j,r,i}^{(t^*)}}\Big)^2\right)$$

$$= \exp\left(\frac{d}{n} - \frac{n\|\boldsymbol{\mu}\|_2^4}{C_D\sigma_\xi^4 d}\right)$$

where we denote $g_i(\boldsymbol{\zeta}) = \sum_{r=1}^{m}\sigma(\langle \mathbf{w}_{-y,r}^{(t^*)}, \boldsymbol{\zeta} + \boldsymbol{\xi}_i\rangle)$. The third and fourth inequalities are by $(s-t)^2 \geq s^2/2 - t^2$. $\qquad\square$

## E. Early Stopping and Sample Selection

*Proof of Proposition 4.3.* The proof follows the same idea as for Theorem D.6, with the difference that both $\sum_{r\in[m]}\gamma_{j,r}^{(T_1)} = \Theta(m)$ and $\sum_{r\in[m]}\overline{\rho}_{\tilde{y}_i,r,i}^{(T_1)} = \Theta(m)$ and $\sum_{r\in[m]}\gamma_{j,r}^{(T_1)} > \sum_{r\in[m]}\overline{\rho}_{\tilde{y}_i,r,i}^{(T_1)}$ for all $j = \pm 1$ and $i \in [n]$ (from the results of Theorem C.9). Then we can bound the test error directly as

$$\mathbb{P}(yf(\mathbf{W}^{(T_1)}, \mathbf{x}) < 0) \leq \frac{1}{n}\sum_{i=1}^{n}\exp\left(\frac{c}{2\pi} - 0.5c\Big(\frac{\sum_{r=1}^{m}\sigma(\langle \mathbf{w}_{y,r}^{(T_1)}, y\boldsymbol{\mu}\rangle) - \sum_{r=1}^{m}\overline{\rho}_{-y,r,i}^{(T_1)}}{\sigma_\xi \sum_{r=1}^{m}\|\mathbf{w}_{y,r}^{(T_1)}\|_2}\Big)^2\right)$$

$$\leq \frac{1}{n}\sum_{i=1}^{n}\exp\left(\frac{c}{2\pi} - 0.5c\Big(\frac{1}{\Theta(d^{-1/2}n^{1/2})}\Big)^2\right)$$

$$= \exp\left(\frac{c}{2\pi} - \Theta\Big(\frac{d}{n}\Big)\right)$$

$$\leq \exp(-\frac{d}{nC_e})$$

where the second inequality follows from $\sum_{r\in[m]}\gamma_{j,r}^{(T_1)} > \sum_{r\in[m]}\overline{\rho}_{\tilde{y}_i,r,i}^{(T_1)} = \Theta(m)$ and $\|\mathbf{w}_{y,r}^{(T_1)}\|_2 \leq \Theta(\sigma_\xi^{-1}d^{-1/2}n^{1/2})$ where $\sum_{i=1}^{n}\overline{\rho}_{j,r,i}^{(T_1)} \leq \Theta(n)$. The last inequality is by choosing a sufficiently large constant $C_e > 0$.

The claim on sample selection follows easily from Theorem 4.1. $\qquad\square$

## F. Additional Experiments

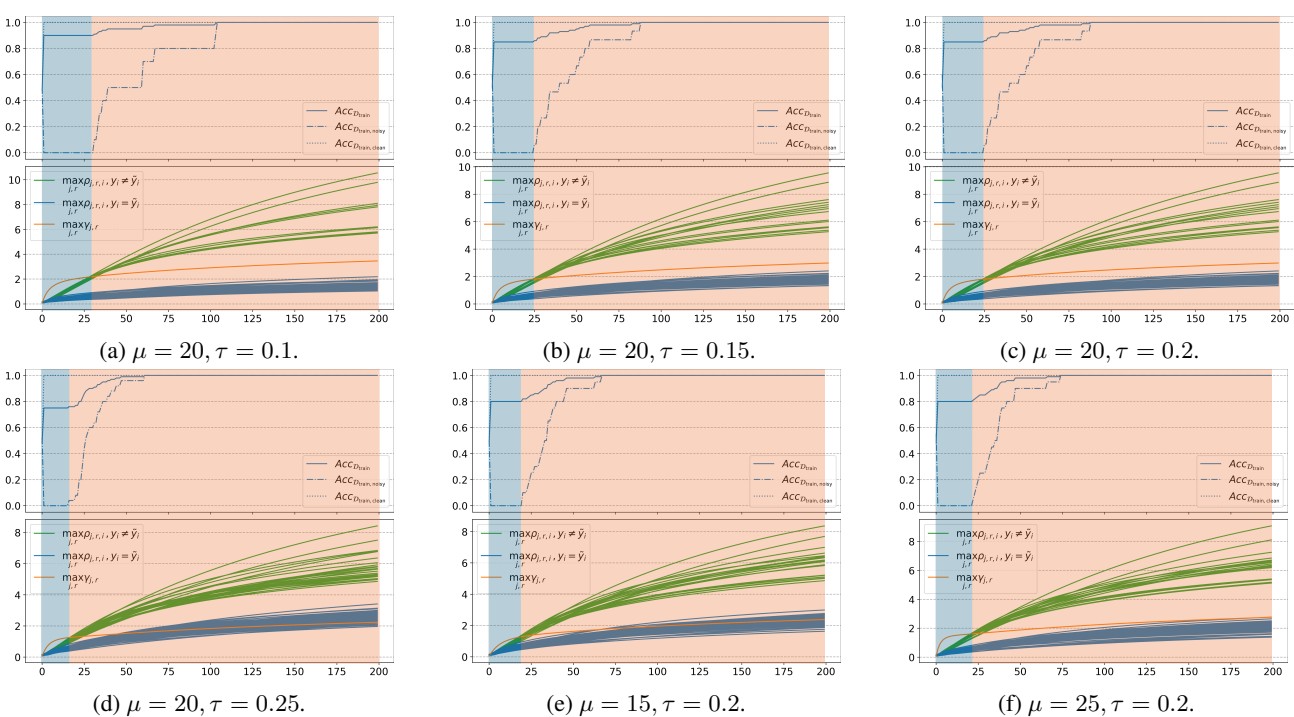

Figure 3: Experiments on synthetic data with varying problem settings, including varying signal strength $\mu$ and label noise ratio $\tau$. We shade the area before noise learning overtakes signal learning of noisy samples in blue. This corresponds to the Stage I in our analysis, where early stopping is beneficial. We shade the area where signal learning exceeds noise learning for noisy samples in orange, which corresponds to Stage II in our analysis.

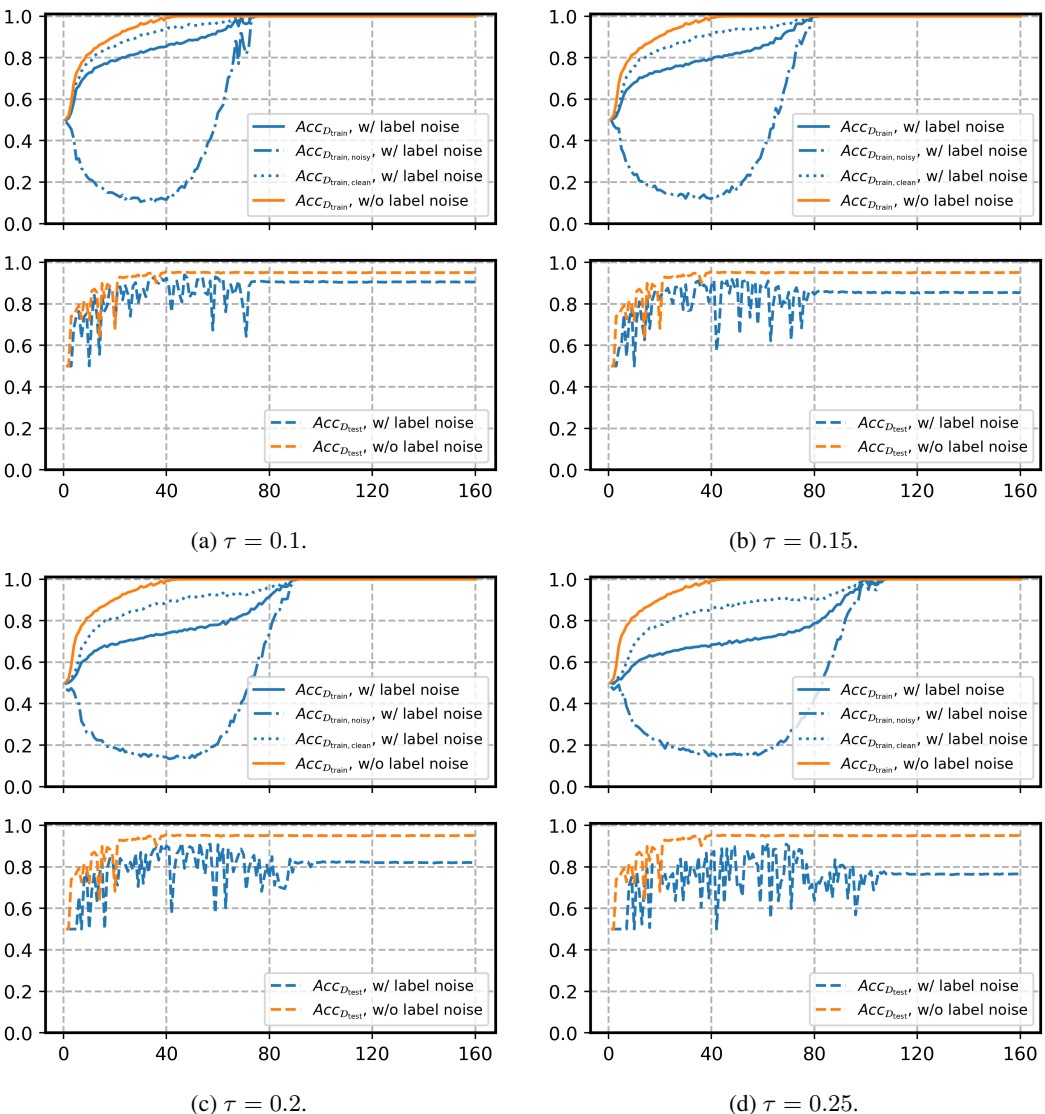

(a) $\tau = 0.1$.

(b) $\tau = 0.15$.

(c) $\tau = 0.2$.

(d) $\tau = 0.25$.

Figure 4: Experiments on CIFAR-10 dataset with varying label noise ratio $\tau$. Across different label noise ratios, we observe a similar pattern that there exist an initial decrease in the training accuracy on noisy samples before an increase to perfect classification. This validates our theoretical findings in real-world settings under various label noise ratios.

