# OpenReview forum: "On the Role of Label Noise in the Feature Learning Process"
_ICML.cc/2025/Conference — ICML 2025 poster_

### Official Review · Reviewer_VPGT · 2025-03-01

**Overall Recommendation:** 4

**Summary:**

This paper provides a theoretical analysis of how label noise impacts the feature learning process in deep neural networks. The authors prove a two-stage learning dynamic for networks trained with label noise:

1. Stage I – The model first learns from clean samples while ignoring noisy ones, leading to good generalization.
2. Stage II – As training continues, the network starts overfitting to noisy labels, resulting in performance degradation.

To support their theoretical framework, the authors analyze the training dynamics of a two-layer convolutional neural network (CNN) on uniform label noise. Their findings highlight the risks of prolonged training with noisy labels and provide a theoretical justification for two commonly used strategies to mitigate label noise: early stopping and sample selection. The paper further validates its claims through synthetic experiments and real-world experiments on CIFAR-10.

**Claims And Evidence:**

Yes

**Essential References Not Discussed:**

Related works have been well discussed.

**Experimental Designs Or Analyses:**

The authors clearly show two-stage phases during training with label noise via synthetic experiments and real-world experiments on CIFAR-10.

**Methods And Evaluation Criteria:**

Yes

**Other Comments Or Suggestions:**

See above.

**Other Strengths And Weaknesses:**

The early stopping technique is widely used in learning with label noise, yet its theoretical justification remains incomplete. This paper provides valuable theoretical support for the phenomenon. However, its impact could be further strengthened if the authors offered a more in-depth explanation, based on their theory, of why the model tends to prioritize learning from clean-labeled samples before incorporating noisy labels.

**Questions For Authors:**

According to Theorem 4.1, $T_1$ marks the transition between the first and second stages. Could $T_1$ be leveraged in practical applications to determine the optimal point for early stopping, even for the synthetic experiment?

**Relation To Broader Scientific Literature:**

The paper provides theoretical support for learning with label noise, especially methods focusing on early stopping.

**Theoretical Claims:**

I haven't carefully checked the proof, but the proof sketch looks reasonable.

---

> ### Author Rebuttal · Authors · 2025-03-31
>
> Thank you for your great efforts on the review of this paper and for recognizing the value of our contributions. We will try our best to address your questions.
>
> **Q1: Suggestions for more explanations of why the model tends to prioritize learning from clean-labeled samples. “However, its impact could be further strengthened if the authors offered a more in-depth explanation, based on their theory, of why the model tends to prioritize learning from clean-labeled samples before incorporating noisy labels.”**
>
> **A1**: Thank you for the suggestion. We will include a more in-depth explanation in the proof sketch. Intuitively, the signal $\mu$ is shared across all samples, and thus it is easier to be captured by the model. In contrast, noise is more complicated and varies across samples, so learning the noisy samples requires the neural network to spend more time exploring a refined structure to memorize the noise for each sample.
>
> **Q2: Questions for applying $T_1$ as the early stopping point in practice. “According to Theorem 4.1, $T_1$ marks the transition between the first and second stages. Could $T_1$ be leveraged in practical applications to determine the optimal point for early stopping, even for the synthetic experiment?”**
>
> **A2**: Thank you for the question. As also discussed in *Reviewer P2ej A4*, we acknowledge that directly computing $T_1$ in real-world scenarios is not feasible due to the complexity of the training dynamics and unknown data distribution. However, our theory suggests that validation accuracy can serve as a practical surrogate for identifying $T_1$. Based on Theorem 4.1 and Theorem 4.2, $T_1$ marks the transition beyond which generalization performance begins to degrade. Therefore, early stopping at the point of maximum validation accuracy aligns with our theoretical insights and the common practice of early stopping.

---

### Official Review · Reviewer_P2ej · 2025-03-08

**Overall Recommendation:** 3

**Summary:**

The paper is concerned with a theoretical analysis of gradient descent when the training samples contain label noise as well as features uncorrelated to the correct class. The chosen methodology is feature learning theory, where highly simplified learning problems are introduced in which the features carrying signal are clearly separated from features representing noise, and a simple network architecture is trained. In this setup, the convergence of the loss can be formally analyzed in great detail as a function of the type of the example. The paper presents theorems that suggest that in the setup they study signal features are picked up by the network much faster than noise features, which creates an initial stage with generalizable learning and a second stage with overfitting.

**Claims And Evidence:**

The theoretical claims in the specific simplified setup are supported by thorough proofs. I have not checked these proofs in detail but the formulation is of high quality. In the experimental section the CIFAR10 experiments are not well described (training setup, etc) and the results there have very little overlap with the presented theory in general (data model, network architecture, training setup, specific assumptions, etc) so they provide only very indirect support for the usefulness of the theory, they basically reiterate well known empirical behavior.

**Essential References Not Discussed:**

No suggestions here that are essential.

**Experimental Designs Or Analyses:**

As mentioned above the CIFAR10 experiments are not described in enough detail and present observations that have been rather well known from research into memorization, as also mentioned in the paper itself (eg that early stopping works for label noise). So overall, the theory does not have a really strong experimental support, but in general it is in line with empirical observations (clean samples converge first).

**Methods And Evaluation Criteria:**

The theoretical methods are appropriate.

**Other Comments Or Suggestions:**

Typo: line 146, I guess F_{-1}(W_{-1}…

**Other Strengths And Weaknesses:**

As for strengths, the paper targets an interesting problem and offers a theoretical approach in a novel framework (feature learning theory) that captures some of the empirically well-known observations such as that clean samples converge first, and early stopping is a good way of filtering noisy labels.

As for weaknesses, the theoretical results of the paper are in a very simplistic setup in which signal and noise are very clearly separated not only in the input, but also within the network architecture. The learning algorithm is GD with a constant learning rate. This makes the analysis possible but the usual question remains: does this approach have any explanatory power?

**Questions For Authors:**

About the early stopping result (prop. 4.3). I was wondering whether it is really early stopping in the sense that although stopping at T_1 is good, T_1 is not necessarily an early stopping point (in the usual sense of having a local maximum accuracy (or  min loss) on a validation set), and that is actually a rather crucial question, because finding T_1 is not possible directly in practice. Do you have any thoughts on that?

**Relation To Broader Scientific Literature:**

The paper can be positioned as a purely theoretical work in the area of feature learning theory, a specific methodology to study the convergence during training. Its contributions lie in studying label noise in this framework and also applying a number of technical innovations in the analysis.

**Theoretical Claims:**

I did not check the proofs. The formalism is well designed and it is possible to follow, although not too easily, eg the two-layer CNN is described extremely briefly, the supplementary could have expanded this a bit. The rest of the presentation is similarly dense, it requires a lot of effort to follow.

---

> ### Author Rebuttal · Authors · 2025-03-31
>
> Thank you for your great efforts on the review of this paper and for appreciating our novelty and contributions! We will try our best to address your questions.
>
> **Q1: Concerns about the presentation. “The formalism is well designed and it is possible to follow, although not too easily, eg the two-layer CNN is described extremely briefly.”**
>
> **A1**: Thank you for your feedback. We will provide a more detailed and clearer explanation of the two-layer CNN in our revision.
>
> **Q2: Concerns about the real-world experiments. “the CIFAR10 experiments are not well described (training setup, etc) and the results there have very little overlap with the presented theory in general (data model, network architecture, ...) so they provide only very indirect support for the usefulness of the theory, they basically reiterate well known empirical behavior.”**
>
> **A2**: Thanks. We would like to clarify that the primary purpose of the CIFAR-10 experiment is to support and illustrate the theoretical findings, rather than to introduce new empirical discoveries. As the reviewer noted, there are differences between the theoretical setup and the CIFAR-10 training setup. However, the observed results align with our theoretical insights and help demonstrate the broader applicability of our theory beyond the simplified setting.
>
> Additionally, in response to the reviewer’s suggestion, we will include more details on the CIFAR-10 experimental setup in the appendix to improve clarity and transparency.
>
> **Q3: Concerns about the simplified theoretical setup. “the theoretical results of the paper are in a very simplistic setup in which signal and noise are very clearly separated not only in the input, but also within the network architecture. The learning algorithm is GD with a constant learning rate. This makes the analysis possible but the usual question remains: does this approach have any explanatory power?”**
>
> **A3**: We sincerely appreciate the reviewer's thoughtful critique regarding our theoretical setup. We address these concerns from three perspectives:
>
> * **Well-Established in the Community**. Our simplified framework follows standard practice in the deep learning theory community. Similar settings (e.g., GD with constant LR, architecturally separated signal/noise) have been adopted in many influential works [1, 2, 3, 4] to gain fundamental insights into neural network behavior. These simplifications are essential for isolating and rigorously analyzing core learning dynamics.
>
> * **Empirically Validated**. Our theoretical results are not merely abstract; they align closely with our real-world experiments (see Section 5). This consistency underscores the validity of our framework and demonstrates its explanatory power in practical scenarios.
>
> * **Challenges of Extension**. While we agree that extending the analysis to more complex settings (e.g., deeper model, realistic dataset) would be valuable, it remains highly challenging due to the nonlinear, nonconvex nature of neural network training. Many existing approaches rely on unrealistic assumptions—such as infinite-width networks [5, 6] or layer-wise training [7]—which can obscure the phenomena our work aims to clarify.
>
> From a broader view, simplified models are foundational across many scientific disciplines, from idealized models in physics to mean-field theories in neuroscience. Thus, we believe our theory provides valuable insights and substantial explanatory power, offering a solid foundation for further research in more realistic settings.
>
> **Q4: “Typo: line 146, I guess F_{-1}(W_{-1}…”**
>
> **A4**: Thanks. We will correct this typo.
>
> **Q5: Questions for applying $T_1$ as the early stopping point in practice. “I was wondering ... T_1 is not necessarily an early stopping point (in the usual sense of having a local maximum accuracy (or min loss) on a validation set), ..., because finding T_1 is not possible directly in practice. Do you have any thoughts on that?”**
>
> **A5**: Insightful question. We agree that directly computing $T_1$ in real-world scenarios is not feasible due to the complexity of the training dynamics and unknown data distribution. However, our theory suggests the existence of a point $T_1$, beyond which further training may degrade generalization performance. This implies that validation accuracy can serve as a practical surrogate for identifying $T_1$, which aligns with the common practice of early stopping at the point of maximum validation accuracy, as mentioned by the reviewer. Indeed, our theory explains the effectiveness of the common practice, which further strengthens the applicability of our theory. We will clarify this point in the revised paper.
>
> ### Reference
> Due to space limit, we defer the reference list to the Reference part in *Rebuttal to Reviewer Q9d8*.

---

### Official Review · Reviewer_Q9d8 · 2025-03-14

**Overall Recommendation:** 3

**Summary:**

This paper analyzes the training dynamics of a (custom) two-layer ConvNet under binary class-conditional label noise. The main result is a a two-stage characterization of "first fitting all clean samples, then overfitting to noisy samples".

**Claims And Evidence:**

Yes, the claims were proved.

**Essential References Not Discussed:**

Essential references are included.

The setup is built upon Kou et al. (2023), it would be good to indicate that in the main text.

**Experimental Designs Or Analyses:**

Yes.

**Methods And Evaluation Criteria:**

Overall makes sense to me.

**Other Comments Or Suggestions:**

The data distribution and Two-Layer ReLU CNN setup seems to follow the one in Kou et al. (2023) and Cao et al. (2022), please indicate that.

**Other Strengths And Weaknesses:**

My understanding is that the paper uses the setup and tools from Kou et al. (2023) to analyze the training dynamics under label noise.

The paper is well written and the results are clear to me. Though novelty is a bit limited, I'm glad to see a new analysis of label noise learning.

**Questions For Authors:**

1) line 142: The "Two-Layer ReLU CNN":
$$
F_j(W_j, x) = \frac{1}{m} \sum_{r=1}^{m} \left( \sigma \langle w_{j,r}, y\mu \rangle + \sigma \langle w_{j,r}, \xi \rangle \right)
$$
seems to corresponds to CNN with stride $d$ and then global average pooling.
I understand that this is the setup considered in Kou et al. (2023), but I am still curious about whether it's possible to consider a more general form that resembles a "standard" CNN.

2) line 376, synthetic experiments: why choose a high-dimensional case $d >> n$? Will the same result hold for $d < n$?

**Relation To Broader Scientific Literature:**

This paper analyzes the training dynamics of NN under label noise beyond the "lazy training" regime, which is a good contribution to the literature.

**Theoretical Claims:**

I've skimmed through the proofs, but have not checked the details.

Insights and proof sketches were provided in the main text, which is good for readers.

---

> ### Author Rebuttal · Authors · 2025-03-31
>
> Thank you for your great efforts on the review of this paper and for recognizing the value of our contributions. We will try our best to address your questions.
>
> **Q1: Suggestion for acknowledgement in setup part. “The setup is built upon Kou et al. (2023), it would be good to indicate that in the main text.”**
>
> **A1**: Thank you for the suggestion. We will include additional commentary in the Preliminary section to explicitly acknowledge that we built our theoretical setup upon [1] and [2].
>
> **Q2: Question on the possibility of extending to a standard CNN. “I understand that this is the setup considered in Kou et al. (2023), but I am still curious about whether it's possible to consider a more general form that resembles a "standard" CNN.”**
>
> **A2**: Good question. We would like to clarify and address the points as follows:
> * **Justification of the Two-Layer CNN Model**. First, we clarify that the two-layer ReLU CNN architecture, while simplified, is a well-established model in deep learning theory for analyzing training dynamics [1, 2, 3, 4]. This choice strikes a balance between capturing essential learning phenomena and maintaining mathematical tractability.
> * **Clarification on Global Average Pooling**. We respectfully point out that our theoretical analysis does not consider global average pooling. In fact, we can rewrite the two-layer ReLU CNN model as
> $$
> f(\mathbf W, \mathbf x) = \frac{1}{2m} \sum_{r = 1}^{2m} a_r \left(\sigma(\langle \mathbf w_{r}, y\mu\rangle) +\sigma(\langle \mathbf w_{r}, \xi\rangle) \right).
> $$
> where $a_r$ represents the second-layer weights, which are fixed as $a_r = -1$ for $r \leq m$ and $a_r = +1$ for $r > m$. This is different to global average pooling where $a_r = 1$ for all $r$. We will clarify this point in the revised paper.
> * **Extension to "Standard" CNN**. We have also carefully considered the extension to a more “standard” CNN. Based on the review’s context, we interpret the term “standard” into two aspects:
>      1. *Flexible stride in the convolutional layer*. We acknowledge that variable strides are commonly used in practice. Extending the data distribution to a multi-patch case allows us to use a larger stride (increase the size of the receptive field), which is more aligned with standard CNN architectures.
>      2. *Multi-layer architectures*. We recognize the importance of deeper networks. However, the analysis of training dynamics beyond two-layer CNN remains an open challenge and often requires unrealistic assumptions, e.g., infinite width [5, 6] layer-wise training [7].
>
> Overall, while we acknowledge the value of extending our analysis to a more complex setup, this lies beyond the scope of our paper and is deferred to future work.
>
> **Q3: Concerns about the synthetic experiments. “line 376, synthetic experiments: why choose a high-dimensional case $d \gg n$? Will the same result hold for $d < n$?”**
>
> **A3**: Good question. First, we clarify that we choose $d \gg n$ according to our theoretical setting (Condition 4.1), where we analyze the over-parameterization regime.
>
> Nevertheless, we followed your suggestion to conduct a new experiment with $n > d$. Specifically, we choose $n = 200$, $d = 180$. Results are presented at [Figure (please click this link)](https://anonymous.4open.science/r/ICML2025-label-noise-26D3). Consistently with our main paper, we observe a two-stage behaviour, where the model initially fits the clean samples and then overfits to the noisy ones.
>
> **Q4: Suggestions for emphasizing Appendix A. “the authors may need to emphasize Appendix A because it's crucial when evaluating novelty of this work.”**
>
> **A4**: Thank you for the suggestion. We will follow your suggestion to include a “technical novelty” paragraph to highlight Appendix A in our revised paper.
>
> ### Reference
>
> [1] Cao et al. Benign overfitting in two-layer convolutional neural networks. NeurIPS 2022.
>
> [2] Kou et al. Benign overfitting in two-layer relu convolutional neural networks. ICML 2023.
>
> [3] Zou et al. The benefits of mixup for feature learning. ICML 2023.
>
> [4] Chen et al. Understanding and improving feature learning for out-of-distribution generalization. NeurIPS 2023.
>
> [5] Du et al. Gradient descent finds global minima of deep neural networks. ICML 2019.
>
> [6] Allen-Zhu et al. A convergence theory for deep learning via over-parameterization. ICML 2019.
>
> [7] Chen et al. Which layer is learning faster? a systematic exploration of layer-wise convergence rate for deep neural networks. ICLR 2023.

---

> > ### Comment · Reviewer_Q9d8 · 2025-04-04
> >
> > Thank you for the detailed response.

---

### Decision · Program_Chairs · 2025-05-01

**Decision:**

Accept (poster)

**Comment:**

This paper presents a rigorous theoretical analysis of how label noise impacts feature learning in neural networks. Using a signal-noise decomposition of the data distribution and a two-layer CNN model, the authors identify a two-phase training dynamic: in the first phase, the network fits clean samples and learns useful features; in the second, it begins to overfit to noisy labels, leading to degraded generalisation. The theoretical findings also provide justification for common heuristics like early stopping and sample selection, and are supported by experiments on both synthetic and real datasets.

Reviewers appreciated the clarity of the setup, the relevance of the analysis, and the combination of theory with supporting experiments. However, some concerns remained—particularly around the restrictive assumptions (e.g., model architecture and data distribution), and how these might limit generalisability to more complex or realistic settings. While these limitations are acknowledged, the core insights are sound and the contribution to understanding label noise in deep learning is meaningful.

Overall, the paper makes a valuable and timely contribution to the theory of training dynamics under label noise, and I recommend acceptance.